# Identifying and Analyzing Task-Encoding Tokens in Large Language Models

## Abstract

In-context learning (ICL) has emerged as an effective solution for few-shot learning with large language models (LLMs). Previous research suggests that LLMs perform ICL by analogizing from the provided demonstrations, similar to how humans learn new tasks. However, how LLMs leverage demonstrations to specify a task and learn a corresponding computational function through ICL remains underexplored. Drawing from the way humans learn from content-label mappings in demonstrations, we categorize the tokens in an ICL prompt into content, stopword, and template tokens, with the latter two typically ignored by humans due to their uninformative nature. Our goal is to identify the type of tokens whose representations highly and directly influence LLM's performance, a property we refer to as *task-encoding*. By ablating representations from the attention of the test example, we find that the representations of informative content tokens have less influence on performance, while template and stopword tokens are more prone to be task-encoding tokens, which contrasts with the human attention to informative words. We further give evidence about the function of task-encoding tokens by showing that their representations aggregate information from the content tokens. Moreover, we demonstrate experimentally that lexical meaning, repetition, and structural cues are the main distinguishing characteristics of these tokens. Our work sheds light on how LLMs learn to perform tasks from demonstrations and deepens our understanding of the roles different types of tokens play in LLMs.

## 1 Introduction

In-context learning (ICL) has become a popular technique employed with large language models (LLMs) (Brown et al., 2020). However, ICL has been shown to be unstable in that slight changes to the in-context prompts (e.g., reordering of demonstrations) can lead to substantial differences in performance (Lu et al., 2022; Zhang et al., 2022). This circumstance is difficult to control due to a lack of understanding of the model's working mechanisms, leaving us uncertain about the exact process by which LLMs learn to infer a task specification from demonstrations and produce a computation function to implement that task specification. Previous papers have begun to explore this issue, focusing on specific aspects such as the label space (Min et al., 2022) and the hidden states of the last prompt token (Hendel et al., 2023; Todd et al., 2023), but have been limited in scope.

In this work, we aim to conduct a comprehensive study on how LLMs extract information that is valuable for improving task performance from demonstrations. Drawing from the way humans learn through content-label mappings in demonstrations, we categorize the tokens in an ICL prompt into content, stopword (Sarica & Luo, 2021), and template tokens, with the latter two typically ignored by humans due to their uninformative nature (Lenartowicz et al., 2014; Whitaker et al., 2018; Chirimuuta, 2021). With these categories in mind, we ablate the representations of different token types from the attention of ICL test examples, masking partial information during the model's task-solving process, as shown in Figure 1. This ablation is intended to identify the types of tokens whose representations LLMs directly depend on to achieve high-level performance, thereby explaining how LLMs learn from demonstrations. These tokens critical for performance are referred to as **task-encoding tokens**.

Results of these experiments provide evidence that template tokens and stopword tokens are the most prone to be task-encoding tokens as ablating their representations significantly decreases performance. In contrast, content tokens have a negligible impact on performance, as the task performance is not

affected when their representations are eliminated from the attention of the test examples. This finding is counterintuitive since the template and stopword tokens do not possess the information found in the demonstrations. To further explain this, we study the relationship among different types of tokens through ablation experiments that cut off the information flow between different kinds of tokens. We show that content tokens are indirectly leveraged by LLMs during ICL through aggregating their information into the representations of task-encoding tokens.

Beyond identifying task-encoding tokens, we analyze them to better understand how they are leveraged by LLMs. We first investigate the relationship among task-encoding tokens to determine whether these tokens work partially or depend on each other. By ablating the representation of different parts of template tokens, we confirm that it is necessary to retain all these representations for preserving the task performance. We also investigate the characteristics which differentiate them from other tokens. We find the following three distinguishing characteristics: the **lexical meaning** of tokens as it relates to the task being solved, the **repetition** of tokens throughout the prompt, and the **structural cues** which the tokens provide to the prompt. Our findings indicate that the lexical meaning, repetition, and structural cues of task-encoding tokens contribute to task performance across all model sizes, suggesting that these characteristics are a crucial part of the identity of task-encoding tokens and hence disrupting them may lead to performance degradation.

Our work reveals that we can identify and characterize the types of tokens whose representations are the most important in directly maintaining ICL task performance. This identification of task-encoding tokens suggests that previous claims about ICL are more nuanced, in that representations of tokens beyond label words (Wang et al., 2023) may also directly impact the task performance. We investigate the characteristics of lexical meaning, repetition, and structural cue re-

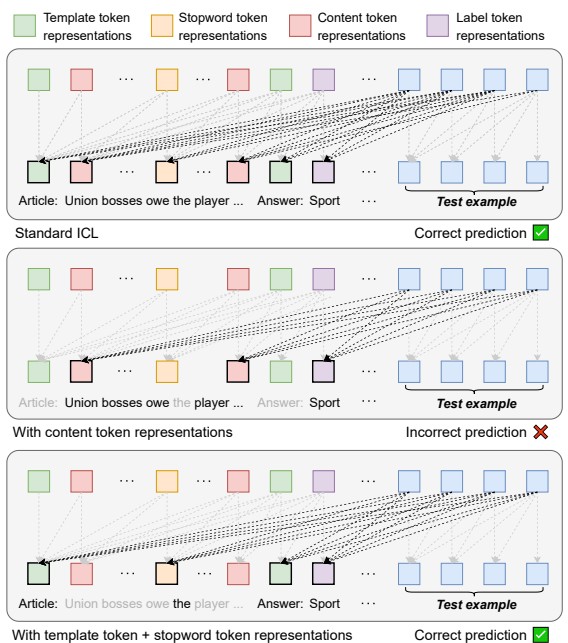

Figure 1: An illustration of the 4-way text classification on AGNews with different parts of its 4-shot ICL demonstrations masked with respect to the attention of the test example. Masking the representations of what we call the template and stopword tokens from the attention of the test example leads to a significant drop in performance while masking representations of the content tokens leaves the performance relatively unchanged. The dash lines represent the attention between every pair of tokens while those from the test example to the ICL prompt are unshaded.

lated to task-encoding tokens which allow us to partially explain the importance as it relates to task performance of task-encoding tokens and help us better understand how to avoid performance instability while using ICL. Our findings deepen the understanding of the roles different types of tokens play in large language models, suggesting future work based on leveraging specific representations of different token types. Code and data will be released in the camera-ready version.

## 2 RELATED WORK

### 2.1 WORKING MECHANISMS OF IN-CONTEXT LEARNING

Since the proposal of in-context learning (Brown et al., 2020), its working mechanisms have been extensively studied by the research community (Min et al., 2022; Liu et al., 2021; Olsson et al., 2022; Bhattamishra et al., 2023). Min et al. (2022) suggest that demonstrations primarily provide the label space, the distribution of the input text, and the format of the sequence for the test example. They argue that the precise ground truth labels do not have significant importance. In contrast, Yoo et al. (2022) propose a differing view, stating that the impact of the ground truth labels depends on the experimental

configuration. Xie et al. (2021) explain ICL as implicit Bayesian inference, while Akyürek et al. (2022) explore ICL learning process using linear models. Theoretical explanations (Guo et al., 2023; Bai et al., 2023; Li et al., 2023b) and gradient descent explanations have also been proposed. Mao et al. (2024) analyze in-context learning from the perspective of data generation. The perspective of supported training data is also leveraged to analyze ICL (Han et al., 2023). Zhao et al. (2024) propose to use coordinate systems to understand the working mechanism of in-context learning. Zhou et al. (2023) propose a comprehensive survey on the interpretation and analysis of in-context learning.

Additional analyses exploring different aspects of ICL have also been studied. For instance, order sensitivity where task performance fluctuates based on the order of the same ICL demonstrations has been identified as a limitation of ICL (Lu et al., 2022). Yan et al. (2023) propose that repetitive patterns in the prompt could affect the ICL performance in both positive and negative ways. Pan et al. (2023) analyze the ICL process by disentangling it into task recognition and task learning. Madaan & Yazdanbakhsh (2022) propose to define text and patterns while using counterfactual prompting for attributing token importance in chain-of-thought techniques.

Our work investigates the working process of ICL in LLMs at inference time, demonstrating that certain specific tokens are more likely to possess representations that could affect the processing of the final test sample, improving the task performance.

## 2.2 FUNCTION VECTORS OF IN-CONTEXT LEARNING

Todd et al. (2023) and Hendel et al. (2023) provide evidence of function vectors that store information used to solve a task in ICL. They probe and extract the hidden representations of the final tokens in the prompt. These vectors can then be added to, or used to replace, the corresponding vectors in a zero-shot example, achieving results comparable to those obtained when the model uses all demonstrations as context. In addition, Liu et al. (2023a) also propose using an in-context vector to represent the target task and applying feature shifting to query examples. They first feed each input and its corresponding target separately into an LLM, then concatenate all the latent states. A PCA method is applied to derive a vector that is more closely aligned with the task. Finally, Wang et al. (2023) propose that label words in the demonstration examples function as information anchors by aggregating the information from previous demonstrations and providing it to the test example. This finding suggests that we may view label tokens as satisfying our definition of task-encoding tokens.

All these previous studies either solely focus on a single token (i.e., the last prediction prompt token or label token) of the ICL prompt or treat the entire demonstration as a single unit, neglecting the other tokens within it. Our research focuses on all the tokens in the prompt and reveals that there are additional tokens with specific characteristics whose representations significantly affect the final ICL performance.

## 3 PRELIMINARIES

### 3.1 NOTATION

In-context learning (ICL) is a technique that enables large language models (LLMs) to perform tasks in a few-shot manner by placing task demonstrations (e.g., input-output pairs) in the context fed to a large language model (Brown et al., 2020). In ICL, these demonstrations are leveraged to construct a structured prompt that guides the model in predicting the final answer. Formally, the structural prompt consists of the following components: the instruction $\mathbf{I}$, the templates $\mathbf{T}^{\text{in}}$, $\mathbf{T}^{\text{out}}$, and the demonstrations $\mathbf{D}_i^{\text{in}}$, $\mathbf{D}_i^{\text{out}}$, where $i$ denotes the $i^{\text{th}}$ demonstration while in and out refer to the input text and output labels, respectively. These prompt components are concatenated to form the ICL prompt, $P$, as shown in Table 1. During inference, the templated version of the test example without its answer, $\mathbf{T}^{\text{in}} \cdot \mathbf{D}_{\text{test}}^{\text{in}} \cdot \mathbf{T}^{\text{out}}$, is appended to the ICL prompt and then sent to the large language model to predict the corresponding answer, where $\cdot$ denotes the concatenation of token sequences.

### 3.2 EXPERIMENTAL SETTINGS

In this section, we describe the experimental setup for all of our experiments.

Table 1: An example of the components of a 2-shot ICL prompt in the AGNews dataset.

| Component notation | Component example |
| --- | --- |
| I | Classify the news articles into the categories of World, Sports, Business, and Technology.\n\n |
| $\mathbf{T}^{\text{in}}$ | Article: {$\mathbf{D}^{\text{in}}$}\n |
| $\mathbf{T}^{\text{out}}$ | Answer: {$\mathbf{D}^{\text{out}}$}\n\n |
| $\mathbf{D}_1^{\text{in}}$ | Radio veteran Karmazin joins Sirius. Sirius Satellite Radio Inc. named former Viacom Inc. president Mel... |
| $\mathbf{D}_1^{\text{out}}$ | Business |
| $\mathbf{D}_2^{\text{in}}$ | Numbers point to NY. NEW YORK - The New York Yankees can achieve two milestones with one more victory... |
| $\mathbf{D}_2^{\text{out}}$ | Sports |
| | Classify the news articles into the categories of World, Sports, Business, and Technology. |
| ICL
Prompt | Article: Radio veteran Karmazin joins Sirius. Sirius Satellite Radio Inc. named former Viacom Inc. president Mel...
Answer: Business |
| | Article: Numbers point to NY. NEW YORK - The New York Yankees can achieve two milestones with one more victory...
Answer: Sports |

For the datasets, we consider the most widely used text classification datasets used by previous studies (Zhao et al., 2021). For topic classification, we use the 4-way and 14-way datasets AGNews and DBPedia (Zhang et al., 2015). For textual entailment, we use the 3-way CB (De Marneffe et al., 2019) and 2-way RTE dataset (Dagan et al., 2005). We also use SST2 (Socher et al., 2013) and TREC (Voorhees & Tice, 2000) for sentiment and question classification tasks.

For each dataset, we randomly select 4 training demonstrations from the training set using 15 different random seeds limited by the computational cost of the inference stage of LLMs. For testing, we evaluate each setting on 500 randomly selected test examples. We show that this sample size is sufficient by comparing experiment results with 500 test examples and with the whole dataset using OpenLlama 3B and Llama 7B models, shown in the Appendix H. Instruction prompt $\mathbf{I}$ is retained in all the different kinds of ablations since it is essential for enhancing the classification performance of the model (Yin et al., 2023). We keep one fixed $\mathbf{I}$ in each task for all the main results while providing additional experimental results with different $\mathbf{I}$ in Appendix I to show that changing $\mathbf{I}$ would not affect the main findings of this paper.

For the LLMs, we utilize the 7B, 13B, and 33B versions of the Llama model and a 3B OpenLlama model. We also included additional results using Llama 2 7B, Llama 2 13B, and Mistral 7B models in the Appendix D. Models after supervised fine-tuning process are also tested in Appendix E. All the experiments are conducted using a single A100 80G GPU. For the 13B and 33B models, we apply 8-bit quantization to ensure the model fits into a single GPU. The experiments are conducted using Huggingface Transformers (Wolf et al., 2020).

## 4 IDENTIFICATION OF TASK-ENCODING TOKENS

In this section, we aim to find the task-encoding tokens in the ICL prompt. We first formally define what task-encoding tokens are. Then, we structurally categorize all the tokens in the prompt into three types: template, stopword, and content tokens. We provide supporting evidence from the view of task performance to show that the template and stopword tokens are the most prone to be task-encoding tokens. Finally, we demonstrate that the information of content tokens serve to indirectly contribute to the performance by being propagated into the representations of the task-encoding tokens by LLMs.

### 4.1 DEFINITION OF TASK-ENCODING TOKEN

Conceptually, task-encoding tokens are defined as tokens whose representations encode the task-solving procedures. However, it is difficult to directly determine whether this information is encoded in the hidden representations of LLMs. Previous work has used performance variations to determine whether certain representations are related to downstream tasks (Todd et al., 2023; Hendel et al., 2023). Hence, as a practical proxy, we measure the performance variation before and after incorporating the representations of specific tokens into the attention scope of the test example, and define task-encoding tokens as the tokens that lead to both a noticeable performance improvement when their representations are included in the attention of test examples and performance degradation when they are excluded from the attention of test examples.

Let $M$ be a large language model and $D$ be a classification dataset. Further, recall that the definition of the prompt, $P$, we use to conduct ICL from Section 3.1 may be written as

$$P = \mathbf{I} \cdot \mathbf{T}^{\text{in}} \cdot \mathbf{D}_1^{\text{in}} \cdot \mathbf{T}^{\text{out}} \cdot \mathbf{D}_1^{\text{out}} \cdot \ldots \cdot \mathbf{T}^{\text{in}} \cdot \mathbf{D}_n^{\text{in}} \cdot \mathbf{T}^{\text{out}} \cdot \mathbf{D}_n^{\text{out}} \tag{1}$$

where $\cdot$ denotes the concatenation of token sequences.

We define $H_P$ as the set of representations of each token in the ICL prompt $P$ and $H_{\text{test}}$ as the set of representations of the test demonstration which is appended to $P$ for prediction (i.e., $\mathbf{T}^{\text{in}} \cdot \mathbf{D}_{\text{test}}^{\text{in}} \cdot \mathbf{T}^{\text{out}}$). In addition, we let $H_{\text{attend}} \subseteq H_P$ be some set of representations which $M$ may attend to from $H_{\text{test}}$ at inference time while performing ICL. For instance, $H_{\text{attend}} := H_{\mathbf{I}}$ would imply that, when $M$ is predicting the label of the test demonstration, the attention from the test example is restricted to the prompt's instruction token representations.

To provide a practical definition for the task-encoding tokens, we let $\mathbf{Acc}(M, D, H_{\text{attend}})$ be the accuracy achieved by a LLM $M$ when performing ICL on the classification dataset $D$ where the only representations which the test example may attend to at inference time are $H_{\text{attend}}$. Given a partition $\mathcal{P}$ of $H_P$, we say that a set of tokens $H^* \in \mathcal{P}$ is *task-encoding* if

$$\mathbf{Acc}(M, D, H^*) \gg \mathbf{Acc}(M, D, \emptyset) \qquad \& \qquad (2)$$

$$\mathbf{Acc}(M, D, H_P) \gg \mathbf{Acc}(M, D, H_P - H^*) \qquad (3)$$

We note that examining the possibility of each token being task-encoding (i.e., $|H^*| = 1$) in an ICL prompt would be computationally intractable. We instead categorize all the tokens based on the role they play in the prompt and identify which types of tokens are more likely to be task-encoding.

## 4.2 TOKEN TYPES

We categorize ICL tokens based on the structure of the ICL prompt, following our notation in Table 1. Firstly, we find it natural to categorize tokens based on the structure of ICL prompts where the tokens from the demonstration examples $\mathbf{D}^{\text{in}}$ and the labels $\mathbf{D}^{\text{out}}$ are separated by template tokens from $\mathbf{T}^{\text{in}}$ and $\mathbf{T}^{\text{out}}$. Second, $\mathbf{D}^{\text{in}}$ can be subdivided into content and stopword tokens, with the latter typically providing less useful information and often being ignored when humans use analogy to learn specific tasks. Guided by these intuitions, we categorize all the tokens in the ICL prompt into template tokens, stopword tokens, and content tokens. The definitions of all types of tokens are shown as follows:

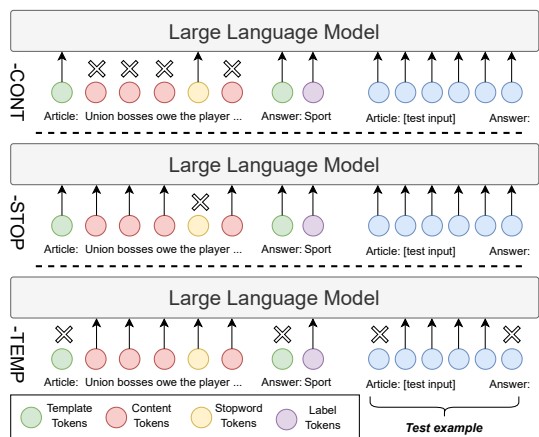

Figure 2: An illustrative example of the token-level ablation methods we use to analyze the working mechanism of task-encoding tokens.

**Template tokens (TEMP):** In defining template tokens, we include all the tokens which serve as templates for the ICL prompt. This includes the tokens in $\mathbf{T}^{\text{in}}$ and $\mathbf{T}^{\text{out}}$, as shown in Table 1.

**Stopword tokens (STOP):** In defining stopword tokens, we include punctuation and conjunction words, such as [,], [.], etc., in the ICL prompt. We use the stopword tokens which appear in the instructions[1]. The stopword token list is shown in Appendix F.

**Content tokens (CONT):** In defining content tokens, we include all the tokens from $\mathbf{D}^{\text{in}}$ except for the ones that are already stopword tokens. We use the term "content tokens" as they convey the meaningful information found in the demonstrations.

Researchers might typically expect content tokens to be critical, as they contain the primary information from the demonstrations. However, in the following experiments, we find that the representations of template and stopword tokens have the greatest impact on performance.

The above categorization is also supported by the attention distribution shown in previous work (Wang et al., 2023; Liu et al., 2023b; Ge et al., 2023), where the representations of template tokens are highly attended when predicting the answer during ICL, while stopword token representations possess a different role from the content token representations in the language modeling task.

---

[1]Ablation with the complete NLTK (Loper & Bird, 2002) stopwords list are conducted in Appendix F.

## 4.3 ABLATION ON TOKEN TYPES

To determine which token types are more likely to be task-encoding tokens whose representations directly affect the final performance significantly, we design two experiments which ablate representations or tokens based on token types. The first involves keeping and masking representations of different token types from the attention of the test example. The second involves dropping the various kinds of tokens from the ICL prompt. The main purpose of the first experiment is to identify the task-encoding tokens defined in Section 4.1, while the second experiment aims to cut off the information propagation of different types of tokens to further explore the working of task-encoding tokens. Illustrations of these two methods which we refer to as representation-level and token-level ablations are shown in Figure 1 and Figure 2. More detailed examples for the representation-level ablation is provided in Appendix C.

### 4.3.1 REPRESENTATION-LEVEL ABLATION

Our first ablation stems from the intuition that if LLMs essentially rely on the representations of certain token types to achieve high-level performance, then the model should perform the target task adequately with only these representations. Meanwhile, performance should decrease significantly if we remove them from the attention of the test example. Hence, we first pass the entire ICL prompt to the LLM and then restrict the attention of the test example such that the LLM may only attend to the representations of tokens of a particular type (or types)[2] during its solving of the task. We compute task performances with every possible ablation combination, removing the representations of one (e.g., Standard ICL − TEMP) or two token types (e.g., Zero-shot + CONT[3]) from the attention of the test example. All the task performances and the averaged relative performance changes are reported, shown in Table 2 and Table 3. An illustration of this set of experiments is shown in Figure 1.

Overall, these results demonstrate that **template and stopword tokens** are more likely to be task-encoding tokens than content tokens, conforming to our definition in Equ.(2) and Equ.(3). On the one hand, template token representations are crucial for LLMs' task-solving ability via ICL, achieving an average performance 39.8% higher than the zero-shot baseline by only utilizing these representations at inference time. If the representations of stopword tokens are further included (i.e., Standard ICL−CONT), the performance is nearly equivalent to that of the Standard ICL. In contrast, content token representations only bring an average improvement of 10.7%. On the other hand, the performance decreases the most with Standard ICL−TEMP, highlighting the significance of template tokens again[4]. Considering the number of tokens in each type, content tokens exhibits a way larger number than the other two tokens. Hence, the averaged impacts of the template and stopword tokens provide concrete evidences that they are more prone to be task-encoding tokens.

Table 2: The accuracy results of the representation-level ablation study where, for example, + TEMP refers to allowing attention only to template tokens. All values are presented as percentages. Except where noted with *, all test statistics reported correspond to p-values < 0.05. The best results are in bold.

| Models | Setting | AGNews | SST2 | TREC | DBPedia | RTE | CB | △Avg. |
|---|---|---|---|---|---|---|---|---|
| | Zero-shot | 22.0 | 20.0 | 23.6 | 5.4 | 44.4 | 1.8 | 19.5 |
| OpenLlama | + CONT | 26.2 | 52.1 | 30.1 | 7.4 | 51.9 | 37.9 | +14.8 |
| 3B | + STOP | 36.7 | 82.9 | 32.0* | 52.4 | **58.8** | **56.2** | +33.7 |
| | + TEMP | **56.5** | **86.7** | 27.1 | **62.2** | 56.4 | 52.3 | **+37.4** |
| | Zero-shot | 25.0 | 29.2 | 41.4 | 0.0 | 54.2 | 3.6 | 25.6 |
| Llama | + CONT | 32.4 | 57.9 | 42.5 | 12.5 | 55.5 | 46.1 | +15.6 |
| 7B | + STOP | 57.3 | 83.7 | 49.8 | 43.0 | 55.9 | 50.7 | +31.1 |
| | + TEMP | **70.8** | **90.2** | **58.4** | **66.2** | **66.3** | **73.5** | **+45.3** |
| | Zero-shot | 59.0 | 18.0 | 37.0 | 0.0 | 0.0 | 0.0 | 19.0 |
| Llama | + CONT | 27.7 | 52.4 | 33.5 | 10.9 | 61.7 | 41.7 | +19.0 |
| 13B | + STOP | 72.2 | 73.5 | 46.8 | 50.7 | 58.6 | 30.6 | +36.4 |
| | + TEMP | **80.0** | **92.3** | **58.6** | **76.9** | **68.5** | **47.7** | **+51.7** |
| | Zero-shot | 70.2 | 88.6 | 60.6 | 30.2 | 58.1 | 19.6 | 54.6 |
| Llama | + CONT | 24.4 | 61.7 | 62.1 | 10.5 | 65.2 | 63.6 | −6.7 |
| 33B | + STOP | 72.9 | 92.7 | **66.7*** | 69.1 | 69.6 | 63.0 | +17.7 |
| | + TEMP | **80.5** | **95.2** | 65.2 | **75.2** | **79.0** | **80.0** | **+24.6** |

Rare exception cases appear when performance is relatively poor with Standard ICL (e.g., OpenLlama 3B in TREC). In some cases, masking the representations of the content tokens brings even better performance than the Standard ICL method, which is possibly due to the elimination of noisy information in the demonstration content. Another interesting observation is that the performance results of Standard ICL−STOP and Standard ICL−CONT where the attention to the content and stopword tokens is ablated respectively are close, with an average difference of only 5.4%. This

---

[2] Since $\mathbf{D}^{out}$ tokens have been shown to significantly impact performance (Wang et al., 2023), we always preserve the attention on the representations of the $\mathbf{D}^{out}$ tokens.

[3] Removing two types of tokens from Standard ICL is equivalent to adding the other type to Zero-shot.

[4] Both STOP and TEMP include the "\n" token; we mask the attention to the "\n" token as long as one of them is ablated in this set of experiments. Analyses about this experimental setting are shown in Appendix G.

indicates that the representation of stopword tokens may contain overlapping information with their preceding content tokens. We believe that this could enable LLMs to model long sequences without significant architectural changes (e.g., using stopword token representations as synthesis checkpoints) and leave the verification of this hypothesis to future work.

**Results for generation and question answering (QA) tasks**: Besides the classification tasks, we also present results in machine translation and QA tasks to show that our findings can also be extended to text generation tasks. Results and analyses are attached to Appendix J and Appendix K.

### 4.3.2 TOKEN-LEVEL ABLATION

In this section, we modify the ICL prompt by removing certain types of tokens from the ICL prompt[5] to further investigate the relationship between different kinds of tokens, by cutting off the information flow between the representations of different tokens, shown in Figure 2. When we ablate the template tokens, we preserve the answer and next-line tokens in the templates to maintain a basic separator between the demonstration inputs and outputs. Results averaged on all the datasets are presented in Figure 3. Detailed results on each dataset could be seen in Appendix M.

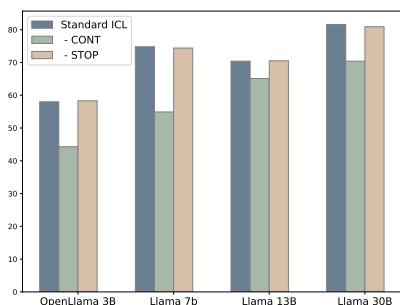

Figure 3: Results of the token-level ablation where, for example, −STOP refers to the ablation where stopword tokens are dropped from the ICL prompt. Models **without template tokens** consistently yielded **an accuracy of 0%** and are thus omitted from this figure.

Table 3: The accuracy results of the representation-level ablation study where, for example, − TEMP refers to allowing attention only to content and stopword tokens. All values are presented as percentages. The results showing the greatest decrease from ablation are underlined.

| Models | Setting | AGNews | SST2 | TREC | DBPedia | RTE | CB | △Avg. |
|---|---|---|---|---|---|---|---|---|
| OpenLlama 3B | Standard ICL | 63.7 | 91.2 | 21.9 | 61.9 | 57.4 | 52.0 | 58.0 |
| | − CONT | 58.2 | 86.9 | 27.6 | 61.9 | 56.5 | 51.7 | −0.9 |
| | − STOP | 51.8 | 78.9 | 28.8 | 30.3 | 53.6 | 45.2 | −9.9 |
| | − TEMP | 26.2 | 52.1 | 30.1 | 7.4 | 51.9 | 37.9 | −23.8 |
| Llama 7B | Standard ICL | 82.4 | 94.3 | 63.5 | 68.7 | 68.6 | 71.3 | 74.8 |
| | − CONT | 77.9 | 91.5 | 58.5 | 66.5 | 67.8 | 74.4 | −2.0 |
| | − STOP | 78.5 | 88.7 | 39.3 | 66.7 | 60.6 | 60.4 | −9.1 |
| | − TEMP | 20.8 | 58.2 | 32.4 | 11.6 | 54.4 | 46.0 | −37.6 |
| Llama 13B | Standard ICL | 81.6 | 94.3 | 60.0 | 76.1 | 70.6 | 39.9 | 70.4 |
| | − CONT | 81.4 | 93.1 | 58.9 | 75.7 | 69.6 | 45.1 | +0.2 |
| | − STOP | 79.8 | 85.8 | 64.4 | 73.6 | 64.5 | 40.2 | −2.4 |
| | − TEMP | 27.8 | 52.4 | 33.5 | 10.9 | 63.1 | 45.6 | −31.5 |
| Llama 33B | Standard ICL | 85.0 | 96.5 | 68.1 | 78.4 | 78.5 | 83.3 | 81.6 |
| | − CONT | 82.3 | 95.4 | 64.9 | 76.1 | 80.4 | 82.0 | −1.5 |
| | − STOP | 84.8 | 94.9 | 62.1 | 77.3 | 70.5 | 74.4 | −4.3 |
| | − TEMP | 24.4 | 61.7 | 60.6 | 10.5 | 67.7 | 68.5 | −32.7 |

Our first finding from this ablation is that removing template tokens causes the LLMs to completely lose their ability to solve tasks via ICL with an overall task accuracy performance of 0% for all sizes and all tasks. We hypothesize that this is because the model no longer has an explicit cue to generate the target label, which is further discussed in Section 5.2.3. In this case, if we add back the last prompt token after the next-line token, the results return to their original level due to the introduction of a template token. This finding confirms previous claims that preserving the format of ICL prompts plays a significant role in retaining the task performance (Min et al., 2022). Notably, even without stopword or content tokens, the model can still acquire limited predictive ability.

In addition, the contrast between the representation-level and token-level ablation also indicates that information is being propagated from the representations of content tokens to the representations of the task-encoding tokens. The representations of the template tokens and stopword tokens alone (i.e., Standard ICL − CONT in Figure 3) are less effective at encoding tasks (i.e., leading to worse performance) without incorporating the information from the content token representations (i.e., Standard ICL − CONT in Table 3).

These findings provide us with additional insights about how LLMs leverage different kinds of tokens during ICL. Firstly, this circumstance means that even though the representations of the content tokens are not directly used when LLMs predict the answer, the encoding of these tokens contribute to the final performance indirectly through being aggregated into the representations of the task-encoding tokens. Secondly, it also suggests that LLMs prefer to utilize the the task-encoding tokens to aggregate the indirect information from the demonstration rather than others (i.e., content

---

[5]For template tokens, this includes *both* the tokens in the demonstrations and the test example to maintain their consistency. We included the analyses of only ablating the tokens in the demonstrations in Appendix N.

tokens). It is their incorporation of this information that makes them better at encoding tasks, partially explaining the working mechanism of in-context learning.

### 4.4 FINDINGS

To summarize, we find that template and stopword tokens are the most likely to be task-encoding tokens. Specifically, the representations of template tokens contribute significantly to performance improvement. Meanwhile, the representations of stopword tokens play a more supportive role in the spectrum of task-encoding tokens by summarizing the information of content tokens. In contrast, the representations of content tokens do not directly facilitate task-solving, but they are aggregated into the representations of the other two types of tokens. We discuss the possible applications of these findings in Appendix O. Furthermore, this finding raises additional questions: 1) Are all the task-encoding tokens working together? 2) What are the characteristics for a token to be perceived by a LLM as a task-encoding token?

## 5 ANALYSES OF TASK-ENCODING TOKENS

To answer the above questions, we provide analyses of the tokens whose representations we believe mainly store information that directly affects the performance of a task drastically. We focus on the template tokens since, as evidenced by the findings in Section 4.3.1 and 4.3.2, **their representations are the most important to maintaining task performance**. Our analyses include the effects of different parts of template tokens on the performance and the distinguishing characteristics of them.

### 5.1 EFFECTS OF DIFFERENT TASK-ENCODING TOKENS

In this section, we aim at examining the relationship among the representations of different task-encoding tokens. To achieve this, we test the effectiveness (i.e., how much they could affect the downstream task performance) of each part of task-encoding tokens to see if they could work without each other.

To achieve this, we ablate the representations of each task-encoding token, similar to Section 4.3.1. In Section 4.3.1, we assume that the label token $\mathbf{D}^{out}$ is needed for ICL to achieve performance results on par with Standard ICL, as suggested by previous work (Wang et al.,

Table 4: Ablation for different template token representations with and without $\mathbf{D}^{out}$, presented as percentages. The results showing the greatest impact from ablation are underlined.

| Models | Settings | RTE | | Settings | RTE | |
|---|---|---|---|---|---|---|
| | | with ":" | w/o ":" | | with ":" | w/o ":" |
| Llama 7B | TEMP with $\mathbf{D}^{out}$ | 66.3 | 59.5 | TEMP w/o $\mathbf{D}^{out}$ | 40.7 | 42.5 |
| | $-\mathbf{T}^{in}$ | 58.9 | 56.5 | $-\mathbf{T}^{in}$ | 43.7 | 49.9 |
| | $-\mathbf{T}^{out}$ | 56.7 | 56.7 | $-\mathbf{T}^{out}$ | 56.0 | 55.6 |
| Llama 13B | TEMP with $\mathbf{D}^{out}$ | 68.5 | 59.8 | TEMP w/o $\mathbf{D}^{out}$ | 57.5 | 53.7 |
| | $-\mathbf{T}^{in}$ | 65.5 | 59.0 | $-\mathbf{T}^{in}$ | 53.6 | 52.8 |
| | $-\mathbf{T}^{out}$ | 61.2 | 58.4 | $-\mathbf{T}^{out}$ | 54.8 | 53.7 |
| Llama 33B | TEMP with $\mathbf{D}^{out}$ | 79.0 | 77.1 | TEMP w/o $\mathbf{D}^{out}$ | 71.8 | 65.8 |
| | $-\mathbf{T}^{in}$ | 77.4 | 75.4 | $-\mathbf{T}^{in}$ | 70.6 | 67.8 |
| | $-\mathbf{T}^{out}$ | 72.8 | 70.0 | $-\mathbf{T}^{out}$ | 67.0 | 61.3 |

2023). However, it is still not known how the other task-encoding tokens affect the performance without $\mathbf{D}^{out}$. Hence, we divide our experiments by including or excluding the label tokens $\mathbf{D}^{out}$ to further specifically investigate their effectiveness. We present the results on RTE datasets in Table 4 while full results are shown in Appendix P.

Overall, the above experiments show that the task-encoding tokens **should be utilized together** to provide the best performance and that removing some of them would cause **performance degeneration** or **instability** issues. From the results with $\mathbf{D}^{out}$, it is observed that all the template tokens (i.e., $\mathbf{T}^{in}$, $\mathbf{T}^{out}$, and ":") contribute to the final performance. Removing one of them would cause a performance degradation. From the results without $\mathbf{D}^{out}$, the performance becomes less predictable, where adding back a template token (e.g., ":") does not always bring performance improvements. Moreover, in some datasets, models without $\mathbf{D}^{out}$ can still achieve relatively high performance. These results show that representations of other template tokens may also be seen as information anchors whose representations aggregate and serve information to the final prediction of LLMs, broadening the conclusions of Wang et al. (2023) who claim that only answer tokens serve as information anchors.

## 5.2 CHARACTERISTICS OF TASK-ENCODING TOKENS

With the task-encoding tokens identified, we turn to determining what characteristics distinguish them from other tokens. By better understanding what characteristics of task-encoding tokens lead them to affect task performance, we provide the community with insights on how to best leverage LLMs for ICL (e.g., What principles should practitioners be using when designing prompt templates?). We hypothesize that the following characteristics are critical for a token to be leveraged as task-encoding tokens: **lexical meaning** referring to the task-related lexical meaning of a task-encoding token, **repetition** referring to the multiple appearances of the task-encoding tokens in the prompt, and **structural cue** referring to how task-encoding tokens format the ICL prompt, shown in Table 1, into structured text.

We design several experiments to test whether these characteristics affect the impact of task-encoding tokens on the task performance, by disrupting each characteristic in the ICL prompts. A characteristic is related if there is a performance drop after the disruption. The disruption is achieved by replacing the template tokens with different kinds of random string templates, shown in Table 5. We use 5 different random string templates which are attached to Appendix R and average all the results for each setting.

Table 5: An example of the ICL template with random strings used in AGNews.

| Settings | Notations | Examples |
|---|---|---|
| $\text{Random}_{\text{fixed}}$ | $\mathbf{T}^{\text{in}}$ 
 $\mathbf{T}^{\text{out}}$ | dsafjkldafdsajk: {$\mathbf{D}^{\text{in}}$}\n 
 reqwiorewsdafjl: {$\mathbf{D}^{\text{out}}$}\n\n |
| Swap | $\mathbf{T}^{\text{in}}$ 
 $\mathbf{T}^{\text{out}}$ | Answer: {$\mathbf{D}^{\text{in}}$}\n 
 Article: {$\mathbf{D}^{\text{out}}$}\n\n |
| $\text{Random}_{\text{nonfixed}}$ | $\mathbf{T}^{\text{in}}_1$ 
 $\mathbf{T}^{\text{out}}_1$ 
 $\mathbf{T}^{\text{in}}_2$ 
 $\mathbf{T}^{\text{out}}_2$ 
 $\mathbf{T}^{\text{in}}_t$ 
 $\mathbf{T}^{\text{out}}_t$ | dsafjkldaasdfjkl: {$\mathbf{D}^{\text{in}}$}\n 
 xiadfjdsalgfweqrjl: {$\mathbf{D}^{\text{out}}$}\n\n 
 ewqroudajfsdafq: {$\mathbf{D}^{\text{in}}$}\n 
 yufoufgaddavfdnsl: {$\mathbf{D}^{\text{out}}$}\n\n 
 vcxnkfgahvczxkl: {$\mathbf{D}^{\text{in}}$}\n 
 dafhglajfdvcaol: {$\mathbf{D}^{\text{out}}$}\n\n |

### 5.2.1 LEXICAL MEANING

A task-encoding token might be more impactful on the performance with specific lexical meaning. One possible hypothesis is that if the token carries specific task-related meanings like "Article" and "Answer", it is more likely to serve as a task-encoding token.

To verify if lexical meanings could affect the formation of task-encoding tokens, we 1) Replace the tokens from $\mathbf{T}^{\text{in}}$ and $\mathbf{T}^{\text{out}}$ with the same random strings across the different demonstrations (**Random$_{\text{fixed}}$**), thus completely disrupting the lexical characteristic of these tokens; 2) Swap $\mathbf{T}^{\text{in}}$ and $\mathbf{T}^{\text{out}}$ (**Swap**), thus partially disrupting the lexical characteristic of these tokens.

Table 6: Results validating the effect of lexical meanings of template tokens, presented as percentages. The results showing the greatest decrease during the disruption are underlined.

| Models | Settings | AGNews | SST2 | TREC | DBPedia | RTE | CB | Avg. |
|---|---|---|---|---|---|---|---|---|
| OpenLlama 3B | Standard ICL | 63.7 | 91.2 | 21.9 | 61.9 | 57.4 | 52.0 | 58.0 |
| | Swap | 64.4 | 86.8 | 21.7 | 58.7 | 60.6 | 54.6 | 57.8 |
| | Random$_{\text{fixed}}$ | 57.5 | 71.4 | 32.4 | 51.2 | 53.3 | 49.8 | 52.6 |
| Llama 7B | Standard ICL | 82.4 | 94.3 | 63.5 | 68.7 | 68.6 | 71.3 | 74.8 |
| | Swap | 70.2 | 11.4 | 44.3 | 58.2 | 64.5 | 50.1 | 49.8 |
| | Random$_{\text{fixed}}$ | 19.5 | 11.4 | 13.2 | 7.4 | 19.7 | 21.7 | 15.5 |
| Llama 13B | Standard ICL | 81.6 | 94.3 | 60.0 | 76.1 | 70.6 | 39.9 | 70.4 |
| | Swap | 81.5 | 67.4 | 36.4 | 75.9 | 69.1 | 52.1 | 63.7 |
| | Random$_{\text{fixed}}$ | 52.1 | 76.8 | 27.7 | 48.9 | 55.7 | 34.5 | 49.3 |
| Llama 33B | Standard ICL | 85.0 | 96.5 | 68.1 | 78.4 | 78.5 | 83.3 | 81.6 |
| | Swap | 84.5 | 94.9 | 60.8 | 75.5 | 68.0 | 55.5 | 73.2 |
| | Random$_{\text{fixed}}$ | 78.7 | 92.5 | 52.2 | 75.8 | 68.9 | 41.1 | 68.2 |

Shown in Table 6, we observe that for smaller models (OpenLlama 3B) disrupting the lexical meaning of tokens would slightly impact task performance. For larger models, the disruption causes more significant drops in performance. Specifically, Llama 7B is particularly sensitive to the lexical meaning of tokens and demonstrates poorer performance when semantics are disturbed via random strings or swapping. Therefore, the lexical meaning of tokens is likely to play a role in their task-encoding nature, especially in the case of larger models.

### 5.2.2 REPETITION

The impact of task-encoding tokens could also be influenced by their repetition throughout the ICL prompt. Intuitively, via the attention mechanism, repetitive patterns are more likely to propagate information through the processing of text. Yan et al. (2023) propose self-reinforcement in in-context learning, also suggesting that repetition could be a significant factor in in-context learning.

Table 7: Results validating the effect of repetitive patterns, presented as percentages. We bold the highest accuracy for each classification task and model size.

| Models | Settings | AGNews | SST2 | TREC | DBPedia | RTE | CB | Avg. |
|---|---|---|---|---|---|---|---|---|
| OpenLlama 3B | Random$_{\text{fixed}}$ | **57.5** | **71.4** | **32.4** | **51.2** | **53.3** | **49.8** | **52.6** |
| | Random$_{\text{nonfixed}}$ | 30.2 | 71.4 | 17.1 | 18.6 | 47.9 | 47.7 | 38.8 |
| Llama 7B | Random$_{\text{fixed}}$ | **19.5** | 11.4 | **13.2** | **7.4** | **19.7** | 21.7 | **15.5** |
| | Random$_{\text{nonfixed}}$ | 15.5 | **11.6** | 10.4 | 1.8 | 4.6 | **25.6** | 11.6 |
| Llama 13B | Random$_{\text{fixed}}$ | **52.1** | **76.8** | **27.7** | **48.9** | **55.7** | **34.5** | **49.3** |
| | Random$_{\text{nonfixed}}$ | 32.1 | 34.5 | 19.2 | 6.0 | 21.0 | 32.8 | 24.3 |
| Llama 33B | Random$_{\text{fixed}}$ | **78.7** | **92.5** | **52.2** | **75.8** | **68.9** | 41.1 | **68.2** |
| | Random$_{\text{nonfixed}}$ | 78.5 | 87.5 | 46.3 | 63.1 | 63.6 | **46.1** | 64.2 |

We experiment with the repetition characteristic by comparing the results of the previously discussed **Random$_{\text{fixed}}$** experiment with an experiment replacing each $\mathbf{T}^{\text{in}}$ and $\mathbf{T}^{\text{out}}$ with different

random strings (**Random**$_{\text{nonfixed}}$), thus breaking the repetition of template tokens present in ICL demonstrations.

We see from Table 7 that without consistent repetition of the task-encoding tokens, the performance for most models decreases. This decrease in performance suggests that information necessary for maintaining the performance of the task may not have been properly accumulated and stored in the representations of the template tokens. These experiments demonstrate that repetitive patterns significantly influence the impact of task-encoding tokens.

Additionally, we conducted supplemental experiments using template tokens with specific lexical meanings for comparison, as detailed in Appendix S. The results are consistent with the previous findings, further reinforcing our claim that repetition is a key characteristic of task-encoding tokens.

### 5.2.3 STRUCTURAL CUE

Beyond lexical meaning and repetition, the performance influence of task-encoding tokens may also be affected by how they format ICL prompts. Similar to our definition of template and stopword tokens, ICL prompts are often formatted with structural cues that assist the model in differentiating between elements with distinct roles, such as task inputs and target labels, within a demonstration. For instance, template tokens (i.e., $\mathbf{T}^{\text{in}}$ and $\mathbf{T}^{\text{out}}$) delimit the presentation of demonstration examples and labels in ICL prompts. Meanwhile, stop-

Table 8: One-shot experimental results validating the effect of structural cues, presented as percentages. Models **without template tokens** consistently yielded **an accuracy of 0%** and are thus omitted from this table.

| Models | Settings | AGNews | SST2 | TREC | DBPedia | RTE | CB | Avg. |
|---|---|---|---|---|---|---|---|---|
| OpenLlama 3B | Standard ICL | 70.7 | 51.7 | 40.4 | 53.5 | 50.2 | 48.6 | 53.3 |
| | Random$_{\text{fixed}}$ | 47.5 | 51.8 | 32.6 | 19.4 | 51.8 | 42.4 | 40.9 |
| Llama 7B | Standard ICL | 72.3 | 77.4 | 54.1 | 64.7 | 53.0 | 64.4 | 64.3 |
| | Random$_{\text{fixed}}$ | 3.9 | 16.9 | 3.5 | 9.6 | 16.9 | 10.4 | 10.2 |
| Llama 13B | Standard ICL | 82.0 | 72.0 | 60.1 | 75.9 | 60.4 | 18.8 | 70.1 |
| | Random$_{\text{fixed}}$ | 46.1 | 47.5 | 25.0 | 50.8 | 47.5 | 21.4 | 39.7 |
| Llama 33B | Standard ICL | 85.3 | 88.3 | 71.2 | 75.5 | 64.1 | 45.5 | 76.9 |
| | Random$_{\text{fixed}}$ | 69.7 | 53.0 | 37.8 | 72.8 | 53.0 | 37.6 | 54.0 |

word tokens (e.g., ",", ".", ":", etc) help structure the content words into different sentence components by marking the beginning or end of sentences. Examples of how task-encoding tokens naturally delimit an ICL prompt are shown in Appendix U. These structural cues are similar to those found in an LLM's pretraining data (e.g., column names in SQL tables). As a result, we suspect that pretraining on such data enables the structuring nature of the task-encoding tokens to be recognized, causing its representations to store higher-level information.

To measure the effect of the structuring characteristic of task-encoding tokens, we perturb the structure of one-shot prompts in two stages. We use the one-shot prompt setting to eliminate the repetition characteristic which may act as a confounding factor in our results. Firstly, we disrupt the lexical meaning of templates tokens similar to Section 5.2.1. We begin with this disruption since the meaning of tokens also help LLMs distinguish the different parts of a prompt. Subsequently, we remove all the template tokens from the prompt to eliminate any source of structure.

The results in Table 8 demonstrate that performance decreases after disrupting the structural cue characteristics, highlighting the importance of structural cues for these tokens in influencing the final performance. In particular, consistent with the findings in Section 4.3.2, removing all template tokens results in 0% performance due to the complete elimination of structural cues. Supplemental experiments in Appendix T are provided to better support the characteristic of structural cue from the perspective of representation-level ablation.

## 6 CONCLUSION

In this paper, we have provided a fine-grained characterization of task-encoding tokens, whose representations LLMs directly depend on to achieve high-level performance. Through a series of experiments, we have examined the roles of template tokens and stopword tokens within ICL as potential task-encoding tokens. Our findings add nuance to previous claims made about ICL, for example, that tokens other than label words could also provide valuable information directly affecting the performance. Overall, our results demonstrate that model performance depends directly on the presence of these tokens and that their lexical meaning, their repetition throughout the ICL prompt, and their structural formatting of ICL demonstrations are likely to play a role in how effectively they allow an LLM to recover the critical information needed to perform a task.

## ETHICS STATEMENT

This work focuses on analyzing the working mechanisms of large language models and, as such, does not present any increased risks of harm beyond the existing norms of natural language processing or computational linguistics research. The associated risks include using a model trained on vast amounts of text, which may inadvertently contain biases. Another concern is the potential misuse of the model for generating misleading or harmful content. However, such a scenario is unlikely in our work, as we concentrate on classification tasks with fixed outputs.

## REPRODUCIBILITY STATEMENT

To ensure the reproducibility of our work, we have made several efforts that are documented throughout this paper. Our experiments utilize the open-source models described in Section 3.2. The prompts and templates used in our experiments are detailed in Section 3.1, Section 5.2 of the main text and in Appendix B, Appendix C, Appendix I, Appendix R, Appendix S. The stopword token list used in our experiments is shown in Appendix F. The complete code for our implementation, including all inference processes, is provided in the supplementary materials. We employed random seeds ranging from 1 to 15 to ensure consistent results across experiments, as specified in Section 3.2 and the supplementary code. All datasets used in our experiments are described comprehensively in Section 3.2, and the supplementary code includes all data processing steps and any preprocessing applied. We encourage other researchers to consult these references for replicating our findings.

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

Table 9: An example of the ICL template used in our experiments.

| Datasets | Notations | Examples |
|---|---|---|
| AGNews | $\mathbf{I}$
$\mathbf{T}^{\mathrm{in}}$
$\mathbf{T}^{\mathrm{out}}$ | Classify the news articles into the categories of World, Sports, Business, and Technology.\n\n
Article: $\{\mathbf{D}^{\mathrm{in}}\}$\n
Answer: $\{\mathbf{D}^{\mathrm{out}}\}$\n\n |
| SST2 | $\mathbf{I}$
$\mathbf{T}^{\mathrm{in}}$
$\mathbf{T}^{\mathrm{out}}$ | Classify the reviews into the categories of Positive and Negative.\n\n
Review: $\{\mathbf{D}^{\mathrm{in}}\}$\n
Sentiment: $\{\mathbf{D}^{\mathrm{out}}\}$\n\n |
| RTE | $\mathbf{I}$

$\mathbf{T}^{\mathrm{in}}$
$\mathbf{T}^{\mathrm{out}}$ | Classify the entailment of the hypothesis and the premise into the categories of True and False.\n\n
Hypothesis: $\{\mathbf{D}^{\mathrm{inA}}\}$\n Premise: $\{\mathbf{D}^{\mathrm{inB}}\}$\n
Answer: $\{\mathbf{D}^{\mathrm{out}}\}$\n\n |
| CB | $\mathbf{I}$

$\mathbf{T}^{\mathrm{in}}$
$\mathbf{T}^{\mathrm{out}}$ | Classify the entailment of the hypothesis and the premise into the categories of true, neither and false.\n\n
Hypothesis: $\{\mathbf{D}^{\mathrm{inA}}\}$\n Premise: $\{\mathbf{D}^{\mathrm{inB}}\}$\n
Answer: $\{\mathbf{D}^{\mathrm{out}}\}$\n\n |
| TREC | $\mathbf{I}$

$\mathbf{T}^{\mathrm{in}}$
$\mathbf{T}^{\mathrm{out}}$ | Classify the questions based on whether their answer type is a Number, Location, Person, Description, Entity, or Abbreviation.\n\n
Question: $\{\mathbf{D}^{\mathrm{in}}\}$\n
Answer Type: $\{\mathbf{D}^{\mathrm{out}}\}$\n\n |
| DBPedia | $\mathbf{I}$

$\mathbf{T}^{\mathrm{in}}$
$\mathbf{T}^{\mathrm{out}}$ | Classify the documents based on whether they are about a Company, School, Artist, Athlete, Politician, Transportation, Building, Nature, Village, Animal, Plant, Album, Film, or Book.\n\n
Article: $\{\mathbf{D}^{\mathrm{in}}\}$\n
Answer: $\{\mathbf{D}^{\mathrm{out}}\}$\n\n |

Table 10: The stopwords used in our experiments.

| Datasets | Stopwords |
|---|---|
| AGNews | "the", "into", "of", "and", ",", ".", "\n" |
| SST2 | "the", "into", "of", "and", ".", "\n" |
| RTE | "the", "of", "into", "and", "into", ".", "\n" |
| CB | "the", "of", "and", "into", ",", ".", "\n" |
| TREC | "the", "based", "on", "whether", "their", "is", "a", ",", "or", ".", "\n" |
| DBPedia | "the", "based", "on", "whether", "they", "are", "about", "a", ",", "or", ".", "\n" |

## A  LIMITATIONS

In this paper, the token categorization is performed manually, leaving room for further refinement, leaving the exploration of other specific content tokens as task-encoding tokens in certain contexts to future work. While the results provide robust support to our categorization, the identification process itself lacks precision. For instance, stopwords may only represent a subset of all in-context task-encoding tokens. The manual nature of our categorization limits our ability to comprehensively track these tokens. Moreover, our experiments are limited to classification, machine translation, and question answewring datasets, suggesting that our conclusions should be further validated for other tasks. Additionally, our focus on task-encoding tokens, whose representations could impact task performance, may overlook other tokens responsible for other possible functions. Another limitation of our study is that we focus exclusively on in-context learning scenarios, meaning that our findings may not be directly applicable to zero-shot learning scenarios.

## B  IN-CONTEXT LEARNING TEMPLATES

In this section, we present all the in-context learning templates used in this paper. For the RTE and CB datasets, there are two distinct inputs in the demonstrations (i.e., the hypothesis and the premise), which we denote as $\mathbf{D}^{\mathrm{inA}}$ and $\mathbf{D}^{\mathrm{inB}}$, respectively. The examples are provided in Table 9. All the notations are consistent with the notations in Table 1. All the next-line tokens are represented as "\n "

Table 11: The whole stopword list from NLTK. We add the punctuation tokens in this case.

| NLTK Stopwords List |
| --- |
| "i", "me", "my", "myself", "we", "our", "ours", "ourselves", "you", "your", "yours", "yourself", "yourselves", "he", "him", "his", "himself", "she", "her", "hers", "herself", "it", "its", "itself", "they", "them", "their", "theirs", "themselves", "what", "which", "who", "whom", "this", "that", "these", "those", "am", "is", "are", "was", "were", "be", "been", "being", "have", "has", "had", "having", "do", "does", "did", "doing", "a", "an", "the", "and", "but", "if", "or", "because", "as", "until", "while", "of", "at", "by", "for", "with", "about", "against", "between", "into", "through", "during", "before", "after", "above", "below", "to", "from", "up", "down", "in", "out", "on", "off", "over", "under", "again", "further", "then", "once", "here", "there", "when", "where", "why", "how", "all", "any", "both", "each", "few", "more", "most", "other", "some", "such", "no", "nor", "not", "only", "own", "same", "so", "than", "too", "very", "s", "t", "can", "will", "just", "don", "should", "now", "", "", "#", "$", "%", "^", "&", "*", "(", ")", "-", "_", "+", "=", "[", "]", "{", "}", "|", "\", ";", "'", "'", "<", ">", ",", ".", "?", "/", "\n" |

Table 12: Statistics for the original test set and the test set number we scaled up of each dataset we used.

| Dataset | AGNews | DBPedia | SST2 | TREC | RTE | CB |
| --- | --- | --- | --- | --- | --- | --- |
| Test set number of the dataset | 7,601 | 70,000 | 1,821 | 500 | 277 | 250 |
| Test set number we scaled up | 5,000 | 5,000 | 1,821 | - | - | - |

## C  REPRESENTATION-LEVEL ABLATION EXAMPLES

We provide a one-shot demonstration of all ablation cases for the representation-level ablation experiments, as shown in Table 14. In these demonstrations, representations of specific token types are masked in the attention mechanism, where <m> denotes that all representations of the token are removed from the attention scope from the test example.

## D  EXPERIMENT RESULTS WITH MORE LARGE LANGUAGE MODELS

Experimental results using the Llama 2 and Mistral models are shown in Table 15. The trends observed in these experiments are consistent with those involving the Llama models. These results further reinforce the findings of this paper, indicating that template tokens and stopword tokens are the most prone to serving as task-encoding tokens.

## E  EXPERIMENT RESULTS WITH INSTRUCTION-TUNED LARGE LANGUAGE MODELS

We present the representation-level ablation results of large language models after instruction tuning to confirm that our findings remain consistent. Specifically, we use Llama 2 7B Chat and Llama 2 13B Chat in our experiments. As shown in Table 16, the results align with the findings discussed in Section 4.3.1, with only one exception: the TREC dataset. In this dataset, the input data is structured in a question-answering format (e.g., Who/What/When did ...). We hypothesize that, during the supervised fine-tuning process, tokens associated with these question formats may also serve as task-encoding tokens, although they are currently categorized as content tokens in our experiments. Overall, these supplemental results prove further evidence to our findings in the main paper.

## F  STOPWORD TOKENS

For the results shown in the main paper, we used the stopword token list shown in Table 10. This list only includes the stopword tokens from the task instruction, aiming to minimize their presence. We

Table 13: Results of the representation-level ablation experiments with more text examples. The best results are in bold while the results showing the greatest decrease from ablation are underlined.

| OpenLlama 3B | | | | | | | |
|---|---|---|---|---|---|---|---|
| Setting | AGNews | SST2 | DBPedia | Setting | AGNews | SST2 | DBPedia |
| Zero-shot+CONT | 25.7 | 52.4 | 7.8 | Standard ICL−CONT | 57.6 | 86.4 | 62.3 |
| Zero-shot+STOP | 37.2 | 82.6 | 53.1 | Standard ICL−STOP | 62.0 | 91.0 | 63.2 |
| Zero-shot+TEMP | **56.2** | **86.3** | **62.5** | Standard ICL−TEMP | 41.5 | 87.2 | 57.3 |
| Llama 7B | | | | | | | |
| Setting | AGNews | SST2 | DBPedia | Setting | AGNews | SST2 | DBPedia |
| Zero-shot+CONT | 32.0 | 58.1 | 13.4 | Standard ICL−CONT | 77.0 | 91.9 | 67.4 |
| Zero-shot+STOP | 58.0 | 84.6 | 43.9 | Standard ICL−STOP | 79.5 | 94.2 | 69.0 |
| Zero-shot+TEMP | **71.4** | **90.7** | **67.4** | Standard ICL−TEMP | 64.3 | 84.9 | 58.6 |

Table 14: An example of the masked tokens from the attention of the test example in the representation-level ablation, where  represents the start of sentence token and <m> means that this token is masked. Tokens that are not masked are bold for clarity.

| Zero-shot+TEMP |
|---|
| ** Classify the news articles into the categories of World, Sports, Business, and Technology.** 

 **Article:** <m> <m> <m> <m> <m> <m> <m> <m> <m> <m> <m> <m> <m> <m> <m> <m> <m> <m> <m> <m> <m> <m> <m> <m> <m> <m> <m> <m> <m> <m> <m> <m> <m> <m> <m> <m> <m> <m> <m> <m> <m> <m> <m> <m> <m> <m> <m> <m> <m> <m> <m> <m> <m> <m> 
 **Answer: Technology** |

| Zero-shot+STOP |
|---|
| ** Classify the news articles into the categories of World, Sports, Business, and Technology.** 

 <m> <m> <m> <m> <m> <m> **the** <m> **.** <m> <m> <m> <m> <m> <m> <m> <m> <m> <m> <m> <m> <m> <m> <m> <m> **the** <m> <m> <m> <m> <m> <m> <m> <m> **,** <m> <m> <m> <m> <m> <m> **of** <m> <m> <m> <m> <m> <m> <m> <m> <m> **into** <m> <m> **the** <m> <m> **.** 
 <m> <m> **Technology** |

| Zero-shot+CONT |
|---|
| ** Classify the news articles into the categories of World, Sports, Business, and Technology.** 

 <m> <m> <m> **First class to** <m> **moon** <m> **London - British airline magnate Richard Branson announced a plan on Monday for** <m> **world's first commercial space flights** <m> **saying "thousands"** <m> **fee-paying astronauts could be sent** <m> **orbit in** <m> **near future** <m> <m> <m> <m> **Technology** <m> <m> |

| Standard ICL−TEMP |
|---|
| ** Classify the news articles into the categories of World, Sports, Business, and Technology.** 

 <m> <m> <m> **First class to the moon. London - British airline magnate Richard Branson announced a plan on Monday for the world's first commercial space flights, saying "thousands" of fee-paying astronauts could be sent into orbit in the near future.** <m> <m> <m> **Technology** <m> <m> |

| Standard ICL−STOP |
|---|
| ** Classify the news articles into the categories of World, Sports, Business, and Technology.** 

 **Article: First class to** <m> **moon** <m> **London - British airline magnate Richard Branson announced a plan on Monday for** <m> **world's first commercial space flights** <m> **saying "thousands"** <m> **fee-paying astronauts could be sent** <m> **orbit in** <m> **near future** <m> <m> **Answer: Technology** <m> <m> |

| Standard ICL−CONT |
|---|
| ** Classify the news articles into the categories of World, Sports, Business, and Technology.** 

 **Article:** <m> <m> <m> **the** <m> **.** <m> <m> <m> <m> <m> <m> <m> <m> <m> <m> <m> <m> <m> <m> <m> <m> <m> **the** <m> <m> <m> <m> <m> <m> <m> <m> **,** <m> <m> <m> <m> <m> <m> **of** <m> <m> <m> <m> <m> <m> <m> <m> <m> **into** <m> <m> **the** <m> <m> **.** 
 **Answer: Technology** |

made this choice under the assumption that task-affecting information should be stored densely in a few tokens. Hence, the number of tokens whose representations affect the final task performance significantly should be small.

Table 15: The accuracy results of the representation-level ablation study using Llama 2 and Mistral models where, for example, +TEMP refers to allowing attention only to template tokens and −TEMP refers to allowing attention only to content and stopword tokens. All values are presented as percentages. Results are acquired with 5 different random seeds. The best results are in bold and the results showing the greatest decrease during the ablation are underlined.

| Models | Setting | △Avg. | AGNews | SST2 | TREC | DBPedia | RTE | CB |
|--------|---------|-------|--------|------|------|---------|-----|-----|
| Llama 2 7B | Zero-shot | 36.0 | 50.2 | 50.4 | 57.2 | 6.4 | 51.6 | 0.0 |
| | + CONT | +1.9 | 0.9 | 61.0 | 50.6 | 12.9 | 48.7 | 53.2 |
| | + STOP | +23.4 | 49.0 | 78.1 | 54.4 | 61.6 | 65.3 | 47.9 |
| | + TEMP | **+31.3** | 81.1 | 82.6 | 55.2 | 65.5 | 63.9 | 55.4 |
| Llama 2 13B | Zero-shot | 53.3 | 56.2 | 90.8 | 49.0 | 7.6 | 70.0 | 46.4 |
| | + CONT | −14.1 | 0.5 | 56.0 | 61.4 | 0.0 | 62.6 | 54.6 |
| | + STOP | +9.6 | 47.2 | 76.8 | 65.2 | 65.3 | 66.5 | 56.8 |
| | + TEMP | **+18.1** | 78.2 | 93.7 | 62.4 | 70.4 | 71.9 | 52.1 |
| Mistral 7B | Zero-shot | 59.5 | 77.8 | 84.4 | 73.0 | 57.8 | 1.8 | 62.1 |
| | + CONT | −10.0 | 43.3 | 52.0 | 66.6 | 10.1 | 64.3 | 60.7 |
| | + STOP | +18.4 | 78.9 | 92.5 | 71.6 | 81.4 | 69.9 | 72.9 |
| | + TEMP | **+19.7** | 81.7 | 95.9 | 63.9 | 83.3 | 77.9 | 72.5 |
| Llama 2 7B | Standard ICL | 70.7 | 85.0 | 93.2 | 58.3 | 66.7 | 66.3 | 55.0 |
| | − CONT | −3.8 | 82.4 | 85.5 | 54.3 | 64.2 | 59.6 | 55.7 |
| | − STOP | −2.5 | 84.8 | 88.0 | 51.7 | 65.7 | 65.8 | 53.2 |
| | − TEMP | −32.8 | 0.9 | 61.0 | 50.6 | 12.9 | 48.5 | 53.6 |
| Llama 2 13B | Standard ICL | 73.6 | 82.8 | 94.9 | 62.8 | 74.6 | 71.2 | 55.4 |
| | − CONT | −1.3 | 79.0 | 94.1 | 62.7 | 72.4 | 72.1 | 53.6 |
| | − STOP | −2.9 | 80.1 | 89.4 | 61.5 | 74.1 | 69.6 | 49.3 |
| | − TEMP | −33.5 | 0.5 | 56.0 | 61.4 | 0.0 | 68.2 | 54.3 |
| Mistral 7B | Standard ICL | 80.2 | 82.2 | 97.0 | 67.4 | 82.4 | 73.6 | 78.6 |
| | − CONT | −0.6 | 81.8 | 96.2 | 64.4 | 83.4 | 78.9 | 72.9 |
| | − STOP | −0.8 | 81.3 | 97.0 | 66.5 | 80.5 | 75.5 | 75.7 |
| | − TEMP | −22.8 | 78.6 | 52.0 | 66.6 | 10.1 | 67.1 | 69.6 |

Table 16: The accuracy results of the representation-level ablation study using supervised finetuned (SFT) version of Llama 2 models where, for example, +TEMP refers to allowing attention only to template tokens and −TEMP refers to allowing attention only to content and stopword tokens. All values are presented as percentages. Results are acquired with 15 different random seeds. The best results are in bold and the results showing the greatest decrease during the ablation are underlined.

| Models | Setting | Avg. | AGNews | SST2 | TREC | DBPedia | RTE | CB |
|--------|---------|------|--------|------|------|---------|-----|-----|
| Llama 2 7B Chat | Zero-shot | - | - | - | - | - | - | - |
| | + CONT | 27.9 | 0.7 | 52.6 | 52.2 | 7.9 | 24.8 | 28.9 |
| | + STOP | 72.9 | 77.1 | 90.6 | 60.2 | 75.4 | 66.7 | 67.5 |
| | + TEMP | **75.6** | **80.1** | **92.6** | **62.6** | **76.6** | **69.5** | **72.1** |
| Llama 2 13B Chat | Zero-shot | - | - | - | - | - | - | - |
| | + CONT | 38.4 | 0.0 | 55.7 | **67.3** | 0.5 | 63.5 | 43.2 |
| | + STOP | 67.1 | 78.5 | 87.6 | 66.9 | 71.4 | 67.2 | 31.1 |
| | + TEMP | **72.3** | **82.0** | **93.7** | 65.6 | **72.3** | **72.3** | **47.9** |
| Llama 2 7B Chat | Standard ICL | - | - | - | - | - | - | - |
| | − CONT | 76.1 | 80.7 | 93.1 | 62.9 | 76.8 | 70.7 | 72.5 |
| | − STOP | 76.4 | 81.7 | 94.4 | 61.9 | 74.9 | 71.0 | 74.3 |
| | − TEMP | 31.4 | 0.7 | 52.6 | 52.2 | 7.9 | 24.1 | 51.1 |
| Llama 2 13B Chat | Standard ICL | - | - | - | - | - | - | - |
| | − CONT | 74.7 | 83.4 | 93.6 | 66.4 | 74.3 | 74.6 | 56.1 |
| | − STOP | 69.8 | 78.9 | 94.1 | 57.7 | 72.8 | 70.1 | 45.4 |
| | − TEMP | 38.7 | 0.0 | 55.7 | 67.3 | 0.5 | 65.1 | 43.6 |

Nevertheless, one might be curious about the results if we used a more complete stopword list. In this case, we utilize a more comprehensive stopword token list of NLTK[6] shown in Table 11 and conduct the representation-level ablation once more. The results are presented in Table 17. It can be observed

---

[6]https://gist.github.com/sebleier/554280

that all the conclusions from Section 4.3.1 are still well established. A few results are different from Table 2 and Table 3 because we masked the representations of the "" token in this set of experiments. We claim that this masking does not impact the main findings of these experiments.

Table 17: The accuracy results of the representation level ablation study where we use the more complete stopword token list of NLTK. All values are presented as percentages. The best results presented by the number of ablated token types are in bold.

| Models | Setting | AGNews | SST2 | TREC | DBPedia | RTE | CB |
|---|---|---|---|---|---|---|---|
| OpenLlama 3B | Zero-shot | 22.0 | 20.0 | 23.6 | 5.4 | 44.4 | 1.8 |
| | + CONT | 26.2 | 52.1 | 30.1 | 7.4 | 51.9 | 37.9 |
| | + STOP | 38.0 | 85.1 | **31.6** | 54.6 | **58.8** | **55.7** |
| | + TEMP | **56.5** | **86.7** | 27.1 | **62.2** | 56.4 | 52.3 |
| | Standard ICL | 63.7 | 91.2 | 21.9 | 61.9 | 57.4 | 52.0 |
| | - TEMP | 42.1 | 87.2 | 25.9 | 56.3 | **58.3** | **57.4** |
| | - CONT | 57.1 | 88.4 | **27.1** | **62.6** | 56.8 | 52.4 |
| | - STOP | **61.6** | **90.7** | 24.8 | 62.2 | 56.7 | 51.9 |
| Llama 7B | Zero-shot | 25.0 | 29.2 | 41.4 | 0.0 | 54.2 | 3.6 |
| | + CONT | 32.4 | 57.9 | 42.5 | 12.5 | 55.5 | 46.1 |
| | + STOP | 59.9 | 85.9 | 51.7 | 28.9 | 56.0 | 52.7 |
| | + TEMP | **70.8** | **90.2** | **58.4** | **66.2** | **66.3** | **73.5** |
| | Standard ICL | 82.4 | 94.3 | 63.5 | 68.7 | 68.6 | 71.3 |
| | - TEMP | 64.7 | 84.1 | 54.0 | 56.7 | 56.1 | 48.2 |
| | - CONT | 75.4 | 93.8 | 59.8 | 67.5 | 66.8 | **74.8** |
| | - STOP | **81.4** | **94.2** | **60.5** | **67.9** | **67.6** | 72.1 |
| Llama 13B | Zero-shot | 59.0 | 18.0 | 37.0 | 0.0 | 0.0 | 0.0 |
| | + CONT | 30.6 | 52.4 | 43.8 | 13.0 | 60.2 | 45.5 |
| | + STOP | 72.7 | 78.7 | 49.2 | 27.4 | 58.5 | 27.1 |
| | + TEMP | **78.5** | **92.3** | **59.0** | **74.2** | **67.4** | **52.3** |
| | Standard ICL | 81.6 | 94.3 | 60.0 | 76.1 | 70.6 | 39.9 |
| | - TEMP | 71.7 | 80.1 | 56.2 | 8.7 | 56.5 | 29.3 |
| | - CONT | **79.3** | 93.4 | **60.1** | **74.1** | 68.4 | **47.6** |
| | - STOP | 79.2 | **94.1** | 59.3 | 73.8 | **68.9** | 44.6 |
| Llama 33B | Zero-shot | 70.2 | 88.6 | 60.6 | 30.2 | 58.1 | 19.6 |
| | + CONT | 27.8 | 61.7 | 61.9 | 10.8 | 64.2 | 68.1 |
| | + STOP | 74.7 | 93.6 | **66.9** | 70.8 | 69.1 | 63.8 |
| | + TEMP | **80.6** | **95.2** | 63.1 | **71.9** | **78.7** | **84.0** |
| | Standard ICL | 85.0 | 96.5 | 68.1 | 78.4 | 78.5 | 83.3 |
| | - TEMP | 79.5 | 93.8 | 58.5 | 62.8 | 68.0 | 68.0 |
| | - CONT | 82.7 | 95.9 | **62.9** | **74.1** | **79.6** | **83.1** |
| | - STOP | **84.4** | **96.1** | 61.8 | 72.8 | 79.4 | 82.1 |

# G    ANALYSIS OF THE NEXT-LINE TOKENS

In this section, we analyze the next-line token, which is ablated whenever any type of the stopword tokens or template tokens are ablated in the representation-level ablation experiments. We analyze this token by not ablating it when the these types of tokens are ablated. Results presented in Table 18 demonstrate that the next-line token is an important task-encoding token, due to the fact that they improved the performance by a large margin compared to the results in Table 3.

# H    COMPARISON EXPERIMENTS WITH MORE TEXT EXAMPLES

In the ideal scenario, our experiments would have been conducted on the full test set. However, in practice, this is infeasible for any of the models studied in our paper due to computational resource constraints. For instance, it took 42 hours for the OpenLlama 3B model to run one round of the representation ablation experiment on the whole test set of DBPedia (i.e., one cell in Table 2 and 3 for the DBPedia column). To verify our number of test examples decision, we provide additional results where we scale up the number of test examples and observe no difference with our original experimental setup. Thus, we believe that limiting our test set sample size to 500 is a reasonable setup. We provide the test set statistics and the experiment results in Table 12 and Table 13.

Table 18: The accuracy results of the representation-level ablation study where, for example, $-$ TEMP refers to allowing attention only to content and stopword tokens. The next-line tokens are always ablated in this set of experiments. All values are presented as percentages. Except where noted with $*$, all test statistics reported correspond to p-values $< 0.05$. The results showing the greatest decrease from ablation are underlined.

| Models | Setting | AGNews | SST2 | TREC | DBPedia | RTE | CB | $\triangle$Avg. |
|---|---|---|---|---|---|---|---|---|
| OpenLlama 3B | Standard ICL | 63.7 | 91.2 | 21.9 | 61.9 | 57.4 | 52.0 | 58.0 |
| | $-$ STOP | 62.3 | 91.0 | 24.8* | 62.9 | 57.1 | 51.1* | +0.2 |
| | $-$ TEMP | 41.9 | 87.2 | 26.0 | 56.3 | 58.5 | 57.4 | $-5.4$ |
| Llama 7B | Standard ICL | 82.4 | 94.3 | 63.5 | 68.7 | 68.6 | 71.3 | 74.8 |
| | $-$ STOP | 80.4 | 94.6 | 61.1 | 68.0 | 67.2 | 72.0 | $-0.9$ |
| | $-$ TEMP | 64.5 | 84.1 | 54.0 | 58.0 | 56.8 | 54.3 | $-12.8$ |
| Llama 13B | Standard ICL | 81.6 | 94.3 | 60.0 | 76.1 | 70.6 | 39.9 | 70.4 |
| | $-$ STOP | 81.2 | 94.1 | 59.3 | 76.9 | 69.2 | 40.6 | $-0.2$ |
| | $-$ TEMP | 74.1 | 80.0 | 46.5 | 30.6 | 58.3 | 25.4 | $-17.9$ |
| Llama 33B | Standard ICL | 85.0 | 96.5 | 68.1 | 78.4 | 78.5 | 83.3 | 81.6 |
| | $-$ STOP | 84.3 | 95.6 | 65.7 | 77.6 | 78.6 | 81.8 | $-1.0$ |
| | $-$ TEMP | 76.6 | 93.9* | 61.2 | 72.7 | 70.3 | 59.6 | $-9.2$ |
| Llama 2 7B | Standard ICL | 70.7 | 85.0 | 93.2 | 58.3 | 66.7 | 66.3 | 55.0 |
| | $-$ STOP | 85.5 | 92.7 | 56.4 | 66.6 | 63.6 | 57.1 | $-0.4$ |
| | $-$ TEMP | 69.8 | 82.8 | 56.3 | 58.8 | 67.5 | 42.9 | $-7.7$ |
| Llama 2 13B | Standard ICL | 73.6 | 82.8 | 94.9 | 62.8 | 74.6 | 71.2 | 55.4 |
| | $-$ STOP | 81.2 | 94.5 | 61.2 | 73.7 | 72.0 | 53.2 | $-1.0$ |
| | $-$ TEMP | 71.1 | 95.6 | 61.0 | 72.4 | 72.9 | 54.3 | $-2.4$ |
| Mistral 7B | Standard ICL | 80.2 | 82.2 | 97.0 | 67.4 | 82.4 | 73.6 | 78.6 |
| | $-$ STOP | 81.2 | 97.3 | 65.5 | 82.0 | 77.6 | 73.9 | $-0.6$ |
| | $-$ TEMP | 78.6 | 89.7 | 67.6 | 79.4 | 70.8 | 72.5 | $-3.8$ |

Table 19: The different instruction prompts used in our experiments. "Ins." represents "Instruction".

| Datasets | Stopwords |
|---|---|
| AGNews Ins. 1 | Classify the text into World, Sports, Business, and Technology. |
| AGNews Ins. 2 | Classify the articles based on whether they are in the categories of World, Sports, Business, and Technology. |
| AGNews Ins. 3 | Classify the news to World, Sports, Business, and Technology. |
| DBPedia Ins. 1 | Classify the text into Company, School, Artist, Athlete, Politician, Transportation, Building, Nature, Village, Animal, Plant, Album, Film, and Book. |
| DBPedia Ins. 2 | Classify the documents into the categories of Company, School, Artist, Athlete, Politician, Transportation, Building, Nature, Village, Animal, Plant, Album, Film, and Book. |
| DBPedia Ins. 3 | Classify the articles based on whether they are in the categories of Company, School, Artist, Athlete, Politician, Transportation, Building, Nature, Village, Animal, Plant, Album, Film, and Book. |

For TREC, RTE, and CB, using 500 test examples won't affect the final results at all since their test set size is smaller than 500. We provide the results of experiments using all test examples in SST2, and 5000 test examples in AGNews and DBPedia here to prove our point that limiting our test set sample size to 500 is a reasonable compromise. Shown in Table 13, compared to the results we show in Table 2 and Table 3, the numbers are changed less than 1% for all the results.

# I    RESULTS USING DIFFERENT INSTRUCTION PROMPTS

We conducted experiments on AGNews and DBPedia with 3 other different instructions to show that the and show the results in Table 20 and Table 21. Based on these additional results, our conclusions remain the same, which shows that our findings are not sensitive to variations of the instruction prompt. The different instruction prompts **I** we used are shown in Table 19.

Table 20: Results of the representation-level ablation experiments with different instruction prompts for AGNews dataset. "Ins." represents "Instruction". The best results are in bold while the results showing the greatest decrease from ablation are underlined.

| OpenLlama 3B | | | | | | | |
|---|---|---|---|---|---|---|---|
| Setting | Ins. 1 | Ins. 2 | Ins. 3 | Setting | Ins. 1 | Ins. 2 | Ins. 3 |
| Zero-shot+CONT | 26.5 | 26.9 | 22.1 | Standard ICL−CONT | 53.1 | 55.6 | 67.9 |
| Zero-shot+STOP | 40.6 | 38.8 | 49.1 | Standard ICL−STOP | 57.5 | 59.8 | 72.0 |
| Zero-shot+TEMP | **51.1** | **53.7** | **67.6** | Standard ICL−TEMP | 43.3 | 42.6 | 53.9 |
| Llama 7B | | | | | | | |
| Setting | Ins. 1 | Ins. 2 | Ins. 3 | Setting | Ins. 1 | Ins. 2 | Ins. 3 |
| Zero-shot+CONT | 31.1 | 30.2 | 35.2 | Standard ICL−CONT | 70.6 | 78.2 | 75.2 |
| Zero-shot+STOP | 51.4 | 63.5 | 61.8 | Standard ICL−STOP | 73.9 | 80.1 | 79.4 |
| Zero-shot+TEMP | **62.5** | **73.2** | **71.1** | Standard ICL−TEMP | 59.9 | 69.1 | 74.0 |

Table 21: Results of the representation-level ablation experiments with different instruction prompts for DBPedia dataset. "Ins." represents "Instruction". The best results are in bold while the results showing the greatest decrease from ablation are underlined.

| OpenLlama 3B | | | | | | | |
|---|---|---|---|---|---|---|---|
| Setting | Ins. 1 | Ins. 2 | Ins. 3 | Setting | Ins. 1 | Ins. 2 | Ins. 3 |
| Zero-shot+CONT | 6.7 | 6.3 | 7.2 | Standard ICL−CONT | 56.1 | 60.4 | 58.1 |
| Zero-shot+STOP | 43.1 | 48.3 | 40.2 | Standard ICL−STOP | 58.1 | 61.4 | 59.6 |
| Zero-shot+TEMP | **55.7** | **59.9** | **57.7** | Standard ICL−TEMP | 48.6 | 54.0 | 47.8 |
| Llama 7B | | | | | | | |
| Setting | Ins. 1 | Ins. 2 | Ins. 3 | Setting | Ins. 1 | Ins. 2 | Ins. 3 |
| Zero-shot+CONT | 15.0 | 15.9 | 6.8 | Standard ICL−CONT | 64.9 | 66.1 | 68.7 |
| Zero-shot+STOP | 49.7 | 48.1 | 48.6 | Standard ICL−STOP | 66.8 | 67.6 | 69.6 |
| Zero-shot+TEMP | **66.1** | **66.5** | **69.0** | Standard ICL−TEMP | 58.9 | 59.2 | 61.7 |

## J    REPRESENTATION-LEVEL ABLATION ON MACHINE TRANSLATION TASKS

Besides the classification tasks, we also show results in the machine translation tasks to show that our findings could also be extended in text generation tasks. We used the Flores MT dataset (Costa-jussà et al., 2022) to conduct this set of 4-shot machine translation experiments. The results are reported with the BLEU metric (Papineni et al., 2002). We investigated three different language directions: English-to-French, English-to-Danish, and English-to-German. We used 10 random seeds for En-Fr and En-De and 15 random seeds for En-Da to randomly choose the demonstrations. 100 test examples are sampled in this set of the experiments as a computational compromise. Similar to the classification tasks, we keep the answer (i.e., target language) unablated for all the settings and ablate different kinds of tokens. Results in Table 22 show the consistent finding to those in Section 4.3.1.

## K    REPRESENTATION-LEVEL ABLATION ON QUESTION ANSWERING TASKS

We show the representation-level experimental results of the question answering (QA) tasks in this section. We used Commonsense QA (Talmor et al., 2019) dataset to test if the template and stopword tokens would directly affect the downstream task performance. We applied the settings of 4 in-context examples and 15 random seeds in this set of experiments. We frame the task as directly answering the questions instead of choosing one answer from the choices because the token types in this scenario are easier to be categorized.

Results shown in Table 23 demonstrate that our main findings, that template and stopword tokens are more likely to serve as task-encoding tokens, still hold in the QA tasks.

Table 22: Results of the representation-level ablation for machine translation tasks. The best results are in bold while the results showing the greatest decrease from ablation are underlined.

| OpenLlama 3B | | | | | | | |
|---|---|---|---|---|---|---|---|
| Settings | En-Fr | En-De | En-Da | Settings | En-Fr | En-De | En-Da |
| Zero-shot+CONT | 0.13 | 0.38 | 0.28 | Standard ICL−CONT | 26.07 | 12.53 | 17.43 |
| Zero-shot+STOP | 16.68 | 9.17 | 13.09 | Standard ICL−STOP | 26.19 | 12.52 | 17.29 |
| Zero-shot+TEMP | **26.06** | **12.92** | **17.17** | Standard ICL−TEMP | 17.38 | 8.88 | 12.9 |
| Llama 7B | | | | | | | |
| Settings | En-Fr | En-De | En-Da | Settings | En-Fr | En-De | En-Da |
| Zero-shot+CONT | 11.76 | 13.83 | 10.18 | Standard ICL−CONT | 35.39 | 24.23 | 30.08 |
| Zero-shot+STOP | 30.23 | 21.76 | 23.34 | Standard ICL−STOP | 35.36 | 24.33 | 29.99 |
| Zero-shot+TEMP | **35.47** | **24.34** | **30.12** | Standard ICL−TEMP | 31.09 | 21.98 | 24.88 |

Table 23: Results of the representation-level ablation for question answering tasks. 15 random seeds are used to acquire all the experimental results. The best results are in bold while the results showing the greatest decrease from ablation are underlined.

| Setting | OpenLlama 3B | Llama 7B | Llama 13B | Llama 33B | Llama 2 7B | Llama 2 13B | Mistral 7B |
|---|---|---|---|---|---|---|---|
| Zero-shot+CONT | 7.42 | 16.62 | 14.49 | 19.47 | 17.51 | 17.78 | 16.64 |
| Zero-shot+STOP | 11.71 | 21.96 | 18.98 | 23.38 | 22.13 | 22.31 | 19.16 |
| Zero-shot+TEMP | **13.24** | 24.38 | **25.73** | **27.20** | **25.42** | **25.11** | **25.56** |
| Standard ICL−CONT | 14.40 | 24.07 | 26.22 | 27.73 | 25.89 | 24.71 | 25.47 |
| Standard ICL−STOP | 11.96 | 21.84 | 21.44 | 26.93 | 24.62 | 23.09 | 23.69 |
| Standard ICL−TEMP | 6.89 | 16.29 | 15.31 | 19.78 | 18.51 | 16.64 | 15.51 |

## L    REPRESENTATION-LEVEL ABLATION BASED ON THE TOKEN COUNT

One possible explanation for the performance variation when different types of tokens are ablated at the representation level is the simple fact that the number of tokens being ablated may vary. Intuitively, template and stopword tokens are far fewer in number compared to content tokens. In this section, we show the statistics of the number count of each type of tokens and include a supplementary experiment that only let the LLM attend to certain number of token representations of each type of the tokens.

We first present the average token count for each type of token across the datasets. Token counts may vary depending on the tokenizer used by the large language models, and all statistics are shown in Table 24. The results indicate that the number of template and stopword tokens is much smaller than the number of content tokens, suggesting that performance variation during ablation is not solely due to differences in token type counts.

Table 24: The token count statistics of different types of tokens. Avg. stands for the average token count for each type of tokens.

| Tokenizer | Setting | Avg. | AGNews | DBPedia | SST2 | TREC | CB | RTE |
|---|---|---|---|---|---|---|---|---|
| OpenLlama 3B | CONT | 204.1 | 207.5 | 278.8 | 116.3 | 48.7 | 295.5 | 278.0 |
| | STOP | 43.0 | 43.2 | 45.1 | 27.7 | 21.8 | 66.7 | 53.7 |
| | TEMP | 56.5 | 43.3 | 49.9 | 42.4 | 48.8 | 78.5 | 76.3 |
| Llama & Llama 2 | CONT | 220.6 | 238.5 | 288.8 | 127.8 | 51.3 | 312.0 | 305.1 |
| | STOP | 45.0 | 43.1 | 46.1 | 29.9 | 21.7 | 71.1 | 57.9 |
| | TEMP | 52.7 | 41.3 | 50.5 | 48.7 | 36.7 | 70.9 | 68.1 |
| Mistral | CONT | 213.6 | 225.0 | 284.7 | 123.7 | 50.1 | 304.2 | 293.7 |
| | STOP | 44.7 | 43.2 | 45.0 | 29.9 | 21.7 | 70.7 | 57.8 |
| | TEMP | 54.6 | 45.3 | 53.5 | 40.5 | 40.7 | 75.0 | 72.5 |

We then conduct an additional ablation experiment in which the model attends to representations from a specific number of tokens of a given type. We also include a baseline where a random subset of token representations from all prompt tokens is unmasked to the test examples. In this set of experiments, the label tokens are always included and are not counted as part of the token numbers.

Results in Table 25 demonstrate that when the model is exposed to an equal number of each type of token representation, the performance consistently improves with template and stopword tokens, outperforming both content tokens and the random baseline. In contrast, models attending to the same number of content tokens consistently underperform relative to the random baseline. Additionally, all results improve when more tokens are included in the attention of test examples. This experiment further supports our claim that template and stopword tokens are more likely to serve as task-encoding tokens.

## M    RESULTS OF THE TOKEN-LEVEL ABLATION

Detailed results of the token-level ablation are shown in Table 26. We omited the $-$TEMP case from here since it constantly yields an accuracy of 0% when both the template token in the demonstrations and the test examples are ablated. Since the setting for the template tokens are not aligned with the ones for the stopword and content tokens, we included another set of experiments where only the the template tokens in the demonstrations are ablated at the token level in Appendix N. We want to emphasize that this experimental design choice does not affect the main findings in Section 4.3.2, where information is being propagated from the representations of content tokens to the representations of the task-encoding tokens and this incorporation of the information makes them better at encoding tasks, partially explaining the working mechanism of in-context learning.

## N    TOKEN-LEVEL ABLATION FOR TEMPLATE TOKENS

To maintain consistency of the templates across both demonstrations and test examples, we choose to ablate the template tokens at the token level in both in Section 4.3.2. This experimental design differs from the other two token-level ablations. This inconsistency does not impact the main findings in Section 4.3.2, which show that information is propagated from the representations of content tokens to the representations of task-encoding tokens and this information aggregation enhances the ability of task-encoding tokens to improve the final task performance, partially explaining the mechanism of in-context learning. For completeness, we provide a supplemental experiment in this section where only the template tokens in the demonstrations are ablated.

Results in Table 27 demonstrate that, although not all values reduce to 0%, large language models perform significantly worse than in the standard in-context learning case and the other two ablation scenarios after the removal of template tokens from the demonstrations except for a few rare cases. This further supports the finding that template tokens are likely important as task-encoding tokens.

## O    POSSIBLE APPLICATIONS

In this section, we discuss several potential applications that could benefit from the findings in our work. These include **long sequence processing**, where our insights can help models handle longer contexts more efficiently; **in-context learning with more demonstrations**, enabling the inclusion of additional examples without compromising performance; **better ICL prompt designing and engineering**, improving the creation of more effective prompts; and **improving model robustness**, ensuring consistent performance despite prompt variations. Each of these areas can be enhanced by understanding the role of task-encoding tokens in large language models.

**Long sequence processing**    As discussed in our paper, we hypothesize that stopword tokens tend to function as task-encoding tokens by encapsulating the semantics of preceding tokens. This finding suggests an opportunity to improve the efficiency of modeling longer sequences by selectively deleting or compressing certain hidden states during the encoding and generation stages of large language models (LLMs). Specifically, by retaining only the essential task-encoding representations while reducing unnecessary content from less informative tokens, models could manage longer inputs

Table 25: The accuracy results of the representation level ablation study where we only include fixed number of certain type of tokens. All values are presented as percentages. The best results presented by the number of ablated token types are in bold. Avg. stands for the average performance. ALL represents all types of tokens

| Models | Setting | AGNews | DBPedia | SST2 | TREC | CB | RTE | Avg. |
|--------|---------|--------|---------|------|------|-----|-----|------|
| **10 Random Tokens** | | | | | | | | |
| Llama 2 7B | From ALL | 14.5 | 16.9 | 71.7 | 47.3 | 60.2 | 57.5 | 44.7 |
| | From CONT | 3.0 | 9.2 | 60.9 | 61.9 | 65.2 | 60.4 | 43.4 |
| | From STOP | 15.2 | 35.6 | 75.2 | 68.5 | 62.0 | 64.1 | 53.4 |
| | From TEMP | 53.0 | 50.1 | 59.1 | 56.2 | 64.4 | 60.2 | **57.2** |
| Mistral 7B | From ALL | 67.8 | 67.2 | 75.0 | 70.9 | 63.8 | 60.7 | 67.6 |
| | From CONT | 36.6 | 64.0 | 63.1 | 67.5 | 59.3 | 57.3 | 58.0 |
| | From STOP | 73.5 | 68.6 | 86.3 | 72.8 | 61.8 | 61.6 | 70.8 |
| | From TEMP | 78.7 | 78.7 | 92.1 | 68.9 | 71.2 | 68.4 | **76.3** |
| **20 Random Tokens** | | | | | | | | |
| Llama 2 7B | From ALL | 12.6 | 24.0 | 77.2 | 54.4 | 60.6 | 58.9 | 47.9 |
| | From CONT | 1.3 | 8.6 | 65.3 | 63.5 | 61.7 | 62.2 | 43.8 |
| | From STOP | 28.3 | 45.5 | 84.5 | 66.8 | 58.0 | 67.1 | 58.4 |
| | From TEMP | 75.0 | 63.7 | 60.3 | 59.0 | 65.4 | 60.7 | **64.0** |
| Mistral 7B | From ALL | 78.4 | 67.8 | 74.0 | 70.1 | 65.4 | 62.0 | 69.6 |
| | From CONT | 69.1 | 63.3 | 59.3 | 68.2 | 60.5 | 56.8 | 62.9 |
| | From STOP | 78.4 | 74.3 | 89.9 | 74.2 | 68.3 | 69.1 | 75.7 |
| | From TEMP | 81.9 | 78.7 | 93.0 | 68.9 | 72.4 | 71.4 | **77.7** |
| **30 Random Tokens** | | | | | | | | |
| Llama 2 7B | From ALL | 27.4 | 27.7 | 77.9 | 54.9 | 63.9 | 62.2 | 52.3 |
| | From CONT | 1.3 | 7.6 | 77.9 | 66.3 | 64.2 | 63.5 | 46.8 |
| | From STOP | 44.8 | 59.8 | 92.8 | 68.3 | 53.2 | 69.3 | 64.7 |
| | From TEMP | 76.9 | 66.1 | 61.0 | 59.6 | 67.9 | 59.8 | **65.2** |
| Mistral 7B | From ALL | 79.9 | 74.6 | 83.0 | 67.8 | 69.0 | 64.6 | 73.2 |
| | From CONT | 71.0 | 62.6 | 58.2 | 67.2 | 59.2 | 59.4 | 62.9 |
| | From STOP | 79.8 | 76.9 | 91.0 | 74.2 | 71.1 | 68.8 | 77.0 |
| | From TEMP | 84.0 | 79.3 | 93.8 | 68.9 | 75.6 | 75.4 | **79.5** |
| **40 Random Tokens** | | | | | | | | |
| Llama 2 7B | From ALL | 45.2 | 27.2 | 77.9 | 54.9 | 58.9 | 62.3 | 54.4 |
| | From CONT | 0.9 | 9.1 | 89.5 | 66.1 | 62.3 | 62.4 | 48.4 |
| | From STOP | 49.2 | 63.7 | 94.0 | 68.7 | 53.7 | 70.8 | **66.7** |
| | From TEMP | 77.9 | 66.1 | 62.9 | 61.5 | 68.1 | 60.1 | 66.1 |
| Mistral 7B | From ALL | 78.8 | 73.4 | 86.9 | 69.2 | 69.3 | 67.4 | 74.2 |
| | From CONT | 74.1 | 59.4 | 55.8 | 68.0 | 61.0 | 59.9 | 63.0 |
| | From STOP | 80.1 | 76.4 | 91.0 | 74.2 | 72.0 | 69.6 | 77.2 |
| | From TEMP | 85.0 | 80.7 | 93.8 | 69.0 | 76.4 | 76.2 | **80.2** |

Table 26: Results of the token-level ablation where, for example, −STOP refers to the ablation where stopword tokens are dropped from the ICL prompt. Models without template tokens consistently yielded an accuracy of 0% and are thus omitted from this table.

| Models | Settings | AGNews | SST2 | TREC | DBPedia | RTE | CB | Avg. |
|---|---|---|---|---|---|---|---|---|
| OpenLlama 3B | Standard ICL | 63.7 | 91.2 | 21.9 | 61.9 | 57.4 | 52.0 | 58.0 |
| | − CONT | 31.5 | 63.0 | 40.6 | 25.4 | 56.1 | 48.9 | 44.3 |
| | − STOP | 64.4 | 91.5 | 20.9 | 62.3 | 57.8 | 52.6 | 58.3 |
| Llama 7B | Standard ICL | 82.4 | 94.3 | 63.5 | 68.7 | 68.6 | 71.3 | 74.8 |
| | − CONT | 55.2 | 67.2 | 42.6 | 50.8 | 57.4 | 56.3 | 54.9 |
| | − STOP | 82.3 | 93.8 | 64.1 | 69.7 | 66.5 | 70.0 | 74.4 |
| Llama 13B | Standard ICL | 81.6 | 94.3 | 60.0 | 76.1 | 70.6 | 39.9 | 70.4 |
| | − CONT | 78.8 | 81.7 | 45.3 | 75.1 | 55.1 | 54.5 | 65.1 |
| | − STOP | 82.5 | 92.5 | 61.5 | 76.5 | 69.6 | 40.5 | 70.5 |
| Llama 33B | Standard ICL | 85.0 | 96.5 | 68.1 | 78.4 | 78.5 | 83.3 | 81.6 |
| | − CONT | 74.0 | 89.6 | 67.0 | 73.0 | 69.8 | 49.0 | 70.4 |
| | − STOP | 85.3 | 96.4 | 66.9 | 77.9 | 77.7 | 81.3 | 80.9 |

Table 27: Results of the token-level ablation where −TEMP refers to the ablation where template tokens are dropped from the ICL demonstration prompt.

| Models | Settings | Avg. | AGNews | SST2 | TREC | DBPedia | RTE | CB |
|---|---|---|---|---|---|---|---|---|
| OpenLlama 3B | Standard ICL | 58.0 | 63.7 | 91.2 | 21.9 | 61.9 | 57.4 | 52.0 |
| | − TEMP | 17.4 | 0.0 | 24.2 | 40.5 | 0.4 | 35.6 | 3.6 |
| Llama 7B | Standard ICL | 74.8 | 82.4 | 94.3 | 63.5 | 68.7 | 68.6 | 71.3 |
| | − TEMP | 29.0 | 41.0 | 11.4 | 39.9 | 0.9 | 49.7 | 31.3 |
| Llama 13B | Standard ICL | 70.4 | 81.6 | 94.3 | 60.0 | 76.1 | 70.6 | 39.9 |
| | − TEMP | 36.9 | 82.1 | 17.5 | 51.4 | 8.6 | 39.5 | 22.0 |
| Llama 33B | Standard ICL | 81.6 | 85.0 | 96.5 | 68.1 | 78.4 | 78.5 | 83.3 |
| | − TEMP | 63.0 | 73.5 | 82.8 | 58.8 | 39.9 | 66.4 | 56.7 |

and outputs without compromising performance. This approach not only conserves computational resources but also addresses token length limitations in LLMs, allowing for extended sequence processing and potentially more nuanced learning from longer contexts.

This area is indeed attracting increased research attention, and our findings could contribute valuable insights into ongoing work on efficient sequence modeling and memory management in LLMs (Liu et al., 2023b; Zhang et al., 2023; Bai et al., 2024). By identifying which tokens retain critical task-related information, our work aligns with and can inform methods focused on compressing intermediate states and improving long-context processing for applications ranging from summarization and document understanding to interactive dialogue systems.

**In-context learning with more demonstrations**  Given our findings, there is a promising avenue for improving in-context learning (ICL) performance by including a greater number of examples in ICL prompts (Li et al., 2023a; Hao et al., 2022; Bertsch et al., 2024). Our results suggest that only a subset of token representations, specifically task-encoding tokens, play a critical role in determining ICL performance, while the representations of other tokens are less impactful. This observation opens up the possibility of selectively compressing or omitting unimportant token representations after the initial encoding of a demonstration. By doing so, it becomes feasible to maximize the use of the model's fixed-length capacity, potentially enabling the inclusion of a higher number of examples within the same prompt length constraints. This approach may enhance the effectiveness of ICL in tasks where the availability of diverse examples contributes to improved model accuracy and stability.

**Better ICL prompt designing and engineering**  Our investigation into which components of ICL prompts are most critical for task performance is worthwhile and useful for directing where to put effort into tuning or improving prompts. Furthermore, the exploration on the characteristics of task-encoding tokens are useful for future design choices in ICL prompting, and help the field understand why some prompts work better than others for ICL. For instance, knowing that template and stopword

Table 28: Ablation for different template token representations with the answer label token representations, presented as percentages. The results showing the greatest impact from ablation are underlined.

| Models | Settings | AGNews | | SST2 | | TREC | | DBPedia | | RTE | | CB | |
|---|---|---|---|---|---|---|---|---|---|---|---|---|---|
| | | with ":" | w/o ":" | with ":" | w/o ":" | with ":" | w/o ":" | with ":" | w/o ":" | with ":" | w/o ":" | with ":" | w/o ":" |
| OpenLlama 3B | TEMP with $\mathbf{D}^{out}$ | 56.5 | 47.4 | 86.7 | 83.7 | 27.1 | 26.5 | 62.2 | 59.8 | 56.4 | 56.0 | 52.3 | 56.1 |
| | $-\mathbf{T}^{in}$ | 50.3 | 47.1 | 85.7 | 84.4 | 28.9 | 24.4 | 57.7 | 57.7 | 56.5 | 56.1 | 53.2 | 55.2 |
| | $-\mathbf{T}^{out}$ | 34.6 | 32.7 | 86.9 | 82.3 | 28.2 | 31.2 | 55.5 | 54.1 | 58.3 | 59.2 | 55.4 | 58.3 |
| Llama 7B | TEMP with $\mathbf{D}^{out}$ | 70.8 | 57.3 | 90.2 | 87.1 | 58.4 | 46.7 | 66.2 | 63.8 | 66.3 | 59.5 | 73.5 | 69.6 |
| | $-\mathbf{T}^{in}$ | 62.7 | 55.1 | 91.6 | 87.1 | 52.8 | 43.3 | 61.6 | 61.8 | 58.9 | 56.5 | 59.2 | 55.7 |
| | $-\mathbf{T}^{out}$ | 50.8 | 48.6 | 84.9 | 82.8 | 46.0 | 50.2 | 57.9 | 55.2 | 56.7 | 56.7 | 66.2 | 64.5 |
| Llama 13B | TEMP with $\mathbf{D}^{out}$ | 80.0 | 76.2 | 92.3 | 89.1 | 58.6 | 54.0 | 76.9 | 71.4 | 68.5 | 59.8 | 47.7 | 35.0 |
| | $-\mathbf{T}^{in}$ | 79.9 | 76.3 | 91.5 | 88.9 | 55.1 | 47.8 | 75.8 | 70.7 | 65.5 | 59.0 | 35.7 | 24.5 |
| | $-\mathbf{T}^{out}$ | 72.0 | 72.1 | 81.1 | 75.9 | 47.1 | 48.3 | 60.3 | 35.5 | 61.2 | 58.4 | 36.2 | 36.0 |
| Llama 33B | TEMP with $\mathbf{D}^{out}$ | 80.5 | 75.0 | 95.2 | 93.3 | 65.2 | 66.7 | 75.2 | 73.5 | 79.0 | 77.1 | 80.0 | 70.7 |
| | $-\mathbf{T}^{in}$ | 78.7 | 71.5 | 95.2 | 92.8 | 68.1 | 67.7 | 75.1 | 73.8 | 77.4 | 75.4 | 73.3 | 62.3 |
| | $-\mathbf{T}^{out}$ | 69.2 | 69.5 | 93.9 | 92.9 | 62.1 | 66.2 | 71.3 | 70.1 | 72.8 | 70.0 | 67.4 | 63.5 |

Table 29: Ablation for different template token representations without the answer label token representations. All values are presented as percentages. The results showing the greatest decrease during the ablation are underlined.

| Models | Settings | AGNews | | SST2 | | TREC | | DBPedia | | RTE | | CB | |
|---|---|---|---|---|---|---|---|---|---|---|---|---|---|
| | | with ":" | w/o ":" | with ":" | w/o ":" | with ":" | w/o ":" | with ":" | w/o ":" | with ":" | w/o ":" | with ":" | w/o ":" |
| OpenLlama 3B | TEMP w/o $\mathbf{D}^{out}$ | 41.5 | 54.6 | 14.3 | 73.2 | 36.5 | 42.0 | 29.4 | 21.7 | 24.7 | 45.7 | 0.7 | 3.5 |
| | $-\mathbf{T}^{in}$ | 42.2 | 52.2 | 18.5 | 79.9 | 39.7 | 42.2 | 22.6 | 22.5 | 49.8 | 57.1 | 3.1 | 6.5 |
| | $-\mathbf{T}^{out}$ | 36.3 | 35.5 | 83.0 | 83.4 | 43.2 | 41.9 | 16.3 | 18.7 | 54.4 | 56.8 | 1.2 | 4.2 |
| Llama 7B | TEMP w/o $\mathbf{D}^{out}$ | 50.4 | 56.6 | 68.2 | 56.1 | 55.3 | 48.5 | 0.2 | 1.3 | 40.7 | 42.5 | 28.5 | 18.8 |
| | $-\mathbf{T}^{in}$ | 46.1 | 50.6 | 61.9 | 55.1 | 43.5 | 44.7 | 0.0 | 0.2 | 43.7 | 49.9 | 27.1 | 26.5 |
| | $-\mathbf{T}^{out}$ | 21.4 | 12.7 | 86.2 | 66.5 | 54.4 | 55.6 | 0.0 | 0.0 | 56.0 | 55.6 | 39.4 | 35.1 |
| Llama 13B | TEMP w/o $\mathbf{D}^{out}$ | 66.9 | 77.0 | 65.6 | 87.9 | 51.8 | 53.1 | 0.1 | 0.1 | 57.5 | 53.7 | 16.7 | 21.9 |
| | $-\mathbf{T}^{in}$ | 72.9 | 76.6 | 83.0 | 89.5 | 45.5 | 48.4 | 0.0 | 0.0 | 53.6 | 52.8 | 16.0 | 20.1 |
| | $-\mathbf{T}^{out}$ | 79.2 | 77.7 | 77.5 | 47.5 | 56.8 | 43.2 | 0.0 | 0.0 | 54.8 | 53.7 | 4.3 | 2.4 |
| Llama 33B | TEMP w/o $\mathbf{D}^{out}$ | 77.3 | 78.2 | 17.3 | 88.9 | 65.4 | 69.3 | 31.0 | 41.7 | 71.8 | 65.8 | 23.8 | 23.0 |
| | $-\mathbf{T}^{in}$ | 72.9 | 72.4 | 29.2 | 87.4 | 65.6 | 70.9 | 14.9 | 37.9 | 70.6 | 67.8 | 19.9 | 21.1 |
| | $-\mathbf{T}^{out}$ | 69.5 | 74.3 | 92.6 | 92.8 | 70.0 | 70.8 | 42.0 | 20.3 | 67.0 | 61.3 | 23.1 | 18.5 |

tokens are particularly task-encoding allows developers to optimize prompts by focusing on specific token structures or repetitions that are most influential. This insight can improve task performance consistency across variations in prompt phrasing and structure, ultimately making prompt creation more efficient and predictable.

**Improving Model Robustness**   The findings in our study can also inform techniques to enhance the robustness of large language models (LLMs). Since prompt sensitivity (e.g., to token arrangement) can often lead to fluctuations in performance, understanding task-encoding tokens helps mitigate these vulnerabilities. By aligning model training and prompt engineering to leverage task-encoding token characteristics, it becomes possible to minimize performance drops due to minor prompt alterations, thereby enhancing the stability and reliability of LLMs in production environments.

# P   RESULTS OF REPRESENTATION-LEVEL PARTIAL TASK-ENCODING TOKEN ABLATION

The full results on all the six datasets are shown in Table 28 and Table 29. Most of the results align with our descriptions in Section 5.1, where the task-encoding tokens **should be utilized together** to provide the best performance and that removing some of them would cause performance degeneration, demonstrated by the performance decrease from Table 28, or instability issues, shown by Table 29.

# Q   SIGNIFICANCE TEST FOR THE REPRESENTATION-LEVEL ABLATION

In this section, we report the p-value of all the pair-wise comparisons in the representation-level ablation experiments in Table 2 and Table 3. Results are shown in Table 30. Most of the ablation results show significant difference among different ablation scenarios.

Table 30: The pair-wise t-test significance results. "T" means True while "F" means False. In this table, "temp" means only keeping temp, which is zero-shot + TEMP. "temp_cont" means ablating the stopword token representations, which is Standard ICL − STOP.

| Models | Settings | AGNews | | SST2 | | TREC | | DBPedia | | RTE | | CB | |
|---|---|---|---|---|---|---|---|---|---|---|---|---|---|
| | | P-value | $p < 0.05$ | P-value | $p < 0.05$ | P-value | $p < 0.05$ | P-value | $p < 0.05$ | P-value | $p < 0.05$ | P-value | $p < 0.05$ |
| OpenLlama 3B | temp <-> cont | 0.0000581 | T | 0.0000000 | T | 0.1952165 | F | 0.0000000 | T | 0.0042427 | T | 0.0027293 | T |
| | temp <-> stop | 0.0001605 | T | 0.0571278 | F | 0.0242797 | T | 0.0000663 | T | 0.0319815 | T | 0.0985942 | F |
| | cont <-> stop | 0.0023957 | T | 0.0000001 | T | 0.1940792 | F | 0.0000000 | T | 0.0000073 | T | 0.0000698 | T |
| | temp_cont <-> cont_stop | 0.0000065 | T | 0.0385760 | T | 0.3206221 | F | 0.0000000 | T | 0.1237570 | F | 0.0544049 | F |
| | temp_stop <-> cont_stop | 0.0001166 | T | 0.4514005 | F | 0.2549225 | F | 0.0000001 | T | 0.0545474 | F | 0.0534521 | F |
| | temp_cont <-> temp_stop | 0.0000507 | T | 0.0096752 | T | 0.0005775 | T | 0.0000000 | T | 0.1208140 | F | 0.3193696 | F |
| Llama 7B | temp <-> cont | 0.0000000 | T | 0.0000001 | T | 0.0000020 | T | 0.0000001 | T | 0.0000004 | T | 0.0000001 | T |
| | temp <-> stop | 0.0000083 | T | 0.0101283 | T | 0.0002883 | T | 0.1438193 | F | 0.0000000 | T | 0.0000031 | T |
| | cont <-> stop | 0.0000060 | T | 0.0000001 | T | 0.0019529 | T | 0.0000001 | T | 0.3392017 | F | 0.0016487 | T |
| | temp_cont <-> cont_stop | 0.0000115 | T | 0.0030175 | T | 0.0005950 | T | 0.0000000 | T | 0.0000000 | T | 0.0001649 | T |
| | temp_stop <-> cont_stop | 0.0002004 | T | 0.0227328 | T | 0.0015468 | T | 0.0000000 | T | 0.0000001 | T | 0.0000094 | T |
| | temp_cont <-> temp_stop | 0.0086396 | T | 0.0003632 | T | 0.0089932 | T | 0.0000007 | T | 0.1637608 | F | 0.1553081 | F |
| Llama 13B | temp <-> cont | 0.0000000 | T | 0.0000000 | T | 0.0000082 | T | 0.0000001 | T | 0.0006445 | T | 0.1060226 | F |
| | temp <-> stop | 0.0003841 | T | 0.0000012 | T | 0.0034370 | T | 0.0002018 | T | 0.0000000 | T | 0.0010178 | T |
| | cont <-> stop | 0.0000000 | T | 0.0000202 | T | 0.0002820 | T | 0.0000000 | T | 0.0098209 | T | 0.0022848 | T |
| | temp_cont <-> cont_stop | 0.0010838 | T | 0.0000730 | T | 0.0004557 | T | 0.0000048 | T | 0.0000001 | T | 0.0002364 | T |
| | temp_stop <-> cont_stop | 0.0007763 | T | 0.0000310 | T | 0.0016544 | T | 0.0000000 | T | 0.0000000 | T | 0.0000888 | T |
| | temp_cont <-> temp_stop | 0.4411518 | F | 0.1158895 | F | 0.3323328 | F | 0.0000000 | T | 0.3148253 | F | 0.0144961 | T |
| Llama 33B | temp <-> cont | 0.0000000 | T | 0.0000003 | T | 0.1534319 | F | 0.0000000 | T | 0.0000000 | T | 0.0002244 | T |
| | temp <-> stop | 0.0007359 | T | 0.0048547 | T | 0.1797405 | F | 0.0000023 | T | 0.0000002 | T | 0.0008789 | T |
| | cont <-> stop | 0.0000000 | T | 0.0000003 | T | 0.0204911 | T | 0.0000000 | T | 0.0002626 | T | 0.4319440 | F |
| | temp_cont <-> cont_stop | 0.0001365 | T | 0.0788756 | F | 0.0032131 | T | 0.0000000 | T | 0.0000098 | T | 0.0003242 | T |
| | temp_stop <-> cont_stop | 0.0006045 | T | 0.0609501 | F | 0.0165374 | T | 0.0000011 | T | 0.0000009 | T | 0.0003821 | T |
| | temp_cont <-> temp_stop | 0.0012936 | T | 0.3583931 | F | 0.1415489 | F | 0.0001034 | T | 0.0009055 | T | 0.3979685 | F |

# R  TEMPLATE USED FOR THE RANDOM STRING EXPERIMENTS

In this section, we present all the in-context learning templates used for the random experiments in Section 5.2. In the **Random**$_{\text{fixed}}$ scenario, the $\mathbf{T}^{\text{in}}$ and $\mathbf{T}^{\text{out}}$ are consistent across all demonstrations. For the **Random**$_{\text{nonfixed}}$ scenario, we employ different random string templates for each demonstration. We use 5 random string templates for each setting, shown in Table 36, Table 37, Table 38, Table 39, and Table 40. The results in Section 5.2 are averaged over the results with all the different random string templates.

# S  SUPPLEMENTAL EXPERIMENTS FOR THE REPETITION CHARACTERISTIC

In Section 5.2.2, we examine the repetition characteristic of task-encoding tokens with random template tokens, which could not be general enough since random string tokens are less used in real-world applications. Hence, we conduct another set of experiments in this section, using template tokens with lexical meanings to test the characteristic of repetition.

These experiments includes two sets of comparisons shown in Table 31 and Table 32. The first set of templates uses meaningful, normal words but exhibits less lexical similarity to the task. The second set of templates is more closely related to the task. All comparisons are made between non-repetitive and repetitive cases.

The results presented in Table 33 show that, when random strings without lexical meanings are not used, the repetitive patterns can also enhance the final performances and help encode the task within the representations of template tokens, proving our claim that repetition is an important characteristic of task-encoding tokens.

# T  SUPPLEMENTAL EXPERIMENTS FOR THE STRUCTURAL CUE CHARACTERISTIC

In this section, we describe a set of supplemental experiments, which support the characteristic of structural cues from the perspective of representation-level ablation. An intuitive method to verify the effect of the structural cue would be using the same random strings to replace $\mathbf{T}^{\text{in}}$ and $\mathbf{T}^{\text{out}}$, making it harder for a model to parse the structure of the text. However, this would bring the factor of repetition into the process, potentially confounding the results. Hence, we instead design a one-shot **Random**$_{\text{fixed}}$ experiment. The one-shot **Random**$_{\text{fixed}}$ setting allows us to control both the characteristics of lexical meaning and repetition since the templates are made up of random strings and there is only one training demonstration. With these two characteristics controlled, we use the masking ablation method from Section 4.3.1 to confirm to what extent these random string tokens

Table 31: A 3-shot example sampled from AGNews dataset using Template 1 and Template 2.

| Template 1 |
|---|
| Classify the news articles into the categories of World, Sports, Business, and Technology. |
| **dog:** First class to the moon. London - British airline magnate Richard Branson announced a plan on Monday for the world's first commercial space flights, saying "thousands" of fee-paying astronauts could be sent into orbit in the near future.
**cat:** Technology |
| **juice:** Amazon's Holiday Pi. Leave it to Amazon.com (Nasdaq: AMZN). Apparently, the holiday season could be a rich opportunity to addict more users to Amazon's A9.
**wine:** Technology |
| **sleep:** Will historic flight launch space tourism?. Regardless, space competitions are poised to become big business.
**wake:** Technology |
| **bunny:** SMART-1 makes lunar orbit. The SMART-1 probe has entered its lunar orbit, and the history books as the first European mission to have done so. Professor David Southwood, director of science for the European Space Agency (ESA), said: "Europe ...
**easter:** |

| Template 2 |
|---|
| Classify the news articles into the categories of World, Sports, Business, and Technology. |
| **dog:** First class to the moon. London - British airline magnate Richard Branson announced a plan on Monday for the world's first commercial space flights, saying "thousands" of fee-paying astronauts could be sent into orbit in the near future.
**cat:** Technology |
| **dog:** Amazon's Holiday Pi. Leave it to Amazon.com (Nasdaq: AMZN). Apparently, the holiday season could be a rich opportunity to addict more users to Amazon's A9.
**cat:** Technology |
| **dog:** Will historic flight launch space tourism?. Regardless, space competitions are poised to become big business.
**cat:** Technology |
| **dog:** SMART-1 makes lunar orbit. The SMART-1 probe has entered its lunar orbit, and the history books as the first European mission to have done so. Professor David Southwood, director of science for the European Space Agency (ESA), said: "Europe...
**cat:** |

can function effectively as delimiters between inputs and outputs in ICL prompts. Specifically, we include results from the Zero-shot + TEMP$_{\text{1-shot}}^{\text{random}}$ and Zero-shot + ":"$_{\text{1-shot}}^{\text{random}}$ scenarios, as well as the standard results of one-shot **Random**$_{\text{fixed}}$, for a more comprehensive analysis, shown in Table 34. Examples of all the different model variants are shown in Appendix U.

We observe that adding the attention to random template token representations in the one-shot setting often leads to performance increases while masking the attention to the template tokens and only attending to ":" $+\mathbf{D}^{\text{out}}$ leads to performance decreases. This indicates that the presence of these tokens is critical to maintaining task performance. With all other characteristics being controlled, this leads us to believe that the delimiting nature of template tokens is likely to be an important characteristic in their role as task-encoding tokens.

## U   DISCUSSION ABOUT THE CHARACTERISTIC OF STRUCTURAL CUE

As discussed in Section 5.2.3, we view structural cue as the textual and structural cues present in the prompt allowing the model to distinguish between the different parts of the ICL demonstration. We believe that task-encoding tokens naturally play this role since the same types of tokens are likely to delimit pretraining text (e.g., html, markdown, etc.). An example of how we believe task-encoding tokens naturally delimit an ICL prompt is shown in Table 35, sampled from the SST2 dataset tested in our experiments. We bold and place in brackets the role of each section of the prompt as well as what types of tokens it contains.

Table 32: A 3-shot example sampled from AGNews dataset using Template 3 and Template 4.

| Template 3 |
| --- |
| Classify the news articles into the categories of World, Sports, Business, and Technology. |
| **article:** First class to the moon. London - British airline magnate Richard Branson announced a plan on Monday for the world's first commercial space flights, saying "thousands" of fee-paying astronauts could be sent into orbit in the near future. 
 **answer:** Technology |
| **input:** Amazon's Holiday Pi. Leave it to Amazon.com (Nasdaq: AMZN). Apparently, the holiday season could be a rich opportunity to addict more users to Amazon's A9. 
 **output:** Technology |
| **text:** Will historic flight launch space tourism?. Regardless, space competitions are poised to become big business. 
 **label:** Technology |
| **sentence:** SMART-1 makes lunar orbit. The SMART-1 probe has entered its lunar orbit, and the history books as the first European mission to have done so. Professor David Southwood, director of science for the European Space Agency (ESA), said: "Europe... 
 **result:** |
| Template 4 |
| Classify the news articles into the categories of World, Sports, Business, and Technology. |
| **article:** First class to the moon. London - British airline magnate Richard Branson announced a plan on Monday for the world's first commercial space flights, saying "thousands" of fee-paying astronauts could be sent into orbit in the near future. 
 **answer:** Technology |
| **article:** Amazon's Holiday Pi. Leave it to Amazon.com (Nasdaq: AMZN). Apparently, the holiday season could be a rich opportunity to addict more users to Amazon's A9. 
 **answer:** Technology |
| **article:** Will historic flight launch space tourism?. Regardless, space competitions are poised to become big business. 
 **answer:** Technology |
| **article:** SMART-1 makes lunar orbit. The SMART-1 probe has entered its lunar orbit, and the history books as the first European mission to have done so. Professor David Southwood, director of science for the European Space Agency (ESA), said: "Europe... 
 **answer:** |

During LLM pre-training, it is very likely that the model has seen text formatted in a similar way (e.g. in html, plain text with headings), in which the LLM would learn to recognize and store information in the representation of these formatting tokens. A recent study also provides supporting facts from the perspective of pretraining for this (Chen et al., 2024).

One possible way to examine if these structural cues are a key characteristic is to use the same template text for the input demonstration and the output label, disturbing the structure of the prompt and making it difficult to recognize the input and the output (e.g.: input: [demonstration]\n input: [label]). However, this would bring the confounding factor of the repetition and lexical meaning when we use multiple demonstrations. Even if we only use one example, the two same templates ("input" and "input") could form a repetition. We therefore choose to use a one-shot Random$_{\text{fixed}}$ scenario to avoid that for the experiments in Appendix T. In this case, there are no repetition or lexical meaning confounds.

Zero-shot + TEMP$_{\text{1-shot}}^{\text{random}}$ and Zero-shot + ":"$_{\text{1-shot}}^{\text{random}}$ To investigate whether these random string tokens are working as task-encoding tokens, given that they only serve as providing structural cues, we applied the representation-level ablation to see the model's performance when the test examples have or do not have access to the representations of these random string tokens, comparing the performance among [one-shot Random$_{\text{fixed}}$], [Zero-shot + TEMP$_{\text{1-shot}}^{\text{random}}$] and [Zero-shot + ":"$_{\text{1-shot}}^{\text{random}}$]. [Zero-shot + TEMP$_{\text{1-shot}}^{\text{random}}$] and [Zero-shot + ":"$_{\text{1-shot}}^{\text{random}}$] are all the representation-level ablation models based on one-shot Random$_{\text{fixed}}$, where the templates in these settings are all random strings, shown in Table 35.

The results in Table 34 show that in this setting, the model could still store the task-related information in the representations of the random string tokens, shown by the performance drop when removing their representations. There is nothing else for the model to recognize these random strings and store

Table 33: The accuracy results of the repetitive supplemental experiments.

| Models | Setting | AGNews | DBPedia | TREC | △Avg. |
|---|---|---|---|---|---|
| OpenLlama 3B | Template 1 | 33.56 | 9.72 | 5.84 | 16.37 |
| | Template 2 | **55.56** | **61.44** | **22.36** | **46.45** |
| | Template 3 | 48.24 | 50.16 | 24.84 | 41.08 |
| | Template 4 | **69.64** | **63.20** | 20.96 | **51.27** |
| Llama 7B | Template 1 | 6.00 | 0.24 | 12.92 | 6.39 |
| | Template 2 | **26.52** | **51.20** | **25.32** | **34.35** |
| | Template 3 | 19.80 | 62.28 | 1.88 | 27.99 |
| | Template 4 | **36.44** | **64.68** | **18.68** | **39.93** |
| Llama 13B | Template 1 | 7.40 | 0.00 | 5.68 | 4.36 |
| | Template 2 | **46.68** | **71.40** | **35.80** | **51.29** |
| | Template 3 | 15.60 | **76.36** | 4.96 | 32.31 |
| | Template 4 | **49.08** | 75.56 | **23.80** | **49.48** |
| Llama 2 7B | Template 1 | 52.84 | 12.08 | 35.60 | 33.51 |
| | Template 2 | **75.60** | **82.80** | **56.04** | **71.48** |
| | Template 3 | 32.96 | 76.56 | 7.04 | 38.85 |
| | Template 4 | **70.16** | **80.96** | **58.24** | **69.79** |
| Llama 2 13B | Template 1 | 8.28 | 0.04 | 2.68 | 3.67 |
| | Template 2 | **19.80** | **44.52** | **13.92** | **26.08** |
| | Template 3 | 5.84 | 59.88 | 1.72 | 22.48 |
| | Template 4 | **28.60** | **65.04** | **12.64** | **35.43** |
| Mistral 7B | Template 1 | 27.68 | 0.20 | 17.96 | 15.28 |
| | Template 2 | **67.48** | **67.60** | **31.20** | **55.43** |
| | Template 3 | 2.64 | 47.68 | 4.04 | 18.12 |
| | Template 4 | **59.12** | **70.64** | **39.12** | **56.29** |

Table 34: One-shot representation masking experiments conducted to verify if structural template formats could influence the effectiveness of the task-encoding tokens. $\mathbf{D}^{out}$ is preserved in all the settings. The results showing the greatest decrease during the ablation are underlined.

| Models | Settings | AGNews | SST2 | TREC | DBPedia | RTE | CB | Avg. |
|---|---|---|---|---|---|---|---|---|
| OpenLlama 3B | One-shot Random$_{fixed}$ | 47.5 | 51.8 | 32.6 | 19.4 | 51.8 | 42.4 | 40.9 |
| | Zero-shot+TEMP$_{1\text{-shot}}^{random}$ | 39.5 | 49.8 | 27.7 | 13.3 | 49.8 | 44.9 | 37.5 |
| | Zero-shot+":"$_{1\text{-shot}}^{random}$ | _31.5_ | _35.9_ | _23.8_ | _8.0_ | _35.9_ | _33.8_ | _28.2_ |
| Llama 7B | One-shot Random$_{fixed}$ | 3.9 | 16.9 | _3.5_ | 9.6 | 16.9 | 10.4 | 10.2 |
| | Zero-shot+TEMP$_{1\text{-shot}}^{random}$ | _2.1_ | 15.5 | 7.6 | 3.7 | 15.5 | _5.4_ | 8.3 |
| | Zero-shot+":"$_{1\text{-shot}}^{random}$ | 3.6 | _7.5_ | 14.6 | _3.0_ | _7.5_ | 6.8 | _7.2_ |
| Llama 13B | One-shot Random$_{fixed}$ | 46.1 | 47.5 | _25.0_ | 50.8 | 47.5 | 21.4 | 39.7 |
| | Zero-shot+TEMP$_{1\text{-shot}}^{random}$ | 29.2 | 48.9 | 36.1 | 35.7 | 48.9 | _14.0_ | 35.5 |
| | Zero-shot+":"$_{1\text{-shot}}^{random}$ | _14.3_ | _22.4_ | 25.4 | _22.5_ | _22.4_ | 28.9 | _22.7_ |
| Llama 33B | One-shot Random$_{fixed}$ | 69.7 | 53.0 | 37.8 | 72.8 | 53.0 | _37.6_ | 54.0 |
| | Zero-shot+TEMP$_{1\text{-shot}}^{random}$ | 61.2 | 56.3 | 41.1 | 69.2 | 56.3 | 43.0 | 54.5 |
| | Zero-shot+":"$_{1\text{-shot}}^{random}$ | _43.3_ | _41.8_ | _37.4_ | 65.0 | _41.8_ | 39.5 | _44.8_ |

the information in their representations except that these tokens serve as delimiters to inform the model distinguishing the different parts of the prompt.

Table 35: An example, sampled from the SST2 dataset tested in our experiments, of the structural cue characteristic of task-encoding tokens and how they serve as delimiters of the text prompts, where <m> means that this token is masked.

| Standard ICL |
| --- |
| Classify the reviews into the categories of Positive and Negative. *[instruction]*

Review: *[delimiter: template]*
Peppered with witty dialogue and inventive moments. *[demonstration: content + stopword]*
Answer: *[delimiter: template]*
Positive *[label]* |

| One-shot Random$_{\text{fixed}}$ |
| --- |
| Classify the reviews into the categories of Positive and Negative. *[instruction]*

dsafjkldafdsajk: *[delimiter: random template 1]*
Peppered with witty dialogue and inventive moments. *[demonstration]*
reqwiorewsdafjl: *[delimiter: random template 2]*
Positive *[label]* |

| Zero-shot+TEMP$_{\text{1-shot}}^{\text{random}}$ |
| --- |
| Classify the reviews into the categories of Positive and Negative. *[instruction]*

dsafjkldafdsajk: *[delimiter: random template 1]*
<m><m><m> ... <m> *[masked demonstration]*
reqwiorewsdafjl: *[delimiter: random template 2]*
Positive *[label]* |

| Zero-shot+":"$_{\text{1-shot}}^{\text{random}}$ |
| --- |
| Classify the reviews into the categories of Positive and Negative. *[instruction]*

<m>: *[delimiter: random template 1]*
<m><m><m> ... <m> *[masked demonstration]*
<m>: *[delimiter: random template 2]*
Positive *[label]* |

Table 36: Example #1 of the ICL template used in all of our random experiments.

| Datasets | Notations | Examples |
|---|---|---|
| | | Random$_{\text{fixed}}$ |
| CB & RTE | $\mathbf{T}^{\text{in}}$ $\mathbf{T}^{\text{out}}$ | fdafdasjklfdadf: $\{\mathbf{D}^{\text{inA}}\}$\n zcxvnmxcjkfdas: $\{\mathbf{D}^{\text{inB}}\}$\n reqwiorewsdafjl: $\{\mathbf{D}^{\text{out}}\}$\n\n |
| Other tasks | $\mathbf{T}^{\text{in}}$ $\mathbf{T}^{\text{out}}$ | dsafjkldafdsajk: $\{\mathbf{D}^{\text{in}}\}$\n reqwiorewsdafjl: $\{\mathbf{D}^{\text{out}}\}$\n\n |
| | | Random$_{\text{nonfixed}}$ |
| CB & RTE | $\mathbf{T}_1^{\text{in}}$ $\mathbf{T}_1^{\text{out}}$ $\mathbf{T}_2^{\text{in}}$ $\mathbf{T}_2^{\text{out}}$ $\mathbf{T}_3^{\text{in}}$ $\mathbf{T}_3^{\text{out}}$ $\mathbf{T}_4^{\text{in}}$ $\mathbf{T}_4^{\text{out}}$ $\mathbf{T}_t^{\text{in}}$ $\mathbf{T}_t^{\text{out}}$ | fdafdasjklfdadf: $\{\mathbf{D}^{\text{inA}}\}$\n zcxvnmxcjkfdas: $\{\mathbf{D}^{\text{inB}}\}$\n xiadfjdsalgfweqrjl: $\{\mathbf{D}^{\text{out}}\}$\n\n gfhdajkgfhdasfj: $\{\mathbf{D}^{\text{inA}}\}$\n cvxhlkdadsajfk: $\{\mathbf{D}^{\text{inB}}\}$\n yufoufgaddavfdnsl: $\{\mathbf{D}^{\text{out}}\}$\n\n rrqetrizxcsdafq: $\{\mathbf{D}^{\text{inA}}\}$\n vncmxasdgfadsl: $\{\mathbf{D}^{\text{inB}}\}$\n afdgvcxjlzxnvxzla: $\{\mathbf{D}^{\text{out}}\}$\n\n mvfvxadfawewqro: $\{\mathbf{D}^{\text{inA}}\}$\n lkajsdfopsadfp: $\{\mathbf{D}^{\text{inB}}\}$\n fgsgfskjvcdafds: $\{\mathbf{D}^{\text{out}}\}$\n\n sdsajfjdsaczvvv: $\{\mathbf{D}^{\text{inA}}\}$\n hkljfdiabasdfj: $\{\mathbf{D}^{\text{inB}}\}$\n dafhglajfdvcaol: $\{\mathbf{D}^{\text{out}}\}$\n\n |
| Other tasks | $\mathbf{T}_1^{\text{in}}$ $\mathbf{T}_1^{\text{out}}$ $\mathbf{T}_2^{\text{in}}$ $\mathbf{T}_2^{\text{out}}$ $\mathbf{T}_3^{\text{in}}$ $\mathbf{T}_3^{\text{out}}$ $\mathbf{T}_4^{\text{in}}$ $\mathbf{T}_4^{\text{out}}$ $\mathbf{T}_t^{\text{in}}$ $\mathbf{T}_t^{\text{out}}$ | dsafjkldaasdfjkl: $\{\mathbf{D}^{\text{in}}\}$\n xiadfjdsalgfweqrjl: $\{\mathbf{D}^{\text{out}}\}$\n\n ewqroudajfsdafq: $\{\mathbf{D}^{\text{in}}\}$\n yufoufgaddavfdnsl: $\{\mathbf{D}^{\text{out}}\}$\n\n eqdashcxzlreqguio: $\{\mathbf{D}^{\text{in}}\}$\n afdgvcxjlzxnvxzla: $\{\mathbf{D}^{\text{out}}\}$\n\n cxzvadeqrczxdsa: $\{\mathbf{D}^{\text{in}}\}$\n fgsgfskjvcdafds: $\{\mathbf{D}^{\text{out}}\}$\n\n vcxnkfgahvczxkl: $\{\mathbf{D}^{\text{in}}\}$\n dafhglajfdvcaol: $\{\mathbf{D}^{\text{out}}\}$\n\n |
| | | Swap |
| CB & RTE | $\mathbf{T}^{\text{in}}$ $\mathbf{T}^{\text{out}}$ | Answer: $\{\mathbf{D}^{\text{inA}}\}$\n Hypothesis: $\{\mathbf{D}^{\text{inB}}\}$\n Premise: $\{\mathbf{D}^{\text{out}}\}$\n\n |

Table 37: Example #2 of the ICL template used in all of our random experiments.

| Datasets | Notations | Examples |
|---|---|---|
| | | $\text{Random}_{\text{fixed}}$ |
| CB & RTE | $\mathbf{T}^{\text{in}}$
$\mathbf{T}^{\text{out}}$ | eszycidpyopumzg: $\{\mathbf{D}^{\text{inA}}\}$\n sgrlobvqgthjpwz: $\{\mathbf{D}^{\text{inB}}\}$\n
zbyygcrmzfnxlsu: $\{\mathbf{D}^{\text{out}}\}$\n\n |
| Other tasks | $\mathbf{T}^{\text{in}}$
$\mathbf{T}^{\text{out}}$ | eszycidpyopumzg: $\{\mathbf{D}^{\text{in}}\}$\n
zbyygcrmzfnxlsu: $\{\mathbf{D}^{\text{out}}\}$\n\n |
| | | $\text{Random}_{\text{nonfixed}}$ |
| CB & RTE | $\mathbf{T}_1^{\text{in}}$
$\mathbf{T}_1^{\text{out}}$
$\mathbf{T}_2^{\text{in}}$
$\mathbf{T}_2^{\text{out}}$
$\mathbf{T}_3^{\text{in}}$
$\mathbf{T}_3^{\text{out}}$
$\mathbf{T}_4^{\text{in}}$
$\mathbf{T}_4^{\text{out}}$
$\mathbf{T}_t^{\text{in}}$
$\mathbf{T}_t^{\text{out}}$ | eszycidpyopumzg: $\{\mathbf{D}^{\text{inA}}\}$\n sgrlobvqgthjpwz: $\{\mathbf{D}^{\text{inB}}\}$\n
zbyygcrmzfnxlsu: $\{\mathbf{D}^{\text{out}}\}$\n\n
cwknayjkywwvpty: $\{\mathbf{D}^{\text{inA}}\}$\n muzprouhvtidhqe: $\{\mathbf{D}^{\text{inB}}\}$\n
lnlgffeurextxme: $\{\mathbf{D}^{\text{out}}\}$\n\n
pdnizszmpkfjzvo: $\{\mathbf{D}^{\text{inA}}\}$\n ujulhuzkkqlfwkl: $\{\mathbf{D}^{\text{inB}}\}$\n
gflemobnbdjngii: $\{\mathbf{D}^{\text{out}}\}$\n\n
gvsrxbdoxmpablo: $\{\mathbf{D}^{\text{inA}}\}$\n ujulhuzkkqlfwkl: $\{\mathbf{D}^{\text{inB}}\}$\n
gflemobnbdjngii: $\{\mathbf{D}^{\text{out}}\}$\n\n
gvsrxbdoxmpablo: $\{\mathbf{D}^{\text{inA}}\}$\n xipddzrshrhprrb: $\{\mathbf{D}^{\text{inB}}\}$\n
npkxdzaipdpkbrs: $\{\mathbf{D}^{\text{out}}\}$\n\n |
| Other tasks | $\mathbf{T}_1^{\text{in}}$
$\mathbf{T}_1^{\text{out}}$
$\mathbf{T}_2^{\text{in}}$
$\mathbf{T}_2^{\text{out}}$
$\mathbf{T}_3^{\text{in}}$
$\mathbf{T}_3^{\text{out}}$
$\mathbf{T}_4^{\text{in}}$
$\mathbf{T}_4^{\text{out}}$
$\mathbf{T}_t^{\text{in}}$
$\mathbf{T}_t^{\text{out}}$ | eszycidpyopumzg: $\{\mathbf{D}^{\text{in}}\}$\n
zbyygcrmzfnxlsu: $\{\mathbf{D}^{\text{out}}\}$\n\n
cwknayjkywwvpty: $\{\mathbf{D}^{\text{in}}\}$\n
lnlgffeurextxme: $\{\mathbf{D}^{\text{out}}\}$\n\n
pdnizszmpkfjzvo: $\{\mathbf{D}^{\text{in}}\}$\n
gflemobnbdjngii: $\{\mathbf{D}^{\text{out}}\}$\n\n
gvsrxbdoxmpablo: $\{\mathbf{D}^{\text{in}}\}$\n
npkxdzaipdpkbrs: $\{\mathbf{D}^{\text{out}}\}$\n\n
dgldzypdptzcekq: $\{\mathbf{D}^{\text{in}}\}$\n
xobxfpnzsfzipol: $\{\mathbf{D}^{\text{out}}\}$\n\n |

Table 38: Example #3 of the ICL template used in all of our random experiments.

| Datasets | Notations | Examples |
|---|---|---|
| | | Random$_{\text{fixed}}$ |
| CB & RTE | $\mathbf{T}^{\text{in}}$ 
 $\mathbf{T}^{\text{out}}$ | bcclfxzvjitgtbs: {$\mathbf{D}^{\text{inA}}$}\n evtlfrwvtfmjtns: {$\mathbf{D}^{\text{inB}}$}\n 
 qtnheeipeustcwn: {$\mathbf{D}^{\text{out}}$}\n\n |
| Other tasks | $\mathbf{T}^{\text{in}}$ 
 $\mathbf{T}^{\text{out}}$ | bcclfxzvjitgtbs: {$\mathbf{D}^{\text{in}}$}\n 
 qtnheeipeustcwn: {$\mathbf{D}^{\text{out}}$}\n\n |
| | | Random$_{\text{nonfixed}}$ |
| CB & RTE | $\mathbf{T}_1^{\text{in}}$ 
 $\mathbf{T}_1^{\text{out}}$ 
 $\mathbf{T}_2^{\text{in}}$ 
 $\mathbf{T}_2^{\text{out}}$ 
 $\mathbf{T}_3^{\text{in}}$ 
 $\mathbf{T}_3^{\text{out}}$ 
 $\mathbf{T}_4^{\text{in}}$ 
 $\mathbf{T}_4^{\text{out}}$ 
 $\mathbf{T}_t^{\text{in}}$ 
 $\mathbf{T}_t^{\text{out}}$ | bcclfxzvjitgtbs: {$\mathbf{D}^{\text{inA}}$}\n evtlfrwvtfmjtns: {$\mathbf{D}^{\text{inB}}$}\n 
 qtnheeipeustcwn: {$\mathbf{D}^{\text{out}}$}\n\n 
 ymupnggvmbnoobq: {$\mathbf{D}^{\text{inA}}$}\n rrrnpgbmmgqymky: {$\mathbf{D}^{\text{inB}}$}\n 
 xleuwtyqnnfgzjx: {$\mathbf{D}^{\text{out}}$}\n\n 
 pdnizszmpkfjzvo: {$\mathbf{D}^{\text{inA}}$}\n qlfulxzxwfnwbum: {$\mathbf{D}^{\text{inB}}$}\n 
 jpnvgbnjjlawqfo: {$\mathbf{D}^{\text{out}}$}\n\n 
 mfkqxjoxtpmzdrs: {$\mathbf{D}^{\text{inA}}$}\n yyzdeayigwzjosn: {$\mathbf{D}^{\text{inB}}$}\n 
 pdsqooqrhvydszp: {$\mathbf{D}^{\text{out}}$}\n\n 
 rerlkjfvlvyzpmc: {$\mathbf{D}^{\text{inA}}$}\n iuumpcsevursgqe: {$\mathbf{D}^{\text{inB}}$}\n 
 tuaqblysbipihsv: {$\mathbf{D}^{\text{out}}$}\n\n |
| Other tasks | $\mathbf{T}_1^{\text{in}}$ 
 $\mathbf{T}_1^{\text{out}}$ 
 $\mathbf{T}_2^{\text{in}}$ 
 $\mathbf{T}_2^{\text{out}}$ 
 $\mathbf{T}_3^{\text{in}}$ 
 $\mathbf{T}_3^{\text{out}}$ 
 $\mathbf{T}_4^{\text{in}}$ 
 $\mathbf{T}_4^{\text{out}}$ 
 $\mathbf{T}_t^{\text{in}}$ 
 $\mathbf{T}_t^{\text{out}}$ | bcclfxzvjitgtbs: {$\mathbf{D}^{\text{in}}$}\n 
 qtnheeipeustcwn: {$\mathbf{D}^{\text{out}}$}\n\n 
 ymupnggvmbnoobq: {$\mathbf{D}^{\text{in}}$}\n 
 xleuwtyqnnfgzjx: {$\mathbf{D}^{\text{out}}$}\n\n 
 pdwunmjronsmuvu: {$\mathbf{D}^{\text{in}}$}\n 
 jpnvgbnjjlawqfo: {$\mathbf{D}^{\text{out}}$}\n\n 
 mfkqxjoxtpmzdrs: {$\mathbf{D}^{\text{in}}$}\n 
 pdsqooqrhvydszp: {$\mathbf{D}^{\text{out}}$}\n\n 
 rerlkjfvlvyzpmc: {$\mathbf{D}^{\text{in}}$}\n 
 tuaqblysbipihsv: {$\mathbf{D}^{\text{out}}$}\n\n |

Table 39: Example #4 of the ICL template used in all of our random experiments.

| Datasets | Notations | Examples |
|---|---|---|
| | Random$_{\text{fixed}}$ | |
| CB & RTE | $\mathbf{T}^{\text{in}}$ 
 $\mathbf{T}^{\text{out}}$ | hsreltpusctapir: $\{\mathbf{D}^{\text{inA}}\}$\n woxwxgwctxdumok: $\{\mathbf{D}^{\text{inB}}\}$\n 
 prlhxooromawkcp: $\{\mathbf{D}^{\text{out}}\}$\n\n |
| Other tasks | $\mathbf{T}^{\text{in}}$ 
 $\mathbf{T}^{\text{out}}$ | hsreltpusctapir: $\{\mathbf{D}^{\text{in}}\}$\n 
 prlhxooromawkcp: $\{\mathbf{D}^{\text{out}}\}$\n\n |
| | Random$_{\text{nonfixed}}$ | |
| CB & RTE | $\mathbf{T}_1^{\text{in}}$ 
 $\mathbf{T}_1^{\text{out}}$ 
 $\mathbf{T}_2^{\text{in}}$ 
 $\mathbf{T}_2^{\text{out}}$ 
 $\mathbf{T}_3^{\text{in}}$ 
 $\mathbf{T}_3^{\text{out}}$ 
 $\mathbf{T}_4^{\text{in}}$ 
 $\mathbf{T}_4^{\text{out}}$ 
 $\mathbf{T}_t^{\text{in}}$ 
 $\mathbf{T}_t^{\text{out}}$ | hsreltpusctapir: $\{\mathbf{D}^{\text{inA}}\}$\n woxwxgwctxdumok: $\{\mathbf{D}^{\text{inB}}\}$\n 
 prlhxooromawkcp: $\{\mathbf{D}^{\text{out}}\}$\n\n 
 cbptgaytithxayh: $\{\mathbf{D}^{\text{inA}}\}$\n bhxgcstisqmfnpz: $\{\mathbf{D}^{\text{inB}}\}$\n 
 mvpvoeuvgczfemz: $\{\mathbf{D}^{\text{out}}\}$\n\n 
 htkbzfizxwpeqrm: $\{\mathbf{D}^{\text{inA}}\}$\n felxgmjeuabznwd: $\{\mathbf{D}^{\text{inB}}\}$\n 
 glfwilpyrwnsujg: $\{\mathbf{D}^{\text{out}}\}$\n\n 
 frskoasvqybxcob: $\{\mathbf{D}^{\text{inA}}\}$\n bkepuhnckdaqmhx: $\{\mathbf{D}^{\text{inB}}\}$\n 
 ljttiywadveyzah: $\{\mathbf{D}^{\text{out}}\}$\n\n 
 dfpqndhxehhtser: $\{\mathbf{D}^{\text{inA}}\}$\n bvucjofrggmmcsh: $\{\mathbf{D}^{\text{inB}}\}$\n 
 koesxfmmjjjjvmp: $\{\mathbf{D}^{\text{out}}\}$\n\n |
| Other tasks | $\mathbf{T}_1^{\text{in}}$ 
 $\mathbf{T}_1^{\text{out}}$ 
 $\mathbf{T}_2^{\text{in}}$ 
 $\mathbf{T}_2^{\text{out}}$ 
 $\mathbf{T}_3^{\text{in}}$ 
 $\mathbf{T}_3^{\text{out}}$ 
 $\mathbf{T}_4^{\text{in}}$ 
 $\mathbf{T}_4^{\text{out}}$ 
 $\mathbf{T}_t^{\text{in}}$ 
 $\mathbf{T}_t^{\text{out}}$ | hsreltpusctapir: $\{\mathbf{D}^{\text{in}}\}$\n 
 prlhxooromawkcp: $\{\mathbf{D}^{\text{out}}\}$\n\n 
 cbptgaytithxayh: $\{\mathbf{D}^{\text{in}}\}$\n 
 mvpvoeuvgczfemz: $\{\mathbf{D}^{\text{out}}\}$\n\n 
 htkbzfizxwpeqrm: $\{\mathbf{D}^{\text{in}}\}$\n 
 glfwilpyrwnsujg: $\{\mathbf{D}^{\text{out}}\}$\n\n 
 frskoasvqybxcob: $\{\mathbf{D}^{\text{in}}\}$\n 
 ljttiywadveyzah: $\{\mathbf{D}^{\text{out}}\}$\n\n 
 dfpqndhxehhtser: $\{\mathbf{D}^{\text{in}}\}$\n 
 koesxfmmjjjjvmp: $\{\mathbf{D}^{\text{out}}\}$\n\n |

Table 40: Example #5 of the ICL template used in all of our random experiments.

| Datasets | Notations | Examples |
|---|---|---|
| | | Random$_{\text{fixed}}$ |
| CB & RTE | $\mathbf{T}^{\text{in}}$ | hjdxmpeccamrjzy: $\{\mathbf{D}^{\text{inA}}\}$\n agxyhmkawezafde: $\{\mathbf{D}^{\text{inB}}\}$\n |
| | $\mathbf{T}^{\text{out}}$ | ndxtrwvqugyygku: $\{\mathbf{D}^{\text{out}}\}$\n\n |
| Other tasks | $\mathbf{T}^{\text{in}}$ | hjdxmpeccamrjzy: $\{\mathbf{D}^{\text{in}}\}$\n |
| | $\mathbf{T}^{\text{out}}$ | ndxtrwvqugyygku: $\{\mathbf{D}^{\text{out}}\}$\n\n |
| | | Random$_{\text{nonfixed}}$ |
| CB & RTE | $\mathbf{T}_1^{\text{in}}$ | hjdxmpeccamrjzy: $\{\mathbf{D}^{\text{inA}}\}$\n agxyhmkawezafde: $\{\mathbf{D}^{\text{inB}}\}$\n |
| | $\mathbf{T}_1^{\text{out}}$ | ndxtrwvqugyygku: $\{\mathbf{D}^{\text{out}}\}$\n\n |
| | $\mathbf{T}_2^{\text{in}}$ | mcsgenpkdwsfknc: $\{\mathbf{D}^{\text{inA}}\}$\n egnqobhzvxjhsxh: $\{\mathbf{D}^{\text{inB}}\}$\n |
| | $\mathbf{T}_2^{\text{out}}$ | ijkdikcmiskofsg: $\{\mathbf{D}^{\text{out}}\}$\n\n |
| | $\mathbf{T}_3^{\text{in}}$ | cmaqcvtdkemdauv: $\{\mathbf{D}^{\text{inA}}\}$\n oslzaygbefxlwqt: $\{\mathbf{D}^{\text{inB}}\}$\n |
| | $\mathbf{T}_3^{\text{out}}$ | mumrjhndwmidwmj: $\{\mathbf{D}^{\text{out}}\}$\n\n |
| | $\mathbf{T}_4^{\text{in}}$ | cgmylzvslxmojvq: $\{\mathbf{D}^{\text{inA}}\}$\n tlwxsjmnfkolffl: $\{\mathbf{D}^{\text{inB}}\}$\n |
| | $\mathbf{T}_4^{\text{out}}$ | mitaowjyibjwwol: $\{\mathbf{D}^{\text{out}}\}$\n\n |
| | $\mathbf{T}_t^{\text{in}}$ | pvockachyflybtk: $\{\mathbf{D}^{\text{inA}}\}$\n wtjqmtwxbnpyqbp: $\{\mathbf{D}^{\text{inB}}\}$\n |
| | $\mathbf{T}_t^{\text{out}}$ | ydediotfezhfnbx: $\{\mathbf{D}^{\text{out}}\}$\n\n |
| Other tasks | $\mathbf{T}_1^{\text{in}}$ | hsreltpusctapir: $\{\mathbf{D}^{\text{in}}\}$\n |
| | $\mathbf{T}_1^{\text{out}}$ | prlhxooromawkcp: $\{\mathbf{D}^{\text{out}}\}$\n\n |
| | $\mathbf{T}_2^{\text{in}}$ | cbptgaytithxayh: $\{\mathbf{D}^{\text{in}}\}$\n |
| | $\mathbf{T}_2^{\text{out}}$ | mvpvoeuvgczfemz: $\{\mathbf{D}^{\text{out}}\}$\n\n |
| | $\mathbf{T}_3^{\text{in}}$ | htkbzfizxwpeqrm: $\{\mathbf{D}^{\text{in}}\}$\n |
| | $\mathbf{T}_3^{\text{out}}$ | glfwilpyrwnsujg: $\{\mathbf{D}^{\text{out}}\}$\n\n |
| | $\mathbf{T}_4^{\text{in}}$ | frskoasvqybxcob: $\{\mathbf{D}^{\text{in}}\}$\n |
| | $\mathbf{T}_4^{\text{out}}$ | ljttiywadveyzah: $\{\mathbf{D}^{\text{out}}\}$\n\n |
| | $\mathbf{T}_t^{\text{in}}$ | dfpqndhxehhtser: $\{\mathbf{D}^{\text{in}}\}$\n |
| | $\mathbf{T}_t^{\text{out}}$ | koesxfmmjjjjvmp: $\{\mathbf{D}^{\text{out}}\}$\n\n |

