# OpenReview forum: "Identifying and Analyzing Task-Encoding Tokens in Large Language Models"
_ICLR.cc/2025/Conference — Submitted to ICLR 2025_

### Official Review · Reviewer_twTo · 2024-10-27

**Soundness:** 2
**Presentation:** 3
**Contribution:** 3
**Rating:** 5
**Confidence:** 4

**Summary:**

This paper classifies the input tokens and analyzes their functions under the in-context learning (ICL) scenario. Unlike previous work, which only focused on the input sentences and labels, this work extends the ablation analysis to template tokens.

**Strengths:**

1.  The function and selection of ICL templates are not well-discussed by previous work. Some findings about the template could be helpful for the further understanding and usage of ICL.
2.  Besides the ablation study, the authors further analyze the template tokens, e.g., their lexical meaning and repetition. Some findings are novel and interesting.

**Weaknesses:**

However, I still have several concerns about this work.
1.  Ablation study design. Most previous ICL studies simply remove tokens or change them to random ones for the ablation. For this work, the authors choose to mask these tokens. Intuitively, it doesn’t matter. However, the position embedding and other practical factors may affect the conclusion. For example, for SST-2, even if you remove all ICL templates and only feed the input-label pairs, the performance could be much better than masking all template tokens (52.1%, which is almost random guessing). The authors should justify this point to support that the results are valid: showing that masking the template is equivalent to removing or randomizing it.
2.  Since this paper focuses on the task-encoding tokens, simply running experiments on the task classification datasets may be a bit weak. These datasets are more or less similar for ICL: semantic-relevant, simple classification format, … Findings on them could be quite limited as the ICL insights. It would be stronger if some other datasets e.g., QA or reasoning are included and the findings are consistent.
3.  (two minor issues). For experiment random seeds, you can select 4 random examples for each test data to decrease the variance. Then you don’t have to repeat the experiments for that many times. Besides, for the LLaMA-30B, it’s actually 33B.

**Questions:**

See weakness

---

> ### Author Response · Authors · 2024-11-25
> **Author response to Reviewer twTo (1/2)**
>
> Thanks very much to Reviewer #twTo for your kind suggestions and considerations. We appreciate that the reviewer pointed out the findings of our work could be helpful for the understanding and usage of ICL. We also thank the reviewer for acknowledging that our analyses on the characteristics of task-encoding tokens are novel and interesting. We resolve your concerns as follows:
>
> ## On the concern of the justification of the masking representations:
>
> **Reviewer’s comment:** Ablation study design. Most previous ICL studies simply remove tokens or change them to random ones for the ablation. For this work, the authors choose to mask these tokens. Intuitively, it doesn’t matter. However, the position embedding and other practical factors may affect the conclusion. For example, for SST-2, even if you remove all ICL templates and only feed the input-label pairs, the performance could be much better than masking all template tokens (52.1%, which is almost random guessing). The authors should justify this point to support that the results are valid: showing that masking the template is equivalent to removing or randomizing it.
>
> **Rebuttal:** Thank you very much for this comment and the opportunity to clarify our methodology. In our study, we intentionally use three distinct ablation methods—masking, removing, and randomizing tokens—because each serves a unique purpose in testing different aspects of the model’s behavior.
>
> 1. **Masking tokens (representation-level ablation)** allows us to observe how information from previous token representations, including positional information, is aggregated in the representations of task-encoding tokens. This is a key aspect of our study, as we hypothesize that these task-encoding token representations integrate information from preceding tokens to support the model’s task-solving capabilities. By masking tokens rather than removing them, we enable information from other parts of the prompt to propagate, allowing us to study how the model leverages this information to make accurate predictions.
>
> 2. **Removing tokens (token-level ablation),** on the other hand, tests if the information aggregation could actually benefit the task-encoding tokens by cutting off all the information flow between different types of tokens. It could also test whether the model can maintain performance without the presence of certain token types entirely. For example, when template tokens are all removed, our results show a performance drop to 0% in all the tasks. This is because removing these tokens entirely eliminates the structural cues that indicate when the model should predict the label, causing it to lose its ability to perform the task. This finding directly addresses the reviewer's concern about using input-label pairs alone: without the template structure, the model fails to understand when to make predictions.
>
> 3. Finally, **randomizing tokens**, as in our experiments in Section 5, tests how performance is impacted when task-encoding tokens are replaced with non-meaningful symbols, disrupting the specific characteristics (e.g., lexical meaning, structural cues) that we hypothesize are crucial for task-encoding. This method allows us to examine the importance of the specific characteristics of these tokens.
>
> In summary, each ablation method which the reviewer mentions serves a specific purpose in our analysis. Masking tests the model’s use of aggregated representations in task-encoding tokens, removing tests the necessity of the token type for overall task structure, and randomizing tests the role of token-specific information. Together, these approaches provide a comprehensive picture of how various token types contribute to in-context learning. Thanks again for your concerns and we have **added clarifications to the revised paper** to ensure these distinctions are clear.

---

> > ### Comment · Reviewer_twTo · 2024-11-26
> > **Followup Discussion About Weakness 1**
> >
> > Thanks for the clarification! Now the ablation studies are clear to me.
> >
> > However, I'm still curious about the zero-shot performance: why for some models, the zero-shot prompt provides much worse performance than random guess? The template might be problematic.
> >
> > For example, for SST-2, it's very simple binary classifcation and the dataset is quite balance. For OpenLLaMA, LLaMA7B, and LLaMA13B, the zero-shot accuracy is 20.0%, 29.2%, and 18.0%. It's hard to believe the performance is that bad. I, as well as many previous researchers, have run similar experiments on this dataset, and the zero-shot test accuracy (with commonly used prompt template) is at least 90%.
> >
> > If the baseline is problematic, it's hard to believe that the obtained conclusions are convincing.

---

> > > ### Author Response · Authors · 2024-11-27
> > > **Prediction example for Llama-1 models conducting the zero-shot SST2 task**
> > >
> > > ### Llama-1-13B
> > >
> > > **Example #1**
> > >
> > > <s> Classify the reviews into the categories of Positive and Negative.
> > >
> > > Review: the tug-of-war at the core of beijing bicycle becomes weighed down with agonizing contrivances , overheated pathos and long , wistful gazes .
> > >
> > > Sentiment: **the**
> > >
> > > **Example #2**
> > >
> > > <s> Classify the reviews into the categories of Positive and Negative.
> > >
> > > Review: the ill-conceived modern-day ending falls flat where it should deliver a moral punch .
> > >
> > > Sentiment: **the**
> > >
> > > **Example #3**
> > >
> > > <s> Classify the reviews into the categories of Positive and Negative.
> > >
> > > Review: while undercover brother is definitely one for the masses , it 's also full of sharp , smart satire .
> > >
> > > Sentiment: **while**
> > >
> > >
> > > ### Llama-1-7B
> > >
> > > **Example #1**
> > >
> > > <s> Classify the reviews into the categories of Positive and Negative.
> > >
> > > Review: it should be mentioned that the set design and interiors of the haunted vessel are more than effectively creepy and moodily lit .
> > >
> > > Sentiment: **it**
> > >
> > >
> > > **Example #2**
> > >
> > > <s> Classify the reviews into the categories of Positive and Negative.
> > >
> > > Review: winds up feeling like lots of other quirky movies that try to score hipness points with young adults .
> > >
> > > Sentiment: **positive**
> > >
> > > **Example #3**
> > >
> > > <s> Classify the reviews into the categories of Positive and Negative.
> > >
> > > Review: ` dragonfly ' is a movie about a bus wreck that turns into a film wreck .
> > >
> > > Sentiment: **`**
> > >
> > >
> > > ### OpenLlama-3B
> > >
> > > **Example #1**
> > >
> > > <s>Classify the reviews into the categories of Positive and Negative.
> > >
> > > Review: an earnest , heartrending look at the divide between religious fundamentalists and their gay relatives .
> > >
> > > Sentiment: **a**
> > >
> > > **Example #2**
> > >
> > >  <s>Classify the reviews into the categories of Positive and Negative.
> > >
> > > Review: in his debut as a film director , denzel washington delivers a lean and engaging work .
> > >
> > > Sentiment: **positive**
> > >
> > > **Example #3**
> > >
> > > <s>Classify the reviews into the categories of Positive and Negative.
> > >
> > > Review: the cold and dreary weather is a perfect metaphor for the movie itself , which contains few laughs and not much drama .
> > >
> > > Sentiment: **the**

---

> ### Author Response · Authors · 2024-11-25
> **Author response to Reviewer twTo (2/2)**
>
> ## On the concerns of QA tasks:
>
> **Reviewer’s comment:** Since this paper focuses on the task-encoding tokens, simply running experiments on the task classification datasets may be a bit weak. These datasets are more or less similar for ICL: semantic-relevant, simple classification format, … Findings on them could be quite limited as the ICL insights. It would be stronger if some other datasets e.g., QA or reasoning are included and the findings are consistent.
>
> **Response:** Thank you for the valuable suggestion. In response, we have added experiments testing all models (including OpenLlama 3B, Llama 7B, 13B, 33B, Llama 2 7B, 13B, and Mistral 7B) on the Commonsense QA dataset (https://huggingface.co/datasets/tau/commonsense_qa). Results of these additional experiments are presented **in Appendix K and Table 23 of the current revision**. These results demonstrate that our conclusions hold consistently across various model sizes and configurations, extending to QA tasks within in-context learning scenarios. This additional evidence further supports the robustness of our findings, showing that the roles of task-encoding tokens and the patterns observed in previous tasks are similarly applicable in the domain of question answering. We appreciate your suggestions again and hope these supplementary results could resolve your concerns.
>
>
> ## On the concerns of the other issues,
>
> **Reviewer’s comment:**  For experiment random seeds, you can select 4 random examples for each test data to decrease the variance. Then you don’t have to repeat the experiments that many times. Besides, for the LLaMA-30B, it’s actually 33B.
>
> **Response:** Thanks a lot for the suggestion on the experimental design! We have also changed the Llama-30B to Llama-33B in the current revision accordingly.
>
>
> Thanks again for your hard work and kind reviews! We sincerely hope that our clarification and revisions could resolve your concerns.

---

> ### Author Response · Authors · 2024-11-27
> **Author response to Reviewer twTo regarding the followup discussion about weakness 1**
>
> Thanks a lot to Reviewer twTo for your acknowledgement of our ablation studies and your concerns of the zero-shot baseline performance! We address the question as follows:
>
> **Reviewer’s comment:** For example, for SST-2, it's very simple binary classifcation and the dataset is quite balance. For OpenLLaMA, LLaMA7B, and LLaMA13B, the zero-shot accuracy is 20.0%, 29.2%, and 18.0%. It's hard to believe the performance is that bad. I, as well as many previous researchers, have run similar experiments on this dataset, and the zero-shot test accuracy (with commonly used prompt template) is at least 90%.
>
> **Response:** Thanks a lot for the insightful question! We believe that the main reason why LLaMA-1 series performs poorly in the zero-shot scenario is due to **their inefficiency in calibrating the output label space solely from the instruction (which is the case for LLaMA-1 models only)**. We support this point through the following two aspects:
>
> 1. We reran the experiments for these models on the zero-shot SST-2 task, and the results were consistent. We checked the predictions and found that the models could hardly predict the target labels specified in the instructions. (Three examples for each model are attached below.) Only a few predictions were correct in terms of the target label space (Positive, Negative). This is the reason why the zero-shot accuracy for these models is so low. In this case, adding one example to the context greatly helps the model. We conducted another set of one-shot ICL experiments, and the results for these four LLaMA models were 51.7%, 77.3%, 71.6%, and 88.5% (using 15 random seeds), respectively, which aligns with your experience. We also observed that with these one-shot examples, the models were able to predict the correct labels.
>
> 2. We think this scenario is mainly due to the pretraining data used for Llama-1 series models. In our experiments with **Llama-2 and Mistral models**, shown in Table 15 of Appendix D, the zero-shot results are 50.4, 90.8, and 84.4, which aligns with the reviewer’s experience. We used the same pipeline for running all the experiments related to the zero-shot tasks, which could also support the fidelity of our results with Llama-1 models.
>
> 3. You are also welcome to review the code in 'processing_dataset_allrank_llama2.py' and 'processing_dataset_allrank_15.py' from L400 in our supplemental materials to see the code we used for testing the performance of zero-shot and standard ICL models.
>
> Overall, we believe the low performance in the zero-shot scenario is primarily due to the LLaMA-1 models’ inefficiency in calibrating the output label space based on the instruction alone. We hope that **the example inspection, one-shot supplemental results, and the LLaMA-2 & Mistral performance results** will help address your concerns. At a higher level, given these new observations, we believe that part of the function of task-encoding tokens may be to help the model generate over the correct space of labels.  Thanks again for your hard work and thoughtful reviews!

---

> ### Author Response · Authors · 2024-11-29
> **Reply to Followup Discussion About Weakness 1**
>
> Dear Reviewer twTo,
>
> Thank you again for your thoughtful feedback on our submission! We have carefully considered your comments and have provided a detailed rebuttal addressing the points you raised. We would greatly appreciate it if you could kindly review our responses. Your further insights will be invaluable to ensure the clarity and strength of our revisions.
>
> Thank you once again for your time and consideration.
>
> Best regards,
>
> Submission #7037 Authors

---

> ### Author Response · Authors · 2024-12-02
>
> Dear Reviewer twTo,
>
> We hope this message finds you well and sincerely appreciate your time and effort in reviewing our paper. We have submitted a detailed rebuttal addressing all of your comments and suggestions, and we would greatly appreciate your feedback on the revisions. **Please let us know if your concerns and questions are addressed, and if so, we would be grateful if you would be willing to raise your score, thanks so much!** We are happy to engage in further discussions.
>
> We truly value your input and appreciate your time. Thank you once again for your consideration.
>
> Best regards,
>
> Submission #7037 Authors

---

> > ### Comment · Reviewer_twTo · 2024-12-02
> > **Response**
> >
> > Thanks for the detailed explanation. Now I know why the zero-shot setting provides worse-random performance.
> >
> > However, I don't think this is a reasonable settings. Most previous work formalize ICL classification task by looking at the log-probability distribution on label tokens. This work uses the generation setting across the whole vocabulary. Essentially, the performance reveals both:
> >
> > 1. if the model can correctly understand the instruction and limit its output on the label space;
> >
> > 2. if the model can perform the classification task under the zero-shot/ICL setting.
> >
> > Therefore, I don't think the performance can correctly show the ICL performance. In fact, if the authors follow the setting of previous work (looking the distribution on label tokens instead of the whole vocabulary), the improvement would be minors since zero-shot or plain ICL setting is sufficient for the model to perform the tasks (used in the paper) pretty well.
> >
> > Considering above limitations, I would like to keep my score.
> >
> > Best,
> > Reviewer twTo

---

> ### Author Response · Authors · 2024-12-03
> **Response to Reviewer twTo**
>
> Thank you for your detailed feedback and for the constructive points you’ve raised. We appreciate your perspective and the opportunity to further clarify our experimental design. Below, we address your comments:
>
> 1. We understand the reviewer's perspective that our current evaluation, which considers the generated token directly, conflates the model's ability to restrict its output to the label space with its classification performance. However, we would like to clarify that our evaluation methodology, which focuses on the final generated token accuracy, aligns with **most of the ICL literature [1-11]** and also the practical classification setup. Our current results demonstrate consistent trends across various datasets, models, and ablation strategies, making it unlikely that they are solely artifacts of the evaluation setup.
>
> 2. Meanwhile, these two lines of evaluation methods both have their merits. Using logits helps avoid the noise introduced by other words in the entire vocabulary. However, evaluating the predicted label tokens better reflects the observed behavior of large language models (LLMs) and how they perform in real-world applications. In this work, we focus on the observed behavior of LLMs and their applicability to real-world user settings. Hence, we prefer to use the generated responses of LLMs to evaluate performance. We agree that a discussion of existing work studying next-word-prediction probabilities would better situate our paper's contribution within the broader field of LLM analysis. We will add this discussion, along with results evaluated using log-probabilities, in the next revision.
>
> 3. Additionally, identifying the output label space is a key part of the working mechanism of in-context learning, as discussed in Section 5 of [5]. We believe that including this behavior in our analysis is important for providing a more comprehensive picture of the working mechanism of in-context learning.
>
> 4. More importantly, the results from models that are able to successfully identify the target label space and achieve relatively high performance on zero-shot classification tasks (e.g., LLaMA 30B, LLaMA 2 13B, Mistral 7B) provide additional support for our findings since they performed worse after the removal of template and stopword tokens. In addition to the classification tasks, we have also included results for machine translation and question answering tasks in Appendix J and Appendix K. While the zero-shot performance on these tasks is not high, they still align with our main conclusions. We would like to point out that these tasks and models do not entirely align with the statement that 'zero-shot ICL setting is sufficient for the model to perform the tasks (used in the paper) pretty well.' Furthermore, for models with strong zero-shot performance, removing the representations of template tokens still leads to a noticeable decrease in performance compared to the standard ICL. This further supports our conclusion that template and stopword tokens are more likely to function as task-encoding tokens.
>
> We fully respect your decision to retain your score based on the concerns you raised and truly appreciate the thoughtful feedback you have provided. We believe that by addressing these limitations in the discussion section of the next revision, alongside the inclusion of additional experiments comparing different evaluation methods, we can present a more comprehensive and clearer view of the model’s ICL performance.
>
> Once again, we sincerely thank you for your valuable input! We hope that these revisions and responses will help enhance the clarity and robustness of our experimental results.
>
>
> Best regards,
>
> Paper 7037 Authors
>
> -------
>
>
> [1] Unveiling in-context learning: A coordinate system to understand its working mechanism.
>
> [2] Active Example Selection for In-Context Learning
>
> [3] Label words are anchors: An information flow perspective for understanding in-context learning.
>
> [4] What In-Context Learning “Learns” In-Context: Disentangling Task Recognition and Task Learning
>
> [5] Rethinking the Role of Demonstrations: What Makes In-Context Learning Work?
>
> [6] What Makes Good In-Context Examples for GPT-3?
>
> [7] In-Context Learning with Many Demonstration Examples
>
> [8] In-Context Learning Creates Task Vectors
>
> [9] Structured Prompting: Scaling In-Context Learning to 1,000 Examples
>
> [10] Understanding In-Context Learning via Supportive Pretraining Data
>
> [11] Calibrate Before Use: Improving Few-Shot Performance of Language Models

---

### Official Review · Reviewer_SovD · 2024-11-04

**Soundness:** 3
**Presentation:** 3
**Contribution:** 3
**Rating:** 6
**Confidence:** 3

**Summary:**

The authors study task encoding tokens (I'll refer to these as TETs for brevity), tokens whose representations strongly influence an LLM's ICL performance. They use two methods to evaluate TETs: masking attention maps (representation-level ablation) of possible TETs and removing TETs themselves (token-level ablation) by removing these tokens from the input prompt. The authors explore 3 candidates for TETs: stopwords, template tokens, and content (i.e. tokens that aren't stop/template/ICL answers) tokens. The authors find that template tokens are the strongest candidates for TETs, followed by stopwords; ablating content words has surprisingly little impact on performance. This stands in contrast to prior work which only studies how ICL answer tokens impact ICL performance. They dive deeper by analyzing what characteristics of tokens LLMs use for identifying TETs, and they find that lexical meaning, repetition, and structural cues are all important for ICL performance, particularly for 3B-13B size models, suggesting these characteristics are all used by LLMs for identifying TETs.

**Strengths:**

- This is an interesting problem and well-motivated. "How do LLMs represent task specifications and solutions?" is an important question and studying token-level representations seems like a promising approach to answering it.
- Good descriptions of most methods and experiments, including many details in appendices.
- Thorough analyses, including many alternate conditions and a deep dive into what lexical characteristics are used to identify TETs.
- Interesting result that stopwords more strongly encode task information compared with "content" words.

**Weaknesses:**

- I'm confused about the token-level ablation experiment/results (Sec. 4.3.2; Fig. 3). Footnote 5 on page 7 doesn't seem to match Figure 2, Bottom, since the Test example tokens aren't shown as ablated, and I'm not sure how the accuracies in Figure 3 could possibly be so high if all content tokens are removed *in the test example*. Answering the text example question correctly seems to depend on the content words, instead of getting a relatively nonsensical prompt like "the the . by a the : by .". Am I misunderstanding something?
	- I might be willing to raise my score if this was sufficiently addressed.
- The first two intro figures could be improved.
	- Figure 1 is a bit hard to parse, and it's confusing that most boxes are labelled with tokens but the blue boxes are labelled "Test example" in the same font.
	- Figure 2 could benefit from token examples like figure 1 has, and doesn't show that there are multiple ICL examples of (T^in, D^in, T^out, D^out). Assuming my interpretation is correct, the caption of Fig 2 should also mention that this is an example of content/stop-word ablation, since for template word ablation T^in and T^out would be ablated instead of D^in.
- No results for larger/newer models (70b+, IFT), and some of the results such as the analyses in Section 5 seem to apply less for the largest model the authors tested (llama 30b).
- The baselines / null hypothesis seem a little weak. Indeed it is interesting that TETs are template+stop words instead of content words, however, I'm not sure what prior work would predict.
	- (a) If "information is being propagated from the representations of content tokens to the representations of the task-encoding tokens", this definition of TET seems to deviate significantly from "human attention to informative words", which is more like token-level attention. I'm also not sure how humans would do on a similar task, e.g. in an experiment where you mask all words except content words.
	- (b) The authors define "content words" as being any non-[stop/template/answer] tokens, which seems very broad given that "content tokens" in this case includes most tokens (e.g. in Table 14). While it's surprising that performance remains so high with no content tokens, I wonder if there are specific content tokens which are also TETs, such as named entities, or how a baseline of N randomly selected unmasked tokens would do.


Minor:
- Appendix text in many sections is very brief.

**Questions:**

- For token-level ablation, given that you're ablating tokens from "both the demonstration tokens as well as the tokens in the test example", I'm confused how the -CONT results in Figure 3 are this high (or even above chance). How is it possible that the test examples are being answered correctly with inputs like "the the . by a into : by ." (as in Table 14), or can you help me understand what's going on here?
- How would you expect larger models, and those with fine-tuning such as SFT and RLHF/DPO to perform, particularly in light of work such as [1]?
- Can TETs generalize outside the format of few-shot ICL? Do you expect TETs in zero-shot learning (e.g. [2]), and would results such as the importance of template tokens change in 0-shot?
- Could you provide concrete examples of how this work could be used to develop better ICL prompts (Appendix J)?


[1] Wei et al. (2023). Larger language models do in-context learning differently.

[2] Kojima et al. (2022). Large language models are zero-shot reasoners.

---

> ### Author Response · Authors · 2024-11-25
> **Author response to Reviewer SovD (1/4)**
>
> Thank you very much to Review SovD for your kind suggestions and comments. We appreciate that the reviewer acknowledges that the topic of our study is interesting and well-motivated. We also thank the reviewer for pointing out that the overall description, analyses and the findings about the stopword tokens are comprehensive and interesting. Here are our responses for your concerns:
>
> ## Clarification of Token-Level Ablation:
>
> **Reviewer’s comment:** The token-level ablation results seem inconsistent with Figure 2, and it’s unclear how accuracy remains high if content tokens are removed from both demonstrations and the test example.
>
> **Response:** Thanks so much for pointing this out! We apologize for the unclarity and the mistake we made in our initial submission. We want to correct ourselves that: in the token-level ablation experiments, the stopword and content token ablation only includes the tokens in the demonstration; however, for the template tokens, the ablation is conducted both for the demonstration and for the test examples. We made this experimental design choice since we wanted to make sure that the template structure of the demonstrations and the test examples are consistent. This is the reason why the performance of token-level content token ablation is relatively high. We have modified all the related descriptions in the current revision including **the footnote 5 and a new Figure 2**. To compensate for this design choice, we have also added another set of experiments in the **Appendix N** to this revision, where only the template tokens in the demonstrations are ablated. Experimental results in **Table 27** show similar findings to those in the paper that template tokens are important for maintaining the ICL performance.
>
> We are sincerely sorry for this oversight and hope that our revision could clarify your concerns. Meanwhile, we want to emphasize that this revision and the newly added experiments won’t affect the original findings in Section 4.3.2, where information is propagated from the representations of content tokens to the representations of task-encoding tokens and this information aggregation enhances the ability of task-encoding tokens to improve the final task performance, partially explaining the mechanism of in-context learning. Additionally, if you consider this as a major point, we are happy to swap the template-level template token ablation results between the ones in the Appendix N and the ones in Section 4.3.2 to make the token-level ablation analysis and results more consistent across different token types in the next revision.
>
> ## On the concerns of the first two figures:
>
> **Reviewer’s comment:** The first two intro figures could be improved. Figure 1 is a bit hard to parse, and it's confusing that most boxes are labeled with tokens but the blue boxes are labeled "Test example" in the same font. Figure 2 could benefit from token examples like figure 1 has, and doesn't show that there are multiple ICL examples of (T^in, D^in, T^out, D^out). Assuming my interpretation is correct, the caption of Fig 2 should also mention that this is an example of content/stop-word ablation, since for template word ablation T^in and T^out would be ablated instead of D^in.
>
> **Response:** Thank you very much for your suggestion regarding our presentation of Figure 1 and Figure 2. For Figure 1, we changed a separate font for the test examples and used a brace to indicate the difference between the blue test examples and the preceding token representations. For Figure 2, your interpretation is absolutely correct, and we have revised it according to your advice. We introduced test examples to Figure 2 and made a clear distinction between the token-level ablation of content, stopword, and template tokens. Please refer to the current revision to see the modification we made for these two figures.

---

> ### Author Response · Authors · 2024-11-25
> **Author response to Reviewer SovD (2/4)**
>
> ## On the concerns of SFT and Larger models:
>
> **Reviewer’s comment:** No results for larger/newer models (70b+, IFT), and some of the results such as the analyses in Section 5 seem to apply less for the largest model the authors tested (llama 30b).
>
> **Response:** Thank you very much for this suggestion. We added representation-level ablation experiments involving two models after the instruction tuning process (Llama 2 7B Chat and Llama 2 13B Chat) **to this revision at Appendix E and Table 16**. The results align with the findings discussed in Section 4.3.1, with only one exception: the TREC dataset. In this dataset, the input data is structured in a question-answering format (e.g., Who/What/When did ...). We hypothesize that, during the supervised fine-tuning process, tokens associated with these question formats may also serve as task-encoding tokens, although they are currently categorized as content tokens in our experiments. Overall, these supplemental results prove further evidence to our findings in the main paper.
>
> As for larger models, unfortunately we don’t have enough resources to conduct all the representation-level experiments. Nevertheless,  the experimental findings presented in Section 4 are well-supported even with larger models.  For your concern that the analyses in Section 5 seem to apply less for the largest model, we think that larger language models have better capacity and generalization ability to recognize more task-encoding characteristics, leading to their higher tolerance and robustness for the disruption of a single task-encoding characteristic. In other words, as the model size increases, it can learn to use more than just one characteristic to leverage the task-encoding tokens. As a result, the impact of ablating a single characteristic becomes smaller.
>
> **Reviewer’s comment:** How would you expect larger models, and those with fine-tuning such as SFT and RLHF/DPO to perform, particularly in light of work such as [1]?
>
> **Response:** Thanks for the question! As discussed in the above rebuttal, we think the main conclusions where template and stopword tokens are more likely to be task-encoding tokens would still remain consistent as the experiments for representation-level ablation do not change much for larger models. Meanwhile, the model would be more robust in terms of the utilization of task-encoding tokens. We hope our clarification could resolve your concern.

---

> ### Author Response · Authors · 2024-11-25
> **Author response to Reviewer SovD (3/4)**
>
> ## On the concerns of baselines/null hypotheses:
> **Reviewer’s comment:** I wonder if there are specific content tokens which are also TETs, such as named entities, or how a baseline of N randomly selected unmasked tokens would do.
>
> **Response:** Thank you very much for this comment. We agree that there could be other task-encoding tokens within the content tokens, depending on the specific tasks and the post-training processing applied to the models (e.g., the TREC results with instruction-tuned models). However, our study primarily focuses on identifying task-encoding tokens in a broad range of cases. Supported by experimental results across classification, machine translation, and question answering tasks using various pretrained large language models, we found that template and stopword tokens are **more likely** to serve as task-encoding tokens compared to content tokens. We acknowledge that finer-grained token identification in certain contexts is an important direction for future work and have **added it to the limitation section of the current revision.**
>
> We also thank the reviewer for the suggestion of using a baseline of N randomly selected unmasked tokens. To address this concern, we include a detailed breakdown of token counts across the categories (content, stopword, and template) as follows and add the data to **Table 24 of the Appendix L in this revision.** As we can see, the set of template tokens and the stopword tokens is quite small as compared to the set of content tokens, which proves that our findings are not simply a result of ablating a larger set of tokens. Additionally, to control for potential size-based biases, we also implemented a random ablation baseline by randomly selecting the same number of tokens for each token type and letting the test example only attend to their representations, shown in **Table 25 of the Appendix L.** We also included the baseline of random selecting a certain number of tokens from all the tokens as a comparison.
>
> Results demonstrate that when the model is exposed to an equal number of each type of token representation, the performance consistently improves with template and stopword tokens, outperforming both content tokens and the random baseline. In contrast, models attending to the same number of content tokens consistently underperform relative to the random baseline. Additionally, all results improve when more tokens are included in the attention of test examples. This experiment further supports our claim that template and stopword tokens are more likely to serve as task-encoding tokens.
>
>
> **Reviewer’s comment:** I'm also not sure how humans would do on a similar task, e.g. in an experiment where you mask all words except content words.
>
> **Reponse:** Thank you very much for this insightful question. Intuitively, we believe that it would not make a significant difference as long as humans are able to distinguish between the demonstration inputs and outputs, and understand the connections between them. While our definition does deviate somewhat from the "human attention to informative words," the primary goal of this study is to investigate how large language models represent task specifications and solutions at the model level. We believe our main findings remain valid, regardless of how humans might approach similar tasks.
>
> ## On the concerns of the generalization of TET:
>
> **Reviewer’s comment:** Can TETs generalize outside the format of few-shot ICL? Do you expect TETs in zero-shot learning (e.g. [2]), and would results such as the importance of template tokens change in 0-shot?
>
> **Response:** Thank you very much for this thoughtful question. Our study focuses specifically on the scope of in-context learning (ICL), where the model learns tasks from a few examples provided within the context. While we believe that certain tokens may also play a crucial role in zero-shot learning scenarios, additional ablation-based analyses should be run to confirm this hypothesis. For instance, inspired by our ablation approach on ICL prompts, similar analyses could be applied to the zero-shot token setting by treating the final “:” token in the 0-shot prompt as the “test example” and using the same token categorization for the token-level and representation-level ablations of the previous tokens. This setup for a 0-shot ablation analysis would be in line with similar work ((Todd et al. 2023, Hendel et al. 2023). This discussion also shows that our ablation-based analysis method is transferable to other settings. Given time constraints and because our paper aims to further elucidate the working mechanisms of in-context learning specifically, we have chosen to **add this discussion as future work in the limitations section of the current revision.** However, if the reviewer believes this to be a critical additional analysis, we are happy to run additional experiments and place them in the paper’s appendix.

---

> ### Author Response · Authors · 2024-11-25
> **Author response to Reviewer SovD (4/4)**
>
> ## On the concerns of the examples for the prompt engineering:
>
> **Reviewer’s comment:** Could you provide concrete examples of how this work could be used to develop better ICL prompts (Appendix J)?
>
> **Response:** Thank you for the question! In our preliminary explorations for another project, we found examples that especially benefited from our findings. Specifically, we observed that using the default template from OpenAI—where all input prompts are placed in the slot following “User:” and the response is generated after “System:” (shown as Template 1 below)—often resulted in unsatisfactory performance for in-context learning with LLMs. This is likely because the characteristics analyzed in our study are disrupted in this setup. Guided by our insights, we found that structuring the prompt in a more repetitive and organized way significantly improved performance. For example, when we switched to Template 2 (shown below), the few-shot learning performance using Llama2-70b-chat **improved from 31.30 to 53.43 on the 3-shot SuperNI task and from 36.08 to 57.79 on the 5-shot MMLU task.** Such suboptimal prompt designs are common and could benefit substantially from the insights provided by our work. Overall, we believe this example demonstrates how our findings can be applied to create more effective ICL prompts.
>
>
> ## On the concerns of the appendix:
>
> **Reviewer’s comment:** Appendix text in many sections is very brief.
>
> **Response:** Thanks a lot for your comments! We have extended the appendix text that is too brief in this revision.
>
>
> Thanks again for your hard work and kind reviews! We sincerely hope that our clarification and revisions could resolve your concerns.

---

> ### Author Response · Authors · 2024-11-25
> **Template examples used in the experiments of prompt engineering.**
>
> ## Template 1:
>
> **User:**
>
> Task input: [demonstration input]
>
> Task output: [demonstration label]
>
>
> Task input: [demonstration input]
>
> Task output: [demonstration label]
>
>
> Task input: [demonstration input]
>
> Task output: [demonstration label]
>
>
> Task input: [test example input]
>
> Task output:
>
> **System:**
>
>  [predicted answer]
>
>
> ---
>
> ## Template 2:
>
> **User:**
>
> [demonstration input]
>
> **System:**
>
> [demonstration label]
>
> **User:**
>
> [demonstration input]
>
> **System:**
>
> [demonstration label]
>
> **User:**
>
> [demonstration input]
>
> **System:**
>
> [demonstration label]
>
> **User:**
>
> [test input]
>
> **System:**
>
>  [predicted answer]

---

> ### Author Response · Authors · 2024-11-29
>
> Dear Reviewer SovD,
>
> Thank you for your constructive suggestions on our paper! We have carefully revised the manuscript based on your feedback and have included a detailed rebuttal in the current version. We kindly request that you review our responses and let us know if they address your concerns. Your continued feedback is crucial to us as we work towards improving the paper.
>
> We look forward to hearing from you soon.
>
> Best regards,
>
> Submission #7037 Authors

---

> ### Author Response · Authors · 2024-12-02
>
> Dear Reviewer SovD,
>
> Thank you again for your thoughtful review of our paper. We have carefully revised the manuscript in response to the feedback provided and have included a rebuttal addressing all points raised. **Please let us know if your concerns and questions are addressed, and if so, we would be grateful if you would be willing to raise your score, thanks so much!** We are happy to engage in further discussions.
>
> We truly value your input and appreciate your time. Thank you once again for your consideration!
>
> Best regards,
>
> Submission #7037 Authors

---

> ### Comment · Reviewer_SovD · 2024-12-02
>
> Thank you for the thorough responses and updates to your paper. The figures are improved and some of my concerns were sufficiently addressed.
>
> A few standing points for me:
> - I appreciate your templates example to illustrate how your results can influence prompt design.  However, I'm a little confused about whether this change corresponds to the "template" tokens studied in your paper. In the paper, template tokens seem to be tokens such as "Article:" and "Answer:". However, your Template 1/2 example seems to vary whether input text is parsed as a single message (Template 1) instead of a sequence of alternating messages (Template 2).
> - This also makes me wonder what are the exact format of the model inputs is for your new experiments in Appendix E with llama 2 chat models, and whether there you're ablating template tokens such as "Article:" or tokens such as "User:". For these chat models, and for your experiment with Templates 1/2 in your response, did you add appropriate chat template tokens as in [1, 2]?
> - For Templates 1/2 you provided in your response, shouldn't the roles be "User"/"Assistant" instead of "User"/"System"? Usually the "system" role provides instructions, whereas "assistant" responds to user inputs.
> - I appreciate that your resources are limited and so experiments with bigger models (70b+) are difficult. However, I think this still impacts how convincing it is that these results will generalize to bigger and more state-of-the-art LLMs. Limited experiments with bigger models might suffice, and/or experiments with <8 bit quantization.
> - I'm curious about a point that Reviewer twTo raised about previous work, but unfortunately I'm not sure which specific works they're comparing this to.
>
>
>
> I've raised my score to 6. Overall I think this is a pretty good paper with interesting results.
>
>
> [1] https://www.llama.com/docs/model-cards-and-prompt-formats/other-models#meta-llama-2
>
> [2] https://huggingface.co/docs/transformers/main/en/chat_templating

---

> ### Author Response · Authors · 2024-12-03
>
> Thank you very much for your thoughtful feedback and for increasing your score to 6. We greatly appreciate your acknowledgment of our paper, your careful review, and the constructive points you've raised. We address your current questions as follows:
>
> 1. **Confusion about the Influence of Our Findings on Prompt Design:** We appreciate your concern regarding how our findings influence prompt design. The design of the prompts is especially informed by the repetitions and text formatting throughout the entire prompt, which we describe as “structural cues” in the paper. Specifically, in the second prompt, we added more repetitions in the form of “User-System” pairs to indicate the input and output. Additionally, the text formatting was adjusted to provide clearer structural cues, signaling that the model should generate the answer following the “System:” tokens. These modifications are related to our results where the performance would decrease after the disruption of the repetition and structural cue characteristics. We hope this clarification helps make the working of this example clearer.
>
> 2. **Template Used in LLaMA 2 Chat Experiments:** Thank you for your question about the template used in our LLaMA 2 chat experiments. We used the same templates as those in the main body of the paper to maintain consistency and avoid introducing other influencing factors. We agree with your suggestion that templates suitable for supervised fine-tuned (SFT) models could also be tested to further validate our conclusions, and we will include such experiments in the next revision.
>
> 3. **Replacing “System” with “Assistant”:** Thank you for raising this concern. We will retest the results using the "Assistant" label instead of "System" in the next revision and provide updated results.
>
> 4. **Response to Reviewer twTo on Evaluation Methods:** We have responded to Reviewer twTo's concerns regarding our evaluation methods. Please refer to our comments for our detailed response.
>
> Once again, thank you for your valuable feedback and for kindly raising the score. We sincerely appreciate your effort in helping us improve this paper!
>
>
> Best regards,
>
> Paper 7037 Authors

---

### Official Review · Reviewer_1DaS · 2024-11-05

**Soundness:** 3
**Presentation:** 2
**Contribution:** 3
**Rating:** 6
**Confidence:** 3

**Summary:**

This work defines "task encoding tokens" as input tokens where masking its representation leads to a drop in in-context learning performance. They find that template and stopword tokens, which are not informative of the task itself, are more likely to be task encoding than content tokens. They also analyze the representations of template tokens, since they contribute the most to ICL performance, finds that different template tokens are mostly all important and their representations may serve as information anchors, lexically meaningful template tokens lead to better performance, and repetition of template tokens is important.

**Strengths:**

Investigating the contributions of different input tokens to ICL is an interesting study in the broader theme of works attempting to better understand ICL. Diving input tokens into three categories and comparing the importance of each category is an interesting way to conduct this analysis. The paper also performs a series of interesting analyses on the task encoding tokens they find, which I think would be beneficial to LLM understanding.

**Weaknesses:**

My main reservation with this paper is its framing, since one of the main pitch is that non-content tokens are "task encoding" when some results and discussions in the paper alludes to how these tokens mostly serve as an information anchor for carrying information from content tokens, and the representation of the non-content tokens themselves may not be "task encoding" so this is misleading.

The contrast between representation-level and token-level ablations should be signposted earlier in the paper, since it current reads as an abrupt change in findings or interpretation of the ablations. The token-level ablation seems to suggest that contrary to what was discussed in the representation-level ablation, the representations of template and stopword tokens alone do not encode the task well, and it is more from the information of content tokens being propagated to template and stopword tokens that seems relevant to ICL performance. For this reason, I am not convinced by one of the main claims in the paper that template and stopword tokens are "task encoding".

Also, what is the distribution of the number of tokens in each type, and how did you control for the number of tokens being ablated when analyzing the contribution of each type? For example, how do you ensure that ablating template and stopword tokens leads to a drop in performance simply because you're possibly taking a larger set of tokens to ablate? It would be also helpful to have a random baseline where you ablate random tokens to see how much ICL score drops.

**Questions:**

I suggest revisiting the framing of the paper in light of what I mentioned in the above section, and some of my questions are also included in the above section.

---

> ### Author Response · Authors · 2024-11-25
> **Author response to Reviewer 1DaS (1/2)**
>
> Thank you very much to reviewer 1DaS for your thoughtful feedback. We appreciate that the reviewer acknowledges that investigating the contributions of different input tokens to ICL is an interesting study in the broader theme of works attempting to better understand ICL. We also thank the reviewer for thinking that our findings and analyses are interesting and could be beneficial to LLM understanding. We address the concerns of the reviewer as follows:
>
> ## On the Framing of Non-Content Tokens as "Task Encoding":
>
> **Reviewer’s comment:** The term "task encoding" may be misleading, as non-content tokens may primarily serve as information anchors carrying content-token information rather than encoding the task directly.
>
> **Response:** We appreciate the reviewer’s thoughtful feedback on the term "task encoding." Our intention was to gauge a token’s task encoding nature by measuring its role (and the role of its representation) in directly affecting the downstream task performance from the perception of large language models. Our results show that the representations of content tokens hardly directly affect task performance while those of the non-content tokens largely do. In this sense, even though the representations of content tokens may contain information related to the task, our results suggest that this information is only activated/encoded in some task-solving procedure when it is aggregated by non-content tokens. To further clarify this, **we revised the terminology and the framing of our draft, signposted earlier the token-level ablations at the start of Section 4 and Section 4.3.** We also emphasized that non-content tokens primarily serve as anchors that facilitate task-relevant information flow from content tokens. This reframing aligns with the reviewer’s observation and ensures that the contribution of each token type is accurately represented.
>
> ## On the concern of  template and stopword tokens being "task encoding":
>
> **Reviewer’s comment:**  the representations of template and stopword tokens alone do not encode the task well, and it is more from the information of content tokens being propagated to template and stopword tokens that seems relevant to ICL performance. For this reason, I am not convinced by one of the main claims in the paper that template and stopword tokens are "task encoding".
>
> **Response:** Thanks for the valuable comments. We believe that we are aligned in terms of the main findings and implicitation of our experiments, and this concern of the reviewer and the opinions in our paper are actually different views of the same ideas. This is also related to the definition of task-encoding tokens that we discussed in the above rebuttal. Here, we further explain that **1)** We believe that the token-level and the representation-level ablations provide complementary analyses. The representation-level ablations identify the representations of non-content tokens as directly influencing task performance.  The token-level ablations lead us to hypothesize that it is in these representations where information related to solving the task, potentially acquired by aggregating and analyzing informationfound in content token representations, gets encoded and used.  What this concretely tells us is that if a prompt designer wanted to improve their prompt by adding more ICL demonstrations within a limited attention window size then they should feed additional demonstrations to a LLM (keeping both content and non-content tokens) but only conserve the representations of non-content tokens. We believe this conclusion suggests that, practically speaking, the representations of non-content tokens aggregate task-related information and encode this aggregated information in a way which can be used by the LLM to solve the task. **2)** The reviewer mentioned that “the representations of template and stopword tokens alone do not encode the task well.” We fully agree with this observation. However, in standard ICL settings, content tokens are always there to contribute their information, which is aggregated into the representations of non-content tokens, enabling them to become task-encoding. The token-level ablation is a specific experimental setting we propose to verify whether the information from content tokens contributes to the task-encoding representations of non-content tokens.

---

> ### Author Response · Authors · 2024-11-25
> **Author response to Reviewer 1DaS (2/2)**
>
> ## On the concern of the distribution of the number of tokens in each type:
>
> **Reviewer’s comment:** It’s unclear how the distribution of token types was controlled for, and the lack of a random baseline for ablations raises concerns about the results' validity.
>
> **Response:** We appreciate this insightful point! To address the concerns about token distribution, we include a detailed breakdown of token counts across the categories (content, stopword, and template) as follows and add the data to **Table 24 of the Appendix L in this revision**. As we can see, the set of template tokens and the stopword tokens is quite small as compared to the set of content tokens, which proves that our findings are not simply a result of ablating a larger set of tokens. Additionally, to control for potential size-based biases, we also implemented a random ablation baseline by randomly selecting the same number of tokens for each token type and letting the test example only attend to their representations, shown in **Table 25 of the Appendix L**. We also included a baseline of random selecting a certain number of tokens from all the tokens as a comparison.
>
> Results demonstrate that when the model is exposed to an equal number of each type of token representation, the performance consistently improves with template and stopword tokens, outperforming both content tokens and the random baseline. In contrast, models attending to the same number of content tokens consistently underperform relative to the random baseline. Additionally, all results improve when more tokens are included in the attention of test examples. This experiment further supports our claim that template and stopword tokens are more likely to serve as task-encoding tokens.
>
> Thanks again for your hard work and kind reviews! We sincerely hope that our clarification and revisions could resolve your concerns.

---

### Official Review · Reviewer_EzPE · 2024-11-10

**Soundness:** 2
**Presentation:** 2
**Contribution:** 3
**Rating:** 6
**Confidence:** 3

**Summary:**

This paper investigates how large language models (LLMs) handle in-context learning (ICL) by focusing on the role of different token types in few-shot learning prompts. The authors identify three main token types within an ICL prompt—content, stopword, and template tokens—and assess how each impacts model performance. Through a combination of token-level and representation-level ablation studies, they find that template and stopword tokens, rather than content tokens, significantly influence the model's task-solving ability. This finding contradicts typical expectations, as content tokens carry the primary information. They further analyze characteristics of these task-encoding tokens, identifying lexical meaning, repetition, and structural cues as key factors in their importance. The authors conclude that the roles of different tokens are nuanced and that stopwords and template tokens aggregate contextual information that supports task-solving within LLMs.

**Strengths:**

The paper addresses a timely question in ICL, offering new insights into how LLMs encode and use task information. By combining token-level and representation-level ablations, the paper provides evidence for its conclusions about task-encoding tokens. The findings are validated across different model sizes and datasets, enhancing the robustness of the results.

The division of token types into content, stopword, and template tokens is a useful categorization that contributes to understanding which elements of the prompt have greater impacts on LLM performance. This is an interesting research question and the paper’s insights may at some point be used to design prompts for LLMs, as the identified characteristics of task-encoding tokens (lexical meaning, repetition, structural cues) may offer guidelines for practitioners. However, this paper does not go all the way there.

**Weaknesses:**

The definition of task-encoding tokens, based on performance degradation when excluded, could be more conceptually grounded. Additional theoretical justification might enhance clarity of the paper. I found this part a bit ad-hoc.

While the paper briefly mentions the findings’ implications, a deeper discussion on potential applications in prompt engineering or model robustness can significantly add value to the paper.

**Questions:**

NA

---

> ### Author Response · Authors · 2024-11-25
> **Author response to Reviewer EzPE**
>
> Thank you very much to reviewer EzPE for the kind review and suggestions. We thank the reviewer for acknowledging our work offers new insights into how LLMs encode and use task information. We also appreciate that the reviewer points out that our division of token types into content, stopword, and template tokens is a useful categorization that contributes to understanding which elements of the prompt have greater impacts on LLM performance. We are also happy to see that the reviewer thinks the identified characteristics of task-encoding tokens (lexical meaning, repetition, structural cues) may offer guidelines for practitioners. We address the concerns of the reviewer as follows:
>
> ## On the definition of the task-encoding tokens:
>
> **Reviewer’s comment:** The definition of task-encoding tokens, based on performance degradation when excluded, could be more conceptually grounded. Additional theoretical justification might enhance clarity of the paper.
>
> **Response:** We appreciate the reviewer’s suggestion to clarify the conceptual basis for task-encoding tokens. We have added a more conceptually grounded and elaborated definition **in Section 4.1 to the current revision**. Conceptually, task-encoding tokens are the tokens whose representations encode the task-solving procedures needed to solve tasks. From an information-theoretic standpoint, the representations of these tokens can be considered as carrying high mutual information with respect to task-specific objectives and their corresponding procedures. However, these procedures cannot be deduced directly from the representations constructed by the model. As a practical proxy, we take task-encoding tokens to be those tokens whose representations LLMs directly depend on to achieve high-level performance. As a result, our formal definition establishes task-encoding tokens as those whose representation inclusion/exclusion significantly affects task performance as measured with an accuracy metric. In addition, we have drawn more explicitly with prior work in token representations (Todd et al. 2023, Hendel et al. 2023) who operationalize similar definitions with inclusion/exclusion analysis methods. We hope this clarification provides a stronger theoretical foundation for this approach.
>
> [1] Todd et al. (2023), Function Vectors in Large Language Models.
>
> [2] Hendel et al. (2023), In-Context Learning Creates Task Vectors.
>
>
> ## On potential applications:
>
> **Reviewer’s comment:** a deeper discussion on potential applications in prompt engineering or model robustness can significantly add value to the paper.
>
> **Response:** Thank you very much for the valuable feedback! In the initial submission, we included a preliminary discussion on potential applications of our findings **in Appendix O**, exploring areas like designing optimized in-context learning (ICL) prompts, enhancing ICL with more demonstrations, and improving long-sequence processing. Based on your suggestion, **we have now expanded this section** to provide a deeper and more concrete analysis of the principles for applying our findings. Specifically, we discuss how identifying task-encoding tokens can streamline prompt creation by helping practitioners focus on impactful tokens, thereby improving prompt efficiency and predictability. We also highlight applications in model robustness, where leveraging these insights can help mitigate performance variability caused by minor prompt modifications. We also expand the discussion of the current discussion on long-sequence processing and ICL with more demonstrations. We hope these additions address your concerns and provide clearer insights into the application value of our work.
>
> We also have a more concrete example where we found examples that especially benefited from our findings in our preliminary explorations for another project. Specifically, we observed that using the default template from OpenAI—where all input prompts are placed in the slot following “User:” and the response is generated after “System:” (shown as Template 1 below)—often resulted in unsatisfactory performance for in-context learning with LLMs. This is likely because the characteristics analyzed in our study are disrupted in this setup. Guided by our insights, we found that structuring the prompt in a more repetitive and organized way significantly improved performance. For example, when we switched to Template 2 (shown below), the few-shot learning performance using Llama2-70b-chat **improved from 31.30 to 53.43 on the 3-shot SuperNI dataset and from 36.08 to 57.79 on the 5-shot MMLU dataset.** Such suboptimal prompt designs are common and could benefit substantially from the insights provided by our work. Overall, we believe this example demonstrates how our findings can be applied to create more effective ICL prompts.
>
> Thanks again for your hard work and kind reviews! We sincerely hope that our clarification and revisions could resolve your concerns.

---

> > ### Author Response · Authors · 2024-11-25
> > **Template examples used in the experiments of potential applications.**
> >
> > ## Template 1:
> >
> > **User:**
> >
> > Task input: [demonstration input]
> >
> > Task output: [demonstration label]
> >
> >
> > Task input: [demonstration input]
> >
> > Task output: [demonstration label]
> >
> >
> > Task input: [demonstration input]
> >
> > Task output: [demonstration label]
> >
> >
> > Task input: [test example input]
> >
> > Task output:
> >
> > **System:**
> >
> >  [predicted answer]
> >
> >
> > ---
> >
> > ## Template 2:
> >
> > **User:**
> >
> > [demonstration input]
> >
> > **System:**
> >
> > [demonstration label]
> >
> > **User:**
> >
> > [demonstration input]
> >
> > **System:**
> >
> > [demonstration label]
> >
> > **User:**
> >
> > [demonstration input]
> >
> > **System:**
> >
> > [demonstration label]
> >
> > **User:**
> >
> > [test input]
> >
> > **System:**
> >
> >  [predicted answer]

---

> > > ### Comment · Reviewer_EzPE · 2024-11-28
> > > **Thanks**
> > >
> > > Thanks for your additional work in the new experiments. I have slightly improved my score based on these updates. I still think though that we can be a bit more rigorous here in the next version.

---

> > > > ### Author Response · Authors · 2024-11-28
> > > >
> > > > Thank you very much for your thoughtful feedback and for raising your score based on the new experiments! We appreciate your careful consideration of the updates we made. We also acknowledge your suggestion for further rigor and will definitely take it into account as we prepare the next version of the paper. Your insights are invaluable in helping us improve the work.

---

### Author Response · Authors · 2024-12-03
**Summary of the author response**

We are deeply grateful to all the reviewers for their thorough reviews and insightful suggestions to improve our manuscript! We appreciate the many positive remarks we received, such as:

- [EzPE] “The paper addresses a timely question in ICL, offering new insights into how LLMs encode and use task information.”, “The findings are validated across different model sizes and datasets, enhancing the robustness of the results.”

- [1DaS] “ ….. is an interesting study in the broader theme of works attempting to better understand ICL.”, “... is an interesting way to conduct this analysis”,”The paper also performs a series of interesting analyses on the task encoding tokens they find, which I think would be beneficial to LLM understanding.”

- [SovD] “This is an interesting problem and well-motivated.”, “Good descriptions of most methods and experiments”, “Thorough analyses”, “Interesting result”, “Overall I think this is a pretty good paper with interesting results.”

- [twTo] “ could be helpful for the further understanding and usage of ICL.”, “Some findings are novel and interesting.”

Following the suggestions of all the reviewers, we have made further revisions to the draft and included additional experimental results to enhance both the **presentation** and the **soundness** of the findings in the paper during the discussion period, specifically:

- We have modified the framing in Sections 4, 4.1, and 4.3 to better introduce the definition of task-encoding tokens, as suggested by Reviewer EzPE and Reviewer 1DaS.

- We have expanded the discussion on the potential applications of our findings about task-encoding tokens in Appendix O, along with a concrete example of how prompt engineering could benefit from the insights in our paper, as suggested by Reviewer EzPE and Reviewer SovD.

- We have added a series of experiments regarding token number statistics and the masking of random tokens with specific numbers in Appendix L, as suggested by Reviewer 1DaS and Reviewer SovD. These results further support our main findings.

- We have revised the first two figures in the paper and their corresponding descriptions to clarify the token-level ablation, as suggested by Reviewer SovD. Additionally, we have provided further results to justify our experimental choices in the current token-level ablation experiments.

- We have included the results for models after supervised fine-tuning in Appendix E and will add experiments for larger quantized models in the next revision, as suggested by Reviewer SovD.

- We have expanded all appendices that were too brief in the current revision, as suggested by Reviewer SovD.

- We have added another set of experiments for the question answering tasks with all the models tested in our paper (i.e., OpenLlama 3B, LLaMA 7B, 13B, 33B, LLaMA 2 7B, 13B, and Mistral 7B), as suggested by Reviewer twTo. These results further support our main findings regarding task-encoding tokens.

We sincerely believe that these updates will help better deliver the findings related to task-encoding tokens to the ICLR community.

Thank you very much.


Best regards,

Paper 7037 Authors

---

### Meta-Review · Area_Chair_dXLh · 2024-12-17

**Metareview:**

The authors study how LLMs perform ICL and generalize from the provided examples. They show that LLMs rely more on uninformative "template" and "stopword" tokens for task encoding when doing ICL, unlike humans who focus on content. These task-encoding tokens gather information from content tokens and are characterized by lexical meaning, repetition, and structural cues, highlighting a key difference between human and LLM learning. The reviewers appreciate the importance of the problem, the novel approach, and thorough analysis. However, they also raise several concerns, primarily around the definition or framing of the task and concerns around evaluation (regarding ablations, limited comparison to baselines, data diversity, and some methodological choices). The authors provide detailed responses and conduct experiments to address the concerns resulting in two reviewers increasing their score and two not being convinced.

**Additional Comments On Reviewer Discussion:**

The discussions mostly focus around clarifications and experiment design decisions that the authors address in detail. However, some concerns still remain and not all reviewers are convinced. Given the overall reviews and scores and taking into account the discussions I believe this paper still needs some improvements before being ready for publication.

---

### Decision · Program_Chairs · 2025-01-22

Reject